# Exploiting metabolic vulnerability in glioblastoma using a brain-penetrant drug with a safe profile

Audrey Burban [ID][1], Cloe Tessier [ID][2,12], Mathieu Larroquette [ID][1,12], Joris Guyon [ID][3,4], Cloe Lubiato[2], Mathis Pinglaut [ID][1], Maxime Toujas [ID][2], Johanna Galvis[1], Benjamin Dartigues[5], Emmanuelle Georget [ID][2], H Artee Luchman[6,7], Samuel Weiss[6,7], David Cappellen [ID][2], Nathalie Nicot [ID][8], Barbara Klink [ID][8,9,10,11], Macha Nikolski [ID][1,5], Lucie Brisson [ID][2], Thomas Mathivet[2], Andreas Bikfalvi [ID][2✉], Thomas Daubon [ID][1✉] & Ahmad Sharanek [ID][2✉]

## Abstract

**Glioblastoma is one of the most treatment-resistant and lethal cancers, with a subset of self-renewing brain tumour stem cells (BTSCs), driving therapy resistance and relapse. Here, we report that mubritinib effectively impairs BTSC stemness and growth. Mechanistically, bioenergetic assays and rescue experiments showed that mubritinib targets complex I of the electron transport chain, thereby impairing BTSC self-renewal and proliferation. Gene expression profiling and Western blot analysis revealed that mubritinib disrupts the AMPK/p27$^{Kip1}$ pathway, leading to cell-cycle impairment. By employing in vivo pharmacokinetic assays, we established that mubritinib crosses the blood-brain barrier. Using preclinical patient-derived and syngeneic models, we demonstrated that mubritinib delays glioblastoma progression and extends animal survival. Moreover, combining mubritinib with radiotherapy or chemotherapy offers survival advantage to animals. Notably, we showed that mubritinib alleviates hypoxia, thereby enhancing ROS generation, DNA damage, and apoptosis in tumours when combined with radiotherapy. Encouragingly, toxicological and behavioural studies revealed that mubritinib is well tolerated and spares normal cells. Our findings underscore the promising therapeutic potential of mubritinib, warranting its further exploration in clinic for glioblastoma therapy.**

**Keywords** Metabolic Reliance; Oxidative Phosphorylation; Radiotherapy; Hypoxia; Reactive Oxygen Species
**Subject Categories** Cancer; Metabolism; Stem Cells & Regenerative Medicine

## Introduction

Glioblastoma (GB) is the most common and aggressive primary malignant tumour in the adult brain, leading to the majority of ~200,000 deaths related to the central nervous system tumours worldwide each year (Alexander et al, 2018; Brain and Other, 2019). The current standard of care for GB patients includes surgical excision of the tumour followed by ionizing radiation (IR) and chemotherapy (Stupp et al, 2005). Despite intense efforts at targeting various signalling pathways, putative driver mutations and angiogenesis mechanisms, survival has not substantially improved beyond 18 months following diagnosis (Gilbert et al, 2014; Stupp et al, 2017). The main challenges underlying therapeutic failure are rooted to its inter- and intra-tumoural heterogeneity, resistance to therapy and relapse.

Intra-tumoural heterogeneity is thought to be promoted by the brain tumour stem cells (BTSCs), a population of self-renewing, multipotent, and tumour-initiating stem cells that undergo dynamic state transitions (Prager et al, 2020; Singh et al, 2004). BTSCs invade extensively throughout the brain, develop a hyper aggressive phenotype, resist therapies, and lead to tumour recurrence (Chen et al, 2012). Therefore, developing novel approaches to suppress BTSCs is at the forefront of efforts to fight GB.

Metabolic dysregulation is one of the most important hallmarks of GB and contributes to therapy resistance (Agnihotri and Zadeh, 2016; Lokody, 2014; Strickland and Stoll, 2017). For decades, tumour metabolism was believed to rely heavily on aerobic glycolysis, a phenomenon known as the Warburg effect (Warburg, 1956). However, in line with the complex heterogeneity of GB, recent reports have shown the existence of multiple metabolic dependencies in GB, where mitochondrial respiration is highlighted as an essential alternative source of energy and has been shown to play a crucial role in GB tumourigenesis (Hoang-Minh et al, 2018;

[1]University of Bordeaux, CNRS, IBGC, UMR5095, Bordeaux, France. [2]University of Bordeaux, INSERM, UMR1312, BRIC, BoRdeaux Institute of onCology, Bordeaux, France. [3]CHU of Bordeaux, Service de Pharmacologie Médicale, Bordeaux, France. [4]University of Bordeaux, INSERM, BPH, U1219, Bordeaux, France. [5]Bordeaux Bioinformatic Center CBiB, University of Bordeaux, Bordeaux, France. [6]Department of Cell Biology and Anatomy, University of Calgary, Calgary, Alberta, Canada. [7]Arnie Charbonneau Cancer Institute and Hotchkiss Brain Institute, University of Calgary, Calgary, Alberta, Canada. [8]LuxGen Genome Center, Luxembourg Institute of Health, Laboratoire national de santé, Dudelange, Luxembourg. [9]National Center of Genetics (NCG), Laboratoire National de Santé (LNS), Dudelange, Luxembourg. [10]Department of Cancer Research (DoCR), Luxembourg Institute of Health (LIH), Luxembourg 1526, Luxembourg. [11]Department of Life Sciences and Medicine, University of Luxembourg, Esch-sur-Alzette, Luxembourg. [12]These authors contributed equally: Cloe Tessier, Mathieu Larroquette. ✉E-mail: andreas.bikfalvi@u-bordeaux.fr; thomas.daubon@u-bordeaux.fr; ahmad.charanek@u-bordeaux.fr

Talasila et al, 2017). Recent investigations have documented that GB cells also acquire functional mitochondria via tunnelling from other cancerous counterparts (Nakhle et al, 2023; Pinto et al, 2021) or non-malignant host cells in the tumour microenvironment (Salaud et al, 2020; Watson et al, 2023), resulting in enhanced metabolic activity and augmented tumourigenicity. Single-cell RNA expression profiling has identified a mitochondrial subtype in GB tumours that exhibits a mitochondrial signature and is dependent on mitochondrial oxidative phosphorylation (OXPHOS) (Garofano et al, 2021). Initially, it was suggested that BTSCs rely heavily on OXPHOS to address their energy demands (Janiszewska et al, 2012; Vlashi et al, 2011). Recent studies have reinforced this paradigm and suggested that at least certain subtypes of BTSCs, including slow-cycling BTSCs, rely on mitochondrial respiration (Hoang-Minh et al, 2018). This highlights that molecules that are able to affect mitochondrial respiration might be an interesting effective therapeutic strategy to eradicate BTSCs and suppress this devastating tumour.

Due to the dependency of multiple tumours on mitochondrial respiration (Kuntz et al, 2017; Stuani et al, 2021; Sullivan et al, 2015; Xu et al, 2020), targeting this metabolic pathway has emerged as an attractive anticancer strategy (Janku et al, 2022; Schockel et al, 2015; Yap et al, 2023). However, the translation of currently available OXPHOS inhibitors into clinical practice has been hindered by poor potency, e.g., metformin and other biguanides (Bridges et al, 2014; Dykens et al, 2008; Janku et al, 2022), or high toxicity, e.g., IACS-010759, oligomycin, rotenone, BAY 87-2243 and ASP4132 (Betarbet et al, 2000; Janku et al, 2021; Xu et al, 2020; Yap et al, 2023). Here, we investigated the impact of mubritinib on the fate and tumourigenesis of BTSCs. Mubritinib was initially described as an ERBB2 inhibitor but was recently found to inhibit OXPHOS in acute myeloid leukaemia (AML) models (Baccelli et al, 2019). We report for the first time that mubritinib potently impairs the stemness and growth of patient-derived BTSCs by targeting mitochondrial respiration. Importantly, we showed that mubritinib is a brain penetrant drug that significantly suppresses GB tumour progression and sensitizes their response to current standard radio and chemotherapy. By performing profound toxicological and behavioural studies in preclinical animal models, we showed that mubritinib has a well-tolerated and safe profile and holds promise for future clinical trials.

## Results

### Mubritinib is a potent inhibitor of OXPHOS that inhibits BTSC growth

To determine the impact of mubritinib on overall mitochondrial respiration in BTSCs, we examined the oxygen consumption rate (OCR) as a measure of electron transport chain (ETC) activity. Multiple patient-derived BTSCs with variable genetic backgrounds (Appendix Table S1), were treated with different concentrations of mubritinib and subjected to the ultra-sensitive real-time Resipher system (Lucid Scientific) or to the high-resolution respirometer Oroboros to measure the OCR. These analyses revealed a significant and dose-dependent reduction in basal mitochondrial respiration in mubritinib-treated BTSCs compared to vehicle control BTSCs, starting at very low concentrations in the range

of 20 nM (Figs. 1A–C and EV1A). Furthermore, we also evaluated the impact of mubritinib on the bioenergetic fitness of BTSCs by assessing the maximal mitochondrial and spare respiratory capacity (SRC) and found a significant decrease in these respiration parameters upon mubritinib treatment, suggesting that mubritinib alters the mitochondrial reserve in BTSCs (Figs. 1D,E and EV1A).

Having established that mubritinib impairs OXPHOS in BTSCs and given the possible reliance of subsets of BTSCs on OXPHOS, we examined whether mubritinib impairs BTSCs growth. We applied increasing concentrations of mubritinib (20 to 500 nM) to 11 patient-derived BTSC lines and a murine BTSC line (mGB2). Interestingly, we observed significant inhibition of BTSC growth in all the BTSCs tested at very low concentrations of mubritinib in the nM range following 7 days of treatment (Figs. 1F and EV1B). We next asked whether the decrease in cell number in response to mubritinib is due to the inhibition of proliferation or induction of cell death. Annexin V/propidium iodide (PI) flow cytometry analysis revealed that there was no significant induction of cell death or early apoptosis in any of the BTSCs tested, even at concentrations up to 10 µM of mubritinib (Fig. EV1C,D). On the other hand, by performing a 5-ethynyl-2′-deoxyuridine (EdU) assay, we observed a remarkable reduction in the number of EdU-incorporating cells in mubritinib-treated BTSCs (#53, #73 and #147) (Fig. 1G–J). These results indicate that the decrease in cell number by mubritinib is not due to cell death but is likely attributed to the inhibition of proliferation.

Noteworthy, cell viability analysis revealed that BTSCs exhibit different sensitivities to mubritinib treatment (Fig. 1F), therefore, we asked whether this difference in sensitivity is correlated with their respiration capacity. We observed a positive correlation between both the basal OCR and the maximal OCR with the sensitivity to mubritinib (Figs. 1K and EV1E–H), suggesting that the inhibitory effect of mubritinib on BTSC growth could be due to OXPHOS inhibition. To directly link the effect of mubritinib on BTSC growth to OXPHOS inhibition, we performed rescue experiments using the *Saccharomyces cerevisiae* rotenone-insensitive NDI1 gene (Fig. 1L). The expression of NDI1 gene was confirmed by RT-qPCR analysis (Fig. EV1I–K). We found that ectopic expression of the NDI1 gene in multiple BTSCs (#53, #73 and #147), rescued largely the basal, maximal and SRC of the BTSCs inhibited by mubritinib (Figs. 1M–O and EV1L–N), suggesting that mubritinib impairs complex I activity to inhibit mitochondrial respiration in BTSCs. Then, we examined whether the NDI1 gene could reverse the effect of mubritinib treatment on BTSC growth. Interestingly, we found that ectopic expression of NDI1 largely rescued the proliferation of BTSCs after mubritinib treatment (Figs. 1P–R and EV1O). In support, we employed a genetic approach by which we silenced the NADH:ubiquinone oxidoreductase core subunit S7 of complex I (*NDUFS7*) using siRNA (Fig. EV1P,Q). KD of *NDUFS7* phenocopies the inhibition of proliferation as mubritinib (Fig. EV1R,S). Importantly, treating the *NDUFS7* KD cells with mubritinib did not induce additional significant effects (Fig. EV1R,S), confirming that mubritinib exerts its effects through complex I of the ETC. Altogether, these data confirm that mubritinib exerts its effects on BTSC growth through inhibition of complex I of the ETC and subsequent halting of mitochondrial respiration.

Since previous reports suggested that differentiated GB cells have a glycolytic profile and rely to lower extent than GB stem cells on OXPHOS (Vlashi et al, 2011), we thus asked if differentiated GB cells are less sensitive to mubritinib than stem cells. We, therefore,

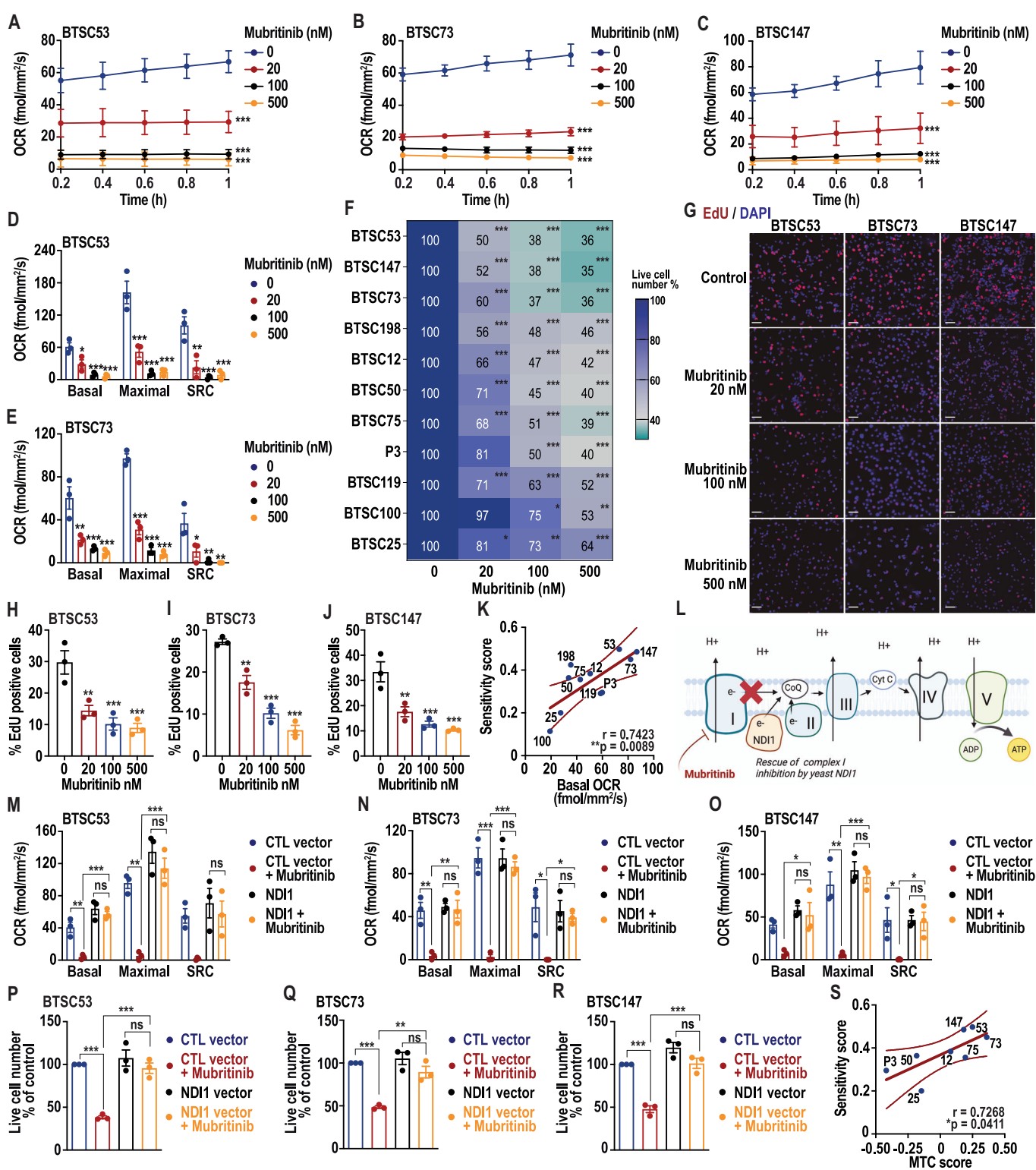

examined the effect of mubritinib on the differentiated BTSC progeny (Fig. EV1T). Our results revealed that, in the four BTSCs tested (#12, #53, #73 and #147) the differentiated progeny were less sensitive to mubritinib than their stem counterparts (Fig. EV1U–X).

In a set of analysis, we attempted to correlate the BTSC sensitivity to mubritinib to their transcriptional subtype signatures (Fig. EV2A,B). By interrogating the RNA-seq data of BTSCs, we analysed their gene expression profiles and generated classification scores based on the basic Wang GB signatures (classical, proneural,

◀ **Figure 1.  Mubritinib alters mitochondrial respiration through complex I inhibition and impairs BTSC growth.**

(A–C) Patient-derived BTSCs, BTSC53 (A), BTSC73 (B) and BTSC147 (C) were subjected to real-time Resipher analysis to measure the basal oxygen consumption rate (OCR) following mubritinib treatment. Data are presented as the means ± SEM, $n = 3$ independent biological experiments. Two-way ANOVA followed by Dunnett's test vs vehicle control. BTSC53 (A): $***p_{20} = 1.9e{-}10$, $***p_{100} = 1.2e{-}14$, $***p_{500} = 1.2e{-}14$. BTSC73 (B): $***p_{20} = 1.2e{-}14$, $***p_{100} = 1.2e{-}14$, $***p_{500} = 1.2e{-}14$. BTSC147 (C): $***p_{20} = 2.2e{-}11$, $***p_{100} = 1.2e{-}14$, $***p_{500} = 1.2e{-}14$. (D, E) Basal mitochondrial respiration, maximal mitochondrial respiration and spare respiratory capacity (SRC) were measured in BTSC53 (D) and BTSC73 (E) following mubritinib treatment using the Resipher system. Data are presented as the means ± SEM, $n = 3$ independent biological experiments. One-way ANOVA followed by Dunnett's test vs vehicle control. BTSC53 (D): $*p_{basal\ (20)} = 0.0105$; $***p_{basal\ (100)} = 0.0005$; $***p_{basal\ (500)} = 0.0004$; $***p_{maximal\ (20)} = 0.0005$, $***p_{maximal\ (100)} = 5.1e{-}5$, $***p_{maximal\ (500)} = 5.9e{-}5$; $***p_{SRC\ (20)} = 0.0018$, $***p_{SRC\ (100)} = 0.0004$, $***p_{SRC\ (500)} = 0.0006$. BTSC73 (E): $**p_{basal\ (20)} = 0.0025$; $***p_{basal\ (100)} = 0.0007$; $***p_{basal\ (500)} = 0.0004$; $***p_{maximal\ (20)} = 2.3e{-}6$, $***p_{maximal\ (100)} = 3.2e{-}7$, $***p_{maximal\ (500)} = 2.3e{-}7$; $*p_{SRC\ (20)} = 0.021$, $**p_{SRC\ (100)} = 0.0038$, $**p_{SRC\ (500)} = 0.0031$. (F) Multiple patient-derived BTSCs, BTSC12, BTSC25, BTSC50, BTSC53, BTSC73, BTSC75, BTSC100, BTSC119, BTSC147, BTSC198 and P3, were exposed to increasing concentrations of mubritinib (0 to 500 nM) for 7 days, followed by live cell counting. Data are presented as the means ± SEM, $n = 3$ independent biological experiments. One-way ANOVA followed by Dunnett's test vs vehicle control. $***p_{BTSC12\ (20)} = 0.0003$, $***p_{BTSC12\ (100)} = 1.4e{-}5$, $***p_{BTSC12\ (500)} = 7e{-}6$; $*p_{BTSC25\ (20)} = 0.0316$, $**p_{BTSC25\ (100)} = 0.0052$, $***p_{BTSC25\ (500)} = 0.0007$; $***p_{BTSC50\ (20)} = 0.0003$, $***p_{BTSC50\ (100)} = 2.6e{-}6$, $***p_{BTSC50\ (500)} = 1e{-}6$; $***p_{BTSC53\ (20)} = 1.1e{-}5$, $***p_{BTSC53\ (100)} = 2e{-}6$, $***p_{BTSC53\ (500)} = 1e{-}6$; $***p_{BTSC73\ (20)} = 0.0007$, $***p_{BTSC73\ (100)} = 3.1e{-}5$, $***p_{BTSC73\ (500)} = 2.6e{-}5$; $***p_{BTSC75\ (20)} = 0.0007$, $***p_{BTSC75\ (100)} = 3.2e{-}5$, $***p_{BTSC75\ (500)} = 6e{-}6$; $*p_{BTSC100\ (100)} = 0.0384$, $***p_{BTSC100\ (500)} = 0.001$; $***p_{BTSC119\ (20)} = 0.0004$, $***p_{BTSC119\ (100)} = 6.9e{-}5$, $***p_{BTSC119\ (500)} = 1e{-}5$; $***p_{BTSC147\ (20)} = 9.4e{-}8$, $***p_{BTSC147\ (100)} = 1.2e{-}8$, $***p_{BTSC147\ (500)} = 8e{-}9$; $***p_{BTSC198\ (20)} = 0.0008$, $***p_{BTSC198\ (100)} = 0.0003$, $***p_{BTSC198\ (500)} = 0.0002$; $***p_{P3\ (100)} = 0.0003$, $***p_{P3\ (500)} = 0.0001$. (G–J) EdU incorporation was analysed by immunofluorescence imaging of BTSCs (#53, #73 and #147) after 4 days of mubritinib treatment at concentrations of 20 nM, 100 nM and 500 nM. Representative images of EdU (red) staining are shown (G). Nuclei were stained with DAPI (blue). Scale bar = 50 µm. The number of EdU-positive cells was quantified with Fiji software in BTSC53 (H), BTSCS73 (I) and BTSC147 (J). Data are presented as the means ± SEM, $n = 3$ independent biological experiments. One-way ANOVA followed by Dunnett's test vs vehicle control. BTSC53 (H): $**p_{20} = 0.0045$, $***p_{100} = 0.001$, $***p_{500} = 0.0006$. BTSC73 (I): $**p_{20} = 0.0012$, $***p_{100} = 2.3e{-}5$, $***p_{500} = 5e{-}6$. BTSC147 (J): $**p_{20} = 0.0033$, $***p_{100} = 0.0006$, $***p_{500} = 0.0003$. (K) Pearson correlation analysis between the basal oxygen consumption rate (OCR) and BTSC sensitivity score to mubritinib following 7 days of treatment was performed. (l) A schematic diagram of the mitochondrial electron transport chain with ectopic NDI1 expression is presented. (M–O) BTSC53 (M), BTSC73 (N) and BTSC147 (O) expressing the control (CTL) or NDI1 vector were treated with vehicle control or 500 nM mubritinib and subjected to Resipher analysis to measure the basal OCR, maximal respiration and SRC. Data are presented as the means ± SEM, $n = 3$ independent biological experiments. One-way ANOVA followed by Tukey's test. BTSC53 (M): $**p_{basal\ (CTL\ vs\ mubritinib)} = 0.0049$, $***p_{basal\ (mubritinib\ vs\ NDI1\ +\ mubritinib)} = 0.0004$, $p_{basal\ (NDI1\ vs\ NDI1\ +\ mubritinib)} = 0.7841$; $**p_{maximal\ (CTL\ vs\ mubritinib)} = 0.0011$, $***p_{maximal\ (mubritinib\ vs\ NDI1\ +\ mubritinib)} = 0.0003$, $p_{maximal\ (NDI1\ vs\ NDI1\ +\ mubritinib)} = 0.5223$; $p_{SRC\ (NDI1\ vs\ NDI1\ +\ mubritinib)} = 0.8773$. BTSC73 (N): $**p_{basal\ (CTL\ vs\ mubritinib)} = 0.0041$, $**p_{basal\ (mubritinib\ vs\ NDI1\ +\ mubritinib)} = 0.0035$, $p_{basal\ (NDI1\ vs\ NDI1\ +\ mubritinib)} = 0.9933$; $***p_{maximal\ (CTL\ vs\ mubritinib)} = 6.1e{-}5$, $***p_{maximal\ (mubritinib\ vs\ NDI1\ +\ mubritinib)} = 0.0001$, $p_{maximal\ (NDI1\ vs\ NDI1\ +\ mubritinib)} = 0.8470$; $*p_{SRC\ (CTL\ vs\ mubritinib)} = 0.016$, $*p_{SRC\ (mubritinib\ vs\ NDI1\ +\ mubritinib)} = 0.0467$, $p_{SRC\ (NDI1\ vs\ NDI1\ +\ mubritinib)} = 0.9615$. BTSC147 (O): $*p_{basal\ (mubritinib\ vs\ NDI1\ +\ mubritinib)} = 0.0192$, $p_{basal\ (NDI1\ vs\ NDI1\ +\ mubritinib)} = 0.9528$; $**p_{maximal\ (CTL\ vs\ mubritinib)} = 0.0018$, $***p_{maximal\ (mubritinib\ vs\ NDI1\ +\ mubritinib)} = 0.0009$, $p_{maximal\ (NDI1\ vs\ NDI1\ +\ mubritinib)} = 0.9472$; $*p_{SRC\ (CTL\ vs\ mubritinib)} = 0.0335$, $*p_{SRC\ (mubritinib\ vs\ NDI1\ +\ mubritinib)} = 0.0396$, $p_{SRC\ (NDI1\ vs\ NDI1\ +\ mubritinib)} = 0.9993$. (P–R) The number of live BTSC53 (P), BTSC73 (Q) and BTSC147 (R) expressing control vector or NDI1 after 7 days of 500 nM mubritinib treatment was measured. Data are presented as the means ± SEM, $n = 3$ independent biological experiments. One-way ANOVA followed by Tukey's test. BTSC53 (P): $***p_{CTL\ vs\ mubritinib} = 0.0003$, $***p_{mubritinib\ vs\ NDI1\ +\ mubritinib} = 0.0005$, $p_{NDI1\ vs\ NDI1\ +\ mubritinib} = 0.4954$. BTSC73 (Q): $***p_{CTL\ vs\ mubritinib} = 0.0004$, $**p_{mubritinib\ vs\ NDI1\ +\ mubritinib} = 0.0017$, $p_{NDI1\ vs\ NDI1\ +\ mubritinib} = 0.1875$. BTSC147 (R): $***p_{CTL\ vs\ mubritinib} = 0.0002$, $***p_{mubritinib\ vs\ NDI1\ +\ mubritinib} = 0.0002$, $p_{NDI1\ vs\ NDI1\ +\ mubritinib} = 0.0904$. (S) Pearson correlation analysis was performed between the BTSC sensitivity score to mubritinib following 7 days of treatment and mitochondrial (MTC) transcriptional subtype signature score. Source data are available online for this figure.

and mesenchymal) using Geneset Variation Analysis (GSVA) (Fig. EV2A). By corelating the subtype scores to the sensitivity to mubritinib, no statistically significant correlations emerged between sensitivity to mubritinib and the mesenchymal or the classical signature, but we observed a negative correlation with the proneural signature (Fig. EV2C–E). Interestingly, we performed similar analysis with the pathway-based transcriptional GB signatures (proliferative/progenitor, neuronal, mitochondrial and glycolytic/plurimetabolic) (Garofano et al, 2021) (Fig. EV2B) and found a positive correlation between sensitivity to mubritinib and the mitochondrial (MTC) subtype signature (Fig. 1S), a GB transcriptional subtype that was identified to rely on OXPHOS (Garofano et al, 2021). No significant correlation emerged with the three other pathway-based transcriptional GB signatures (Fig. EV2F–H). This positive correlation suggests that the MTC-enriched GB could be more sensitive to mubritinib. However, certainly, a study with a larger panel of samples will need to be conducted to conclusively establish whether sensitivity to mubritinib is linked to any specific transcriptional GB subtype.

## Mubritinib impairs AMPK/p27[Kip1] pathway and deregulates the cell cycle

To gain mechanistic insights into the downstream signalling pathways affected by mubritinib in BTSCs, we performed RNA-seq on two patient-derived BTSC lines (#73 and #147). Gene expression profiling identified differentially expressed genes between mubritinib-treated and control BTSCs (adjusted $p$-value < 0.05) (Dataset EV1–2). Gene set enrichment analysis revealed downregulation of cell cycle-related pathways in both BTSC lines (Appendix Fig. S1A,B and Dataset EV3–4). Notably, pathways related to the AMPK/p27[Kip1] signalling axis, including cyclin D1-associated events in G1, SCF SKP2-mediated degradation of p27/p21 and E2F-mediated regulation of DNA replication, were downregulated in both BTSC73 (Fig. 2A–C) and BTSC147 following mubritinib treatment (Appendix Fig. S1C–E).

To validate the role of mubritinib in cell cycle deregulation, we performed functional cell cycle profiling via flow cytometric quantitation of DNA content by PI. A significant increase in the percentage of cells in the G1 phase and a decrease in the percentage of cells in the S phase were observed in the mubritinib-treated BTSCs (Fig. 2D,E; Appendix Fig. S1F).

Next, we tested whether mubritinib impairs the AMPK/p27[Kip1] pathway. AMPK acts as a metabolic sensor activated by metabolic stress to trigger cell cycle arrest mediated by the kinase inhibitory protein p27[Kip1] (Tuo et al, 2019). OXPHOS inhibitors have been reported to induce metabolic stress through deregulation of the ATP/ADP ratio, resulting in the activation of AMPK (Hawley et al, 2010). Therefore, we assessed AMPK phosphorylation by Western blotting and found that mubritinib treatment significantly

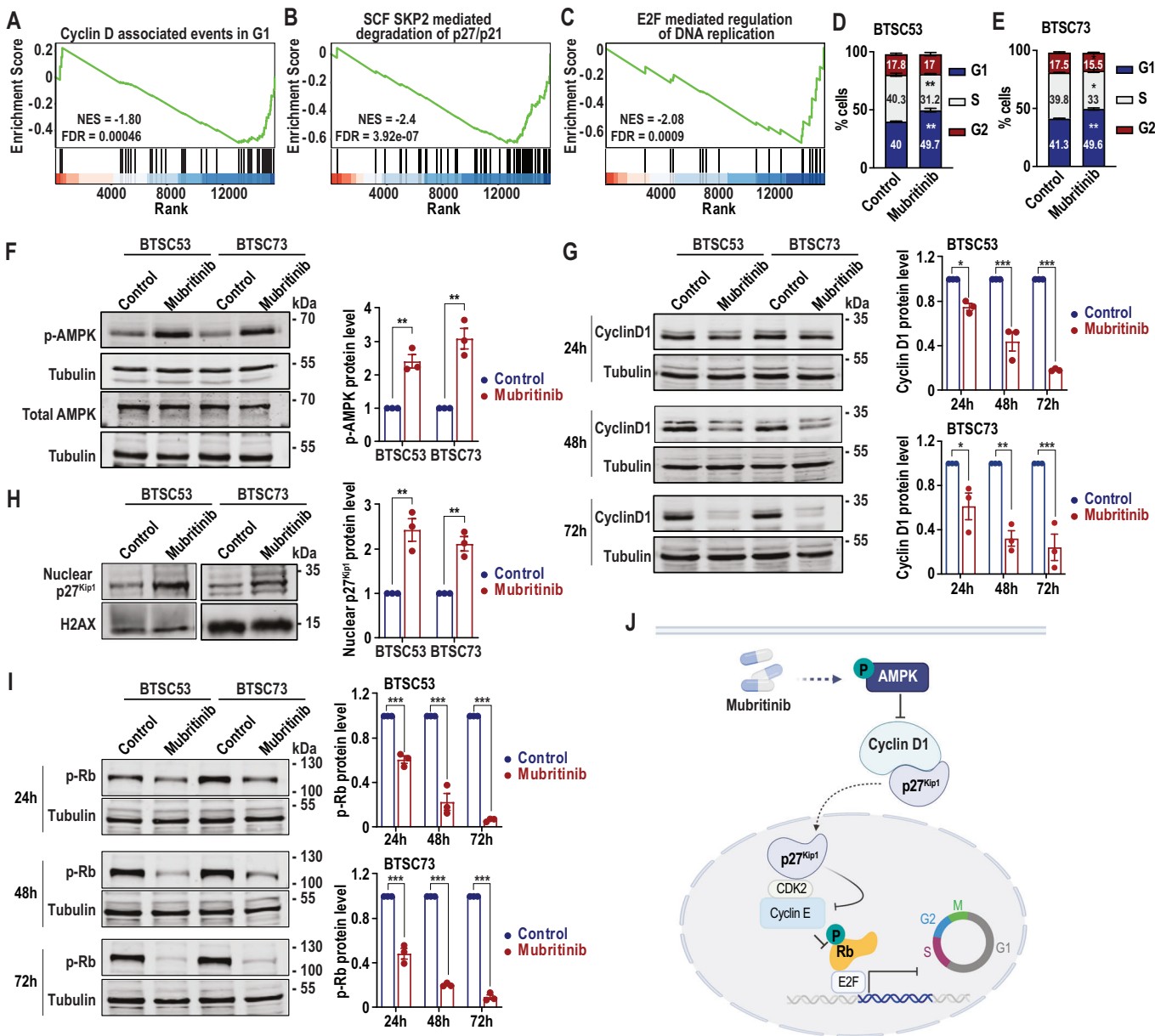

increased AMPK phosphorylation within 24 h (Fig. 2F), with sustained activation observed at 72 h (Appendix Fig. S1G).

Subsequently, we examined cyclin D1 levels, as AMPK activation regulates cyclin D1, which sequesters p27[Kip1] (Zhuang and Miskimins, 2008). We found that mubritinib treatment caused a reduction in cyclin D1 protein levels after 24 h, with more pronounced decrease observed at 48 and 72 h (Fig. 2G), suggesting release and nuclear localization of p27[Kip1]. Interestingly, we examined the levels of the kinase inhibitor p27[Kip1] in the nuclear fractions, and found higher levels in the nuclei of mubritinib-treated BTSCs (Fig. 2H). p27[Kip1] acts to inhibit cyclin E/CDK2 kinase activity and leaving the retinoblastoma protein (Rb) protein unphosphorylated. When Rb is unphosphorylated, the E2F transcription factor is inhibited from transcribing genes necessary to enter the cell cycle (Harbour and Dean, 2000). We analysed Rb protein phosphorylation levels and found a marked reduction in its phosphorylation following mubritinib treatment (Fig. 2I). Taken together, these

data highlight that mubritinib alters the AMPK/Cyclin D1/p27[Kip1]/Rb signalling cascade, impairing the cell cycle in BTSCs (Fig. 2J).

## Mubritinib impairs self-renewal and stemness-related pathways

It has been suggested that OXPHOS activity maintains BTSC stemness (Iranmanesh et al, 2021; Janiszewska et al, 2012; Sharanek et al, 2020). We thus asked if the impairment of mitochondrial respiration by mubritinib impacts BTSC self-renewal. To address this question, we subjected 10 patient-derived BTSC lines and the murine mGB2 line to extreme limiting dilution analysis (ELDA) in the absence and presence of mubritinib. ELDA showed that mubritinib treatment significantly attenuated self-renewal in all the BTSC lines tested (Figs. 3A–F and EV3A–E).

**Figure 2. Mubritinib impairs the AMPK/p27^Kip1 and cell-cycle in BTSCs.**

(A–C) Gene set enrichment analysis of deregulated genes in BTSC73 treated with 500 nM mubritinib for 24 h demonstrating enrichment of gene sets corresponding to cyclin D1-associated events in G1 (A), SCF SKP2-mediated degradation of p27/p21 (B) and E2F-mediated regulation of DNA replication (C) are shown. (D, E) Cell cycle distribution was assessed by flow cytometry after PI staining of BTSC53 (D) and BTSC73 (E) following 24 h of treatment with 500 nM mubritinib. Data are presented as the means ± SEM, $n = 3$ independent biological experiments. Unpaired two-tailed t test. BTSC53 (D): $**p_{G1} = 0.0046$, $**p_S = 0.0023$. BTSC73 (E): $**p_{G1} = 0.0014$, $*p_S = 0.0131$, $*p_{G2} = 0.0471$. (F) BTSC53 and BTSC73 were treated for 24 h with 500 nM of mubritinib or vehicle control and subjected to immunoblotting using antibodies against p-AMPK and total AMPK. Tubulin was used as loading control. Densitometric quantifications of p-AMPK protein level normalized to the corresponding loading control are presented. Data are presented as the means ± SEM, $n = 3$ independent biological experiments. Unpaired two-tailed t test. $**p_{BTSC53} = 0.0022$, $**p_{BTSC73} = 0.0025$. (G) BTSC53 and BTSC73 were treated for 24 h, 48 h or 72 h with 500 nM of mubritinib or vehicle control and subjected to immunoblotting using an antibody against cyclin D1. Tubulin was used as loading control. Densitometric quantifications of cyclin D1 protein level normalized to the corresponding loading control are presented. Data are presented as the means ± SEM, $n = 3$ independent biological experiments. One-way ANOVA followed by Dunnett's test vs vehicle control. $*p_{BTSC53\ 24h} = 0.0123$, $***p_{BTSC53\ 48h} = 6.6e{-}5$, $***p_{BTSC53\ 72h} = 4e{-}6$, $*p_{BTSC73\ 24h} = 0.0422$, $**p_{BTSC73\ 48h} = 0.0020$, $***p_{BTSC73\ 72h} = 0.001$. (H) Nuclear fractions from BTSC53 and BTSC73 treated for 72 h with 500 nM of mubritinib or vehicle control and subjected to immunoblotting using an antibody against p27^Kip1. H2AX is used as nuclear loading control. Densitometric quantifications of nuclear p27^Kip1 protein level normalized to the corresponding loading control are presented. Data are presented as the means ± SEM, $n = 3$ independent biological experiments. Unpaired two-tailed t test. $**p_{BTSC53} = 0.0049$, $**p_{BTSC73} = 0.0023$. (I) BTSC53 and BTSC73 were treated for 24 h, 48 h or 72 h with 500 nM of mubritinib or vehicle control and subjected to immunoblotting using an antibody against phosphorylated retinoblastoma protein (p-Rb). Tubulin was used as loading control. Densitometric quantifications of p-Rb protein level normalized to the corresponding loading control are presented. Data are presented as the means ± SEM, $n = 3$ independent biological experiments. One-way ANOVA followed by Dunnett's test vs vehicle control. $***p_{BTSC53\ 24h} = 0.0005$, $***p_{BTSC53\ 48h} = 3.1e{-}6$, $***p_{BTSC53\ 72h} = 7e{-}7$, $***p_{BTSC73\ 24h} = 2e{-}6$, $***p_{BTSC73\ 48h} = 7e{-}8$, $***p_{BTSC73\ 72h} = 2e{-}8$. (J) Proposed model for the mechanism by which mubritinib impairs the AMPK/p27^Kip1 axis and cell cycle in BTSCs. Source data are available online for this figure.

To further assess the impact of mubritinib on BTSC stemness, we analysed the levels of various stemness markers in multiple BTSCs (#53, #73 #147, #12, #119, P3 and mGB2) by immunoblotting. Our analysis revealed significant reduction in the protein levels of the tested stemness markers, i.e., active (cleaved Notch1), Olig2 and SOX2, in mubritinib-treated BTSCs compared to control (Figs. 3G–L and EV3F).

To determine whether the effects of mubritinib on BTSC maintenance and self-renewal is mediated through complex I inhibition of the electron transport chain, we performed rescue experiments by introducing the *NDI1* gene into BTSC lines (#53 and #73) and assessing if NDI1 expression could reverse the effects of mubritinib on BTSC stemness and self-renewal. ELDA revealed that ectopic expression of NDI1 fully rescued BTSC self-renewal in mubritinib-treated cells (Figs. 3M and EV3G). Moreover, the mubritinib-induced downregulation of stemness markers was also restored by the expression of NDI1 in BTSCs, as demonstrated by immunoblotting (Figs. 3N and EV3H). Together, these results demonstrate that mubritinib impairs BTSC maintenance via OXPHOS inhibition.

Next, we evaluated the impact of mubritinib on the non-oncogenic normal human progenitor cells (hNPCs). Strikingly, ELDA showed that, at an effective dose for BTSCs (500 nM), mubritinib had no apparent impact on self-renewal and SCF of hNPCs (Fig. 3O). Similarly, immunoblot analyses showed no reduction of stemness markers (cleaved Notch1, Olig2 and SOX2) in hNPCs following mubritinib treatment (Fig. 3P). Collectively, these data demonstrate that mubritinib selectively disrupts BTSC stemness while sparing non-oncogenic hNPCs.

## Mubritinib is a brain penetrant drug that impairs tumourigenesis in patient-derived BTSC xenografts and syngeneic models

We next set out to investigate the functional relevance of these findings to GB in vivo. Since insufficient drug exposure in the brain is one of the major hurdles in treatment of brain tumours, we thus asked whether mubritinib could cross the blood-brain barrier (BBB). To address this question, we designed pharmacokinetic experiments to assess the bioavailability of mubritinib in the blood and the brain. After a single intraperitoneal injection of mubritinib, the animals were sacrificed at different time points, and plasma and brain samples were collected and subjected to mubritinib quantification by liquid chromatography/tandem mass spectrometry (LC/MS/MS) (Fig. 4A). Strikingly, we found that mubritinib accumulates in the brain at a brain/blood ratio of 2- to 3.4-fold after 12 h to 36 h, respectively (Fig. 4B). It reached a Cmax of 22 ng/mL/mg of matrix after 4 h in the plasma and 43 ng/mL/mg of matrix after 24 h in the brain (Fig. 4B). Importantly, after 36 h, mubritinib was still detectable in both the plasma and brain (Fig. 4B). These data highlight that mubritinib bypasses the BBB, accumulates in the brain and is cleared slowly from the brain.

Having established that mubritinib penetrates through the BBB, we therefore, investigated whether mubritinib could suppress brain tumours. First, we intracranially implanted murine-derived BTSCs, mGB2, into the brains of immunocompetent C57BL/6N mice (Fig. 4C). The animals were randomized to receive either mubritinib (6 mg/kg) or vehicle control intraperitoneally (Fig. 4C). Tumour formation was monitored by bioluminescence imaging of the luciferase signal using an in vivo real-time optical imaging system (Biospace) (Fig. 4D,E). Luciferase imaging revealed a marked decrease in brain tumour growth in mubritinib-treated mice compared to vehicle-treated mice (Fig. 4D,E). At ~40 days following surgery, the mice receiving the vehicle control formed malignant brain tumours and were at endpoints as assessed by major weight loss and neurological signs. Mubritinib delayed mGB2 tumourigenesis and significantly extended the lifespan of animals (Fig. 4F).

To confirm the effect of mubritinib on GB tumour growth and survival, we performed parallel experiments with two patient-derived BTSC lines (BTSC147 and BTSC73), as described for mGB2. One hundred days after intracranial implantation of BTSC147 in RAGγ2C$^{-/-}$ mice, vehicle-treated mice developed malignant brain tumours, while mice in the mubritinib group exhibited a significant delay in tumour formation, with a ~3-fold decrease in luciferase activity (Fig. 4G,H). Importantly, mubritinib significantly extended the lifespan of animals bearing BTSC147-tumours (Fig. 4I). Using another patient-derived BTSC model

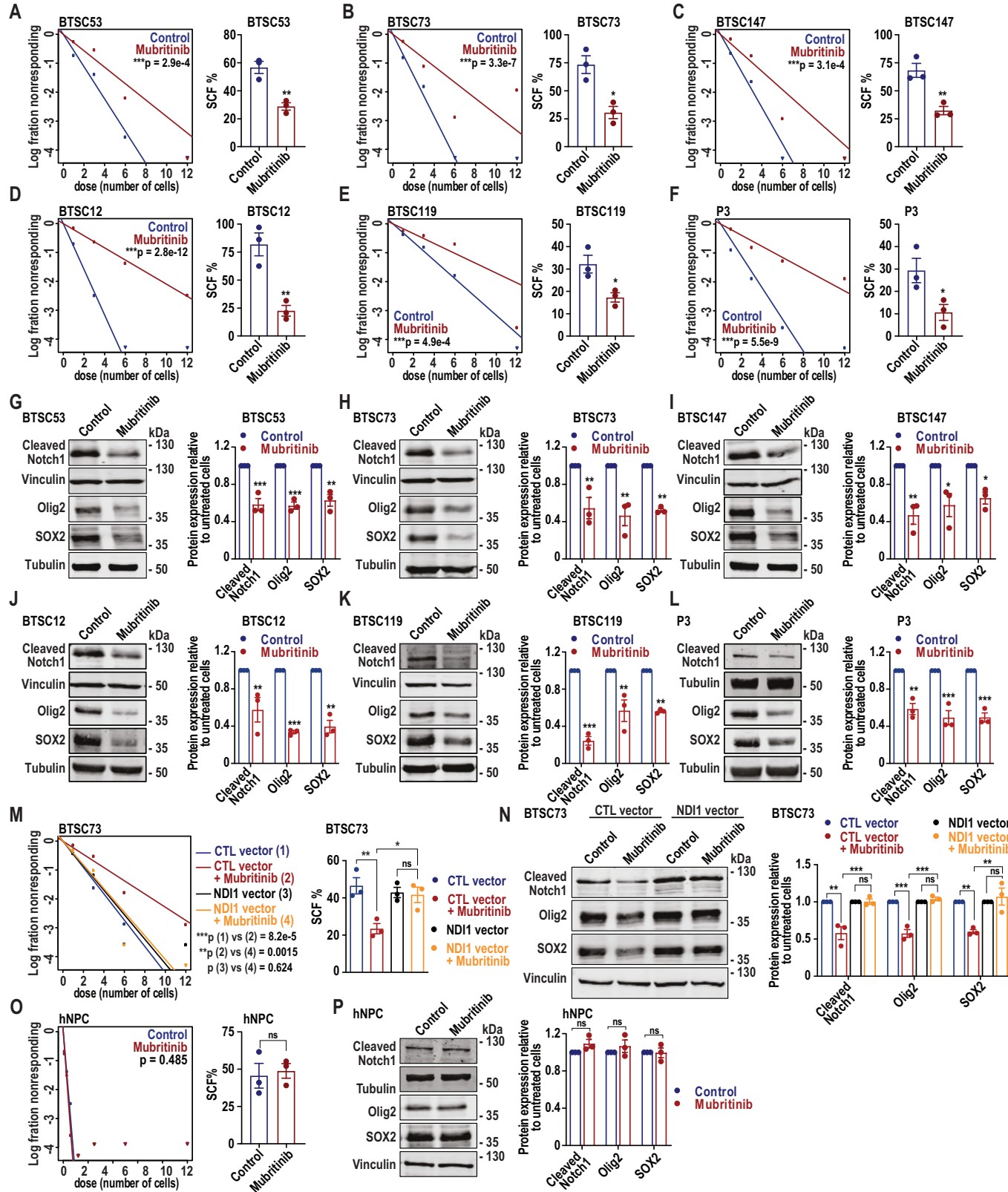

◀  **Figure 3.   Mubritinib impairs the stemness of BTSCs without affecting the non-oncogenic normal hNPCs.**

(A–F) BTSC53 (A), BTSC73 (B), BTSC147 (C), BTSC12 (D), BTSC119 (E) and P3 (F) were subjected to extreme limiting dilution analysis (ELDA) to estimate the stem cell frequencies (SCF), 21 days following treatment with 500 nM mubritinib or vehicle control. Data are presented as the means ± SEM, $n = 3$ independent biological experiments. Chi-square test for ELDA plots and unpaired two-tailed t test for SCF were used. $**p_{BTSC53} = 0.0052$, $*p_{BTSC73} = 0.0112$, $**p_{BTSC147} = 0.0078$, $**p_{BTSC12} = 0.0062$, $*p_{BTSC119} = 0.0287$, $*p_{P3} = 0.0454$. (G–L) BTSC53 (G), BTSC73 (H), BTSC147 (I), BTSC12 (J), BTSC119 (K) and P3 (L) were treated for 4 days with 500 nM mubritinib or vehicle control and subjected to immunoblotting using the antibodies indicated on the blots. Vinculin and tubulin were used as loading controls. Densitometric quantifications of cleaved Notch1, Olig2 and SOX2 protein levels normalized to their corresponding loading controls are presented. Data are presented as the means ± SEM, $n = 3$ independent biological experiments. One-way ANOVA followed by Dunnett's test vs vehicle control. BTSC53 (G): $***p_{Cleaved\ Notch1} = 0.0008$, $***p_{Olig2} = 0.0007$, $**p_{SOX2} = 0.0016$. BTSC73 (H): $**p_{Cleaved\ Notch1} = 0.0093$, $**p_{Olig2} = 0.0036$, $**p_{SOX2} = 0.0071$. BTSC147 (I): $**p_{Cleaved\ Notch1} = 0.0054$, $*p_{Olig2} = 0.0188$, $*p_{SOX2} = 0.0478$. BTSC12 (J): $**p_{Cleaved\ Notch1} = 0.0097$, $***p_{Olig2} = 0.0006$, $**p_{SOX2} = 0.0011$. BTSC119 (K): $***p_{Cleaved\ Notch1} = 7.4e{-}5$, $**p_{Olig2} = 0.0033$, $**p_{SOX2} = 0.0030$. P3 (L): $**p_{Cleaved\ Notch1} = 0.0017$, $***p_{Olig2} = 0.0005$, $***p_{SOX2} = 0.0005$. (M) BTSC73 transduced with the control (CTL) or NDI1 vector were treated with vehicle control or 500 nM mubritinib and subjected to ELDA to estimate the SCF, 21 days following treatment. Data are presented as the means ± SEM, $n = 3$ independent biological experiments. Chi-square test for ELDA plots and one-way ANOVA followed by Tukey's test for SCF were used. $**p_{CTL\ vs\ mubritinib} = 0.0082$, $*p_{mubritinib\ vs\ NDI1\ +\ mubritinib} = 0.0294$, $p_{NDI1\ vs\ NDI1\ +\ mubritinib} = 0.9946$. (N) BTSC73 transduced with the CTL or NDI1 vector were treated with vehicle control or 500 nM mubritinib for 4 days and subjected to immunoblotting using the antibodies indicated on the blots. Vinculin was used as loading control. Densitometric quantifications of cleaved Notch1, Olig2 and SOX2 protein levels normalized to their corresponding loading controls are presented. Data are presented as the means ± SEM, $n = 3$ independent biological experiments. One-way ANOVA followed by Tukey's test. $**p_{Cleaved\ Notch1\ (CTL\ vs\ mubritinib)} = 0.00105$, $***p_{Cleaved\ Notch1\ (mubritinib\ vs\ NDI1\ +\ mubritinib)} = 0.001$, $p_{Cleaved\ Notch1\ (NDI1\ vs\ NDI1\ +\ mubritinib)} = 0.9999$, $***p_{Olig2\ (CTL\ vs\ mubritinib)} = 2.8e{-}5$, $***p_{Olig2\ (Mubritinib\ vs\ NDI1\ +\ mubritinib)} = 1.4e{-}5$, $p_{Olig2\ (NDI1\ vs\ NDI1\ +\ mubritinib)} = 0.7603$, $**p_{SOX2\ (CTL\ vs\ mubritinib)} = 0.006$, $**p_{SOX2\ (mubritinib\ vs\ NDI1\ +\ mubritinib)} = 0.0022$, $p_{SOX2\ (NDI1\ vs\ NDI1\ +\ mubritinib)} = 0.8297$. (O) Human neural progenitor cells (hNPCs) were subjected to ELDA to estimate the SCF, 21 days following treatment with 500 nM mubritinib or vehicle control. Data are presented as the means ± SEM, $n = 3$ independent biological experiments. Chi-square test for ELDA plots and unpaired two-tailed t test for SCF were used. $p = 0.7555$. (P) hNPCs were treated with vehicle control or 500 nM mubritinib for 4 days and subjected to immunoblotting using the antibodies indicated on the blots. Vinculin and tubulin were used as loading controls. Densitometric quantifications of cleaved Notch1, Olig2 and SOX2 protein levels normalized to their corresponding loading controls are presented. Data are presented as the means ± SEM, $n = 3$ independent biological experiments. One-way ANOVA followed by Dunnett's test vs vehicle control. $p_{Cleaved\ Notch1} = 0.4252$, $p_{Olig2} = 0.6578$, $p_{SOX2} = 0.9996$. Source data are available online for this figure.

(BTSC73), we validated the ability of mubritinib to increase the lifespan of the animals (Fig. 4J). Having established that mubritinib impairs proliferation, stemness and self-renewal of BTSCs in vitro, we next asked whether it affects proliferation and stemness in GB tumours in vivo. To address this question, we orthotopically inoculated BTSC73 into RAGγ2C$^{-/-}$ mice. After 7 days, the mice were randomized into two treatment arms: vehicle control or mubritinib (6 mg/kg). Thirty days after surgery, all the mice were sacrificed, and the brains were sectioned and stained for the proliferation marker, Ki67, and the stemness markers Olig2 and SOX2. We observed a significant reduction in the number of Ki67, Olig2- and SOX2-positive cells in the mubritinib-treated mice (Fig. 4K–M), indicating that mubritinib alters proliferation and stemness in GB tumours in vivo.

Next, we set out to determine if the observed impact of mubritinib on GB tumours is due to OXPHOS inhibition. We therefore, conducted in vivo rescue experiments using patient-derived xenograft model. Vector control or NDI1 expressing-BTSC73 were orthotopically implanted into animals. Then, mice received either mubritinib treatment of vehicle control. As expected, KM survival plots revealed that mubritinib expands the lifespan in the vector control bearing group. Interestingly, ectopic expression of NDI1 rescues mubritinib effects, reversing the lifespan extension (Fig. 4N). Collectively, these data demonstrate that mubritinib crosses the BBB, effectively delays GB growth, decreases GB stemness and confers a survival advantage via targeting mitochondrial respiration.

## Combination of mubritinib treatment with IR therapy suppresses GB tumours and improves survival of GB-bearing mice

Recent proteomic analysis of GB tumours revealed an enrichment of OXPHOS-associated proteins in recurrent compared to primary GB tumours, suggesting that OXPHOS could be involved in GB treatment resistance (Tatari et al, 2022). We thus asked whether BTSCs upregulate mitochondrial respiration in response to IR, and whether inhibition of OXPHOS by mubritinib could improve the GB response to IR. To address these questions, we first exposed multiple patient-derived BTSCs to either 2 Gy or 4 Gy of IR, followed by OCR measurement via Resipher. We observed an increase in the OCR in irradiated BTSCs compared to control BTSCs (Fig. EV4A,B). Next, we set out to determine if mubritinib treatment impacts the BTSC response to IR. To begin with, we treated two BTSC lines (#53 and #73) with mubritinib and exposed them to 2 Gy or 4 Gy of IR. Interestingly, ELDA revealed a significant lower self-renewal capacity and SCF in BTSCs receiving the combined treatment of mubritinib and IR compared to either IR or mubritinib (Fig. 5A,B). Furthermore, combinational treatment of mubritinib and IR significantly decreased the number of live cells compared to either IR or mubritinib alone, as demonstrated by cell viability assays (Fig. 5C,D).

Having established that mubritinib improved the response of BTSCs to IR in vitro, we aimed to investigate the functional relevance of these findings to GB in vivo. We thus transplanted luciferase-expressing BTSC73 into the brains of immunodeficient mice. Mice were randomized to receive one of four treatment arms: vehicle control, mubritinib (6 mg/kg), irradiation (2 cycles of 2 Gy) or a combination of mubritinib and IR. Mubritinib administration was started 5 days before the first cycle of IR (Fig. 5E). Luciferase imaging at 21 days following surgery revealed a delay in tumour formation in the group receiving either mubritinib or IR monotherapy, with a ~5-fold decrease in luciferase activity compared to that in the control group. Interestingly, a much greater reduction (~14-fold) in luciferase activity was observed in mice receiving a combinational therapy of mubritinib and IR (Fig. 5F,G). The tumour volume was estimated by H&E staining 30 days after surgery. As observed by luciferase imaging, while IR alone and mubritinib alone decreased tumour volume compared to that of the control, the combination of IR and mubritinib resulted

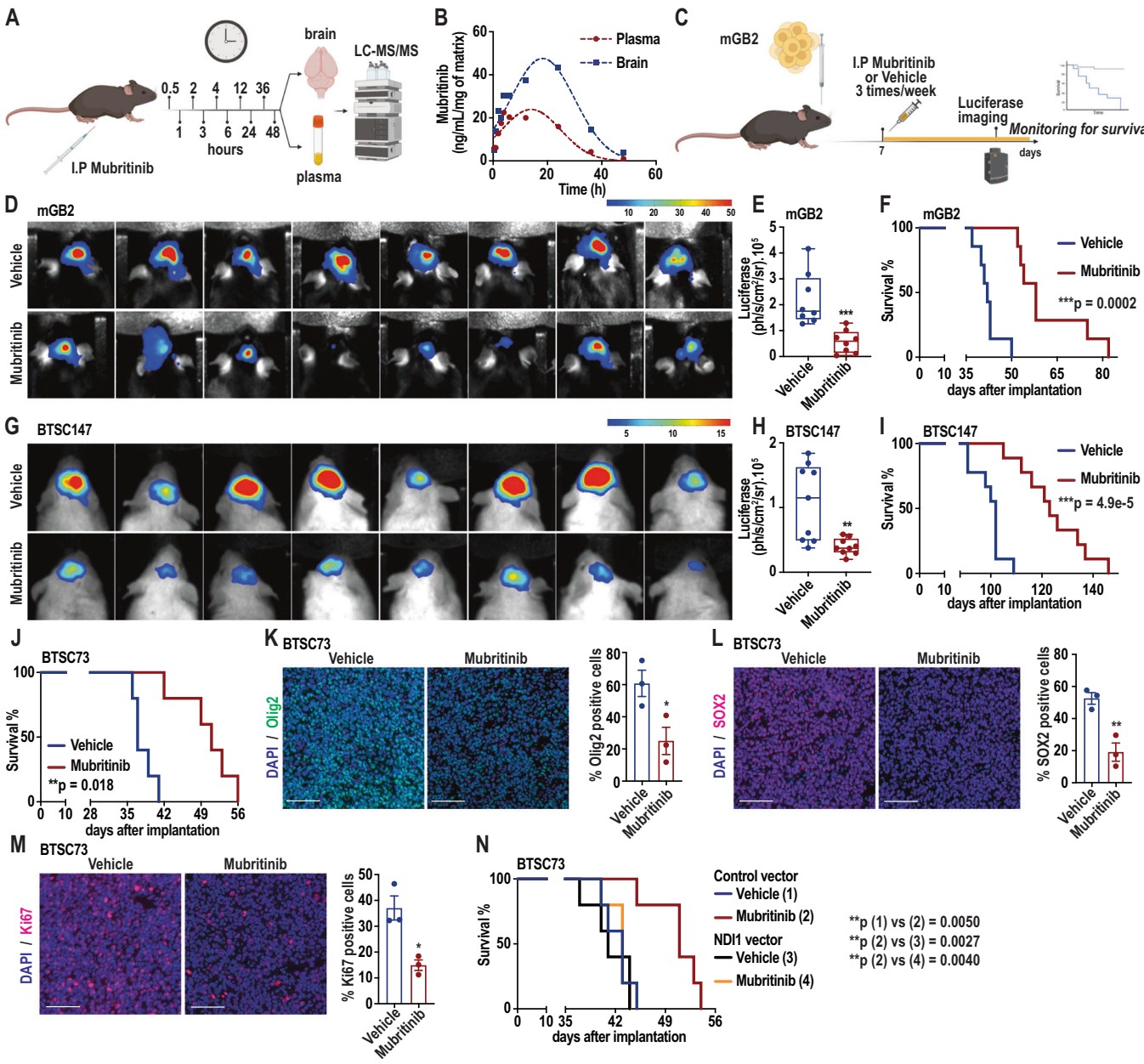

**Figure 4. Mubritinib, a brain penetrant drug, delays BTSC tumourigenesis.**

(A, B) Schematic diagram of mubritinib pharmacokinetic analysis is shown (A). C57BL/6N mice were treated by I.P with a single dose of mubritinib (6 mg/kg). The brain and plasma were collected at different time points (0.5 h to 48 h) after injection. Mubritinib was quantified by LC–MS/MS (B). (C) Schematic diagram of the experimental procedure in which luciferase-expressing mGB2 cells were intracranially injected into C57BL/6N mice. One week after implantation, the mice were randomized into 2 groups: vehicle control or mubritinib (6 mg/kg). Mice were treated 3 times per week (Monday, Wednesday and Friday). Luciferase imaging was used to assess tumour progression. (D, E) Bioluminescence images (D) and quantification of luciferase activity (E) 6 weeks after implantation are presented. Data are presented as box plots showing 25th and 75th percentiles (box), median (centre line), minima and maxima (whiskers), $n = 8$ mice. Unpaired two-tailed t test. ***$p = 0.001$. (F) Kaplan–Meier (KM) survival plots were generated to evaluate the lifespan of the mice in each group, and the mice were collected at the end stage (log-rank test, $n = 7$ mice). (G–I) Luciferase-expressing BTSC147 were intracranially injected into RAGγ2C$^{-/-}$ mice. Bioluminescence images (G) and quantification of luciferase activity (H) 9 weeks after implantation are presented. Data are presented as box plots showing 25th and 75th percentiles (box), median (centre line), minima and maxima (whiskers), $n = 9$ mice. Unpaired two-tailed t test. **$p = 0.0037$. KM survival plot was generated to assess animal lifespan (I). Log-rank test, $n = 9$ mice, ***$p = 4.9e-5$. (J) BTSC73 were intracranially injected into RAGγ2C$^{-/-}$ mice, and a KM survival plot was generated to compare the lifespans of vehicle- and mubritinib-treated mice (log-rank test, $n = 5$ mice). (K–M) Representative immunofluorescence images and quantification of Olig2 (green) (K), SOX2 (red) (L) and Ki67 (red) (M) positive cells in BTSC73 intracranial xenografts in mice treated with mubritinib or vehicle are shown. Nuclei were stained with DAPI (blue). The number of Olig2-, SOX2- and Ki67-positive cells was quantified with Fiji software. Scale bar = 100 μm. Data are presented as the means ± SEM, $n = 3$ mice. Unpaired two-tailed t test. *$p_{olig2} = 0.0386$, **$p_{SOX2} = 0.0075$, *$p_{Ki67} = 0.0122$. (N) Control or NDI1 vector expressing BTSC73 were intracranially injected into RAGγ2C$^{-/-}$ mice and the mice were administered either vehicle control or mubritinib (6 mg/kg). KM survival plot was generated to assess animal lifespan (log-rank test, $n = 5$ mice). Source data are available online for this figure.

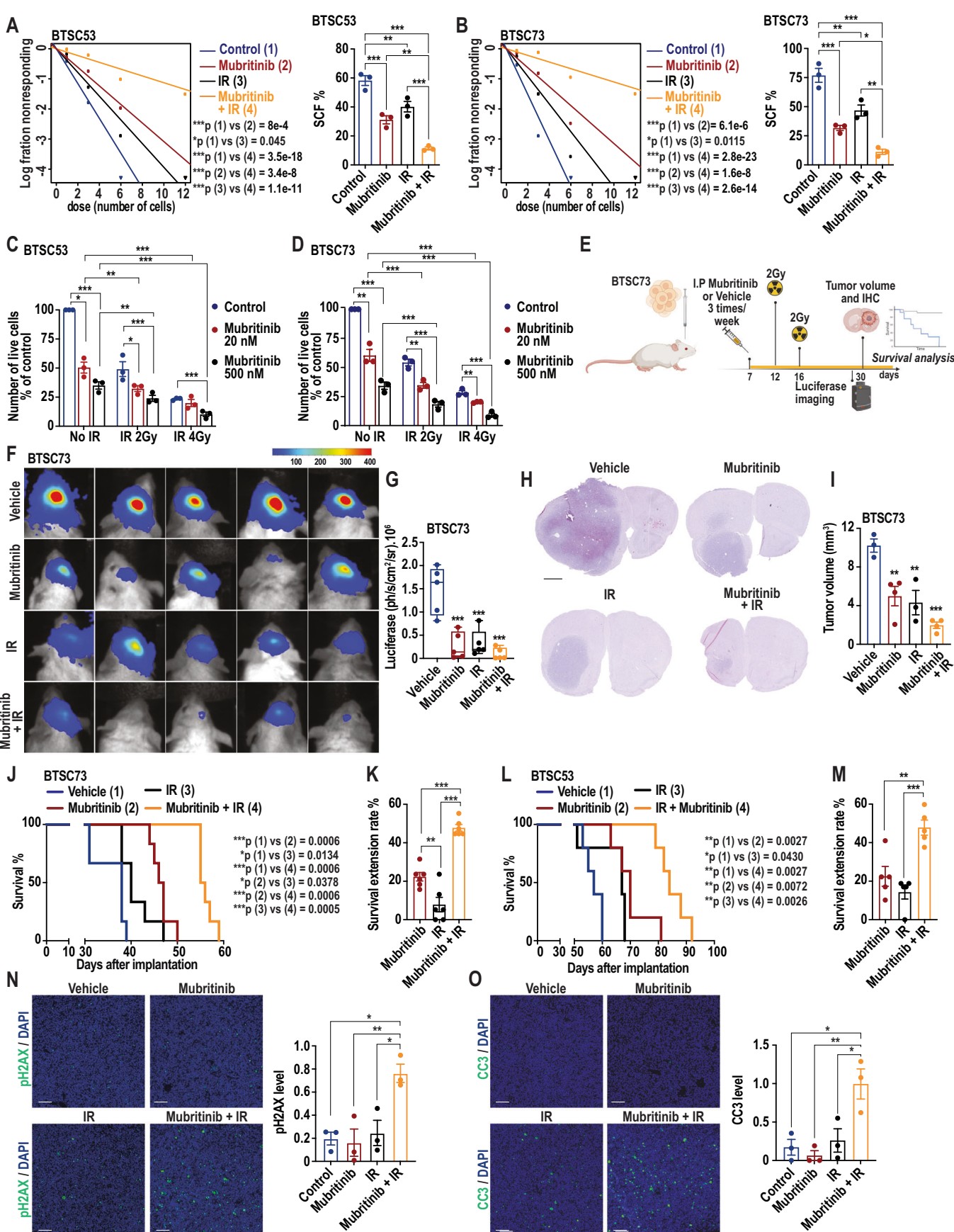

**Figure 5. Mubritinib sensitizes BTSCs and GB tumours to IR.**

(A, B) BTSC53 (A) and BTSC73 (B) were treated with 500 nM mubritinib, irradiated with 2 Gy, and subjected to ELDA to estimate the SCF, 21 days following treatment. Data are presented as the means ± SEM, $n = 3$ independent biological experiments. Chi-square test for ELDA plots and one-way ANOVA followed by Tukey's test for SCF were used. BTSC53 (A): ***$p_{control\ vs\ mubritinib} = 0.0008$, **$p_{control\ vs\ IR} = 0.0098$, ***$p_{control\ vs\ IR\ +\ mubritinib} = 1.5e-5$, ***$p_{IR\ vs\ IR\ +\ mubritinib} = 0.0005$, **$p_{mubritinib\ vs\ IR\ +\ mubritinib} = 0.0058$. BTSC73 (B): ***$p_{control\ vs\ mubritinib} = 0.0002$, **$p_{control\ vs\ IR} = 0.0036$, ***$p_{control\ vs\ IR\ +\ mubritinib} = 1.6e-5$, **$p_{IR\ vs\ IR\ +\ mubritinib} = 0.0013$, *$p_{mubritinib\ vs\ IR\ +\ mubritinib} = 0.0318$. (C, D) BTSC53 (C) and BTSC73 (D) were treated with 20 nM or 500 nM mubritinib, irradiated with 2 Gy or 4 Gy and subjected to live cell counting by PI staining followed by flow cytometry after 7 days of treatment. Data are presented as the means ± SEM, $n = 3$ independent biological experiments. Two-way ANOVA followed by Tukey's test. BTSC53 (C): *$p_{control\ vs\ mubritinib\ 20} = 0.0117$, ***$p_{control\ vs\ mubritinib\ 500} = 0.0001$, *$p_{2Gy\ vs\ 2Gy\ +\ mubritinib\ 20} = 0.0117$, ***$p_{2Gy\ vs\ 2Gy\ +\ mubritinib\ 500} = 0.0001$, ***$p_{4Gy\ vs\ 4Gy\ +\ mubritinib\ 500} = 0.0001$, **$p_{mubritinib\ 20\ vs\ 2Gy\ +\ mubritinib\ 20} = 0.0029$, **$p_{mubritinib\ 500\ vs\ 2Gy\ +\ mubritinib\ 500} = 0.0029$, ***$p_{mubritinib\ 20\ vs\ 4Gy\ +\ mubritinib\ 20} = 3.2e-6$, ***$p_{mubritinib\ 500\ vs\ 4Gy\ +\ mubritinib\ 500} = 3.2e-6$. BTSC73 (D): **$p_{control\ vs\ mubritinib\ 20} = 0.0019$, ***$p_{control\ vs\ mubritinib\ 500} = 4.1e-7$, **$p_{2Gy\ vs\ 2Gy\ +\ mubritinib\ 20} = 0.0019$, ***$p_{2Gy\ vs\ 2Gy\ +\ mubritinib\ 500} = 4.1e-7$, **$p_{4Gy\ vs\ 4Gy\ +\ mubritinib\ 20} = 0.0019$, ***$p_{4Gy\ vs\ 4Gy\ +\ mubritinib\ 500} = 4.1e-7$, ***$p_{mubritinib\ 20\ vs\ 2Gy\ +\ mubritinib\ 20} = 6.8e-5$, ***$p_{mubritinib\ 500\ vs\ 2Gy\ +\ mubritinib\ 500} = 6.8e-5$, ***$p_{mubritinib\ 20\ vs\ 4Gy\ +\ mubritinib\ 20} = 4e-8$, ***$p_{mubritinib\ 500\ vs\ 4Gy\ +\ mubritinib\ 500} = 4e-8$. (E) Schematic diagram of the experimental procedure in which luciferase-expressing BTSC73 cells were intracranially injected into RAG$\gamma$2C$^{-/-}$ mice. One week after implantation, the mice were randomized into 2 groups: vehicle control or mubritinib (6 mg/kg). Mice were treated 3 times per week (Monday, Wednesday and Friday). For IR groups, on day 12 and 16, mice received 2 Gy. (F, G) Bioluminescence images (F) and quantification of luciferase activity (G) 3 weeks after implantation are presented. Data are presented as box plots showing 25th and 75th percentiles (box), median (centre line), minima and maxima (whiskers), $n = 5$ mice. One-way ANOVA followed by Tukey's test. ***$p_{vehicle\ vs\ mubritinib} = 0.0002$, ***$p_{vehicle\ vs\ IR} = 0.0003$, ***$p_{vehicle\ vs\ mubritinib\ +\ IR} = 4.4e-5$. (H, I) H&E stainings were performed on day 30 (H). Scale bar = 1 mm. Tumour volume was estimated based on H&E staining (I). Data are presented as the means ± SEM, $n \geq 3$ mice. One-way ANOVA followed by Tukey's test. **$p_{vehicle\ vs\ mubritinib} = 0.0065$, **$p_{vehicle\ vs\ IR} = 0.0056$, ***$p_{vehicle\ vs\ mubritinib\ +\ IR} = 0.0003$. (J) KM survival plot was graphed to evaluate mice lifespan in each group, mice were collected at end stage (log-rank test, $n = 6$ mice). (K) Survival extensions of mice bearing BTSC73-derived tumours treated with mubritinib, IR, or mubritinib + IR relative to those treated with the vehicle control were calculated. Data are presented as the means ± SEM, $n = 6$ mice. One-way ANOVA followed by Tukey's test. **$p_{mubritinib\ vs\ IR} = 0.0047$, ***$p_{mubritinib\ vs\ mubritinib\ +\ IR} = 2.1e-5$, ***$p_{IR\ vs\ mubritinib\ +\ IR} = 8e-8$. (L, M) BTSC53 were intracranially injected into RAG$\gamma$2C$^{-/-}$ mice. KM was plotted to evaluate mice lifespan in each group (log-rank test, $n = 5$ mice) (L), and survival extensions were calculated (M). Data are presented as the means ± SEM, $n = 5$ mice. One-way ANOVA followed by Tukey's test. **$p_{mubritinib\ vs\ mubritinib\ +\ IR} = 0.0032$, ***$p_{IR\ vs\ mubritinib\ +\ IR} = 0.0003$. (N, O) Representative immunofluorescence images and quantification of phosphorylated histone H2AX (pH2AX) (green) (N) and cleaved caspase 3 (CC3) (green) (O) in BTSC73 intracranial xenografts in mice treated with vehicle control, mubritinib, IR, or mubritinib + IR are shown. Nuclei were stained with DAPI (blue). Quantification was performed with Fiji software. Scale bar = 100 μm. Data are presented as the mean ± SEM, $n = 3$ mice. One-way ANOVA followed by Tukey's test. *$p_{pH2AX\ (vehicle\ vs\ mubritinib\ +\ IR)} = 0.0118$, **$p_{pH2AX\ (mubritinib\ vs\ mubritinib\ +\ IR)} = 0.0084$, *$p_{pH2AX\ (IR\ vs\ mubritinib\ +\ IR)} = 0.0193$, *$p_{CC3\ (vehicle\ vs\ mubritinib\ +\ IR)} = 0.0125$, **$p_{CC3\ (mubritinib\ vs\ mubritinib\ +\ IR)} = 0.0062$, *$p_{CC3\ (IR\ vs\ mubritinib\ +\ IR)} = 0.0229$. Source data are available online for this figure.

in smaller tumours (Fig. 5H,I). Importantly, KM survival plots showed that animals receiving the combination of IR and mubritinib had a survival advantage over animals receiving either IR or mubritinib monotherapy in the BTSC73 patient-derived xenograft model (Fig. 5J). Indeed, a ~50% extension of the survival rate was observed in group receiving IR and mubritinib compared to the control group (Fig. 5K).

To further validate the advantage of combining IR with mubritinib, we implanted a second patient-derived BTSC line (#53) into the brains of RAG$\gamma$2C$^{-/-}$ mice. Similar to the BTSC73 xenograft model, combined treatment with mubritinib and IR significantly extended the lifespan of the animals bearing BTSC53 xenografts (Fig. 5L,M).

Since the brain concentration peak of mubritinib was observed after 24 h of IP administration, in another set of experiments, we evaluated the advantage of combining IR and mubritinib treatment on GB tumours, administering mubritinib 24 h prior to the start of IR (Fig. EV4C). Interestingly, similar to the first treatment scenario, we found an advantage of combining IR and mubritinib in reducing GB progression and expanding the animal lifespan (Fig. EV4D–G). These data suggest that initiating mubritinib treatment, just 24 h prior to IR, is sufficient to sensitize GB tumours to IR.

Upregulation of DNA repair capacity is one of the mechanisms by which BTSCs resist IR-induced DNA damage (Bao et al, 2006). Interestingly, analysis of RNA-seq data revealed that mubritinib significantly downregulates DNA repair pathways (Fig. EV4H,I), raising the question of whether mubritinib potentiates IR-induced DNA damage. To investigate this, we conducted immunofluorescence staining for phosphorylated histone H2A (pH2AX), a DNA damage marker. We observed a significant increase in pH2AX levels in GB xenografts from animals receiving combined IR and

mubritinib treatment compared to those receiving either IR or mubritinib monotherapy (Fig. 5N). Since unrepaired DNA damage leads to apoptosis, we next assessed whether the combined IR and mubritinib treatment enhances apoptosis in GB tumours. Immunofluorescence staining of cleaved caspase-3 (CC3) on GB tumour xenograft sections showed significantly higher CC3 levels in tumours receiving combined IR and mubritinib compared to either treatment alone (Fig. 5O). These data suggest that mubritinib sensitizes GB tumours to IR by enhancing DNA damage and apoptosis. Overall, these results support the potential of combining mubritinib and IR as a promising therapeutic strategy for treating GB tumours.

## Mubritinib alleviates tumour hypoxia and enhances oxidative stress to sensitize GB tumours to IR

Tumour response to IR is highly dependent on oxygen, with hypoxic tumours often exhibiting greater resistance. This is primarily because oxygen is essential for the production of reactive oxygen species (ROS), which are a key mechanism of IR-induced DNA damage (Rakotomalala et al, 2021). Therefore, in addition to downregulating DNA repair pathways, a potential mechanism by which mubritinib may sensitize GB tumours to IR is by alleviating hypoxia. The hypothesis is that, by inhibiting OXPHOS, mubritinib reduces oxygen consumption within the tumour, allowing more oxygen to diffuse into hypoxic regions. This, in turn, enhances ROS production and improves the tumour's response to IR (Benej et al, 2018; Zannella et al, 2013).

To explore this hypothesis, we first tested whether mubritinib affects the oxygen levels in BTSC tumourspheres in vitro. We employed hypoxia probe labelling to assess oxygen availability.

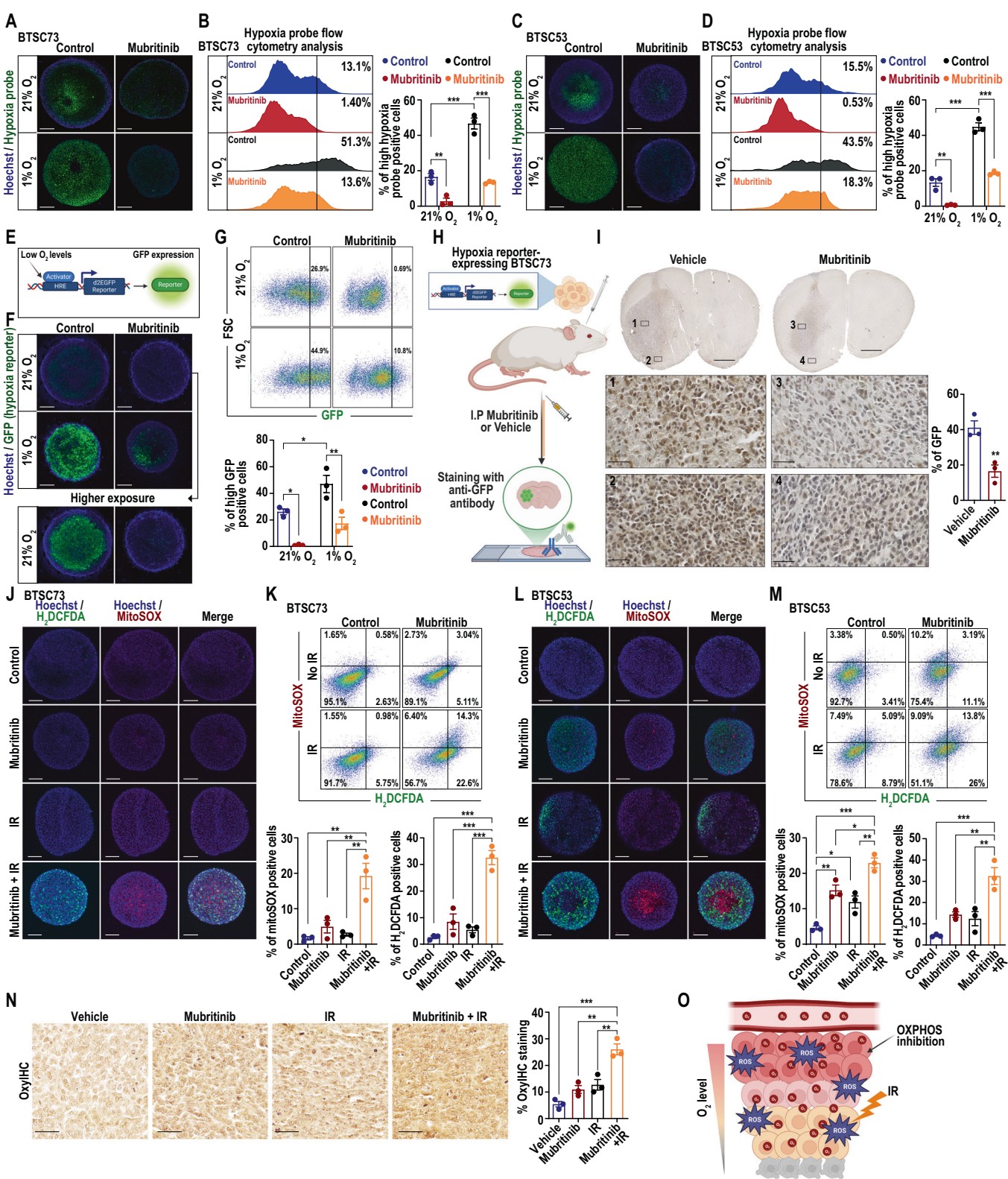

**Figure 6. Mubritinib alleviates tumour hypoxia and enhances oxidative stress to sensitize GB tumours to IR.**

(A–D) BTSC73 (**A, B**) and BTSC53 (**C, D**) tumourspheres were treated with 500 nM mubritinib under normoxic (21% O₂) or hypoxic (1% O₂) conditions and subjected to labelling with hypoxia probe followed by live confocal imaging (**A, C**) or flow cytometric analysis (**B, D**). Representative confocal images of hypoxia probe (green) are shown. Nuclei were stained by Hoechst. Scale bar = 100 μm (**A, C**). The percentages of high hypoxia probe-positive cells quantified by flow cytometry are shown (**B, D**). Data are presented as the mean ± SEM, $n = 3$ independent biological experiments. One-way ANOVA followed by Tukey's test. BTSC73 (**B**): ***$p_{\text{control 21\% O}_2 \text{ vs control 1\% O}_2} = 2.5e-5$, **$p_{\text{control 21\% O}_2 \text{ vs mubritinib 21\% O}_2} = 0.0058$, ***$p_{\text{control 1\% O}_2 \text{ vs mubritinib 1\% O}_2} = 1.2e-5$. BTSC53 (**D**): ***$p_{\text{control 21\% O}_2 \text{ vs control 1\% O}_2} = 3.1e-6$, **$p_{\text{control 21\% O}_2 \text{ vs mubritinib 21\% O}_2} = 0.0022$, ***$p_{\text{control 1\% O}_2 \text{ vs mubritinib 1\% O}_2} = 1.2e-5$. (**E**) A schematic diagram illustrating the GFP hypoxia reporter is shown. (**F, G**) GFP hypoxia reporter-expressing BTSC73 tumourspheres were treated with 500 nM mubritinib under normoxic or hypoxic conditions and subjected to live confocal imaging (**F**) or flow cytometric analysis (**G**). Representative confocal images of GFP hypoxia reporter (green) are shown. Nuclei were stained by Hoechst. Scale bar = 100 μm (**F**). The percentages of high GFP hypoxia reporter-positive cells quantified by flow cytometry are shown (**G**). Data are presented as the mean ± SEM, $n = 3$ independent biological experiments. One-way ANOVA followed by Tukey's test. *$p_{\text{control 21\% O}_2 \text{ vs control 1\% O}_2} = 0.03$, *$p_{\text{control 21\% O}_2 \text{ vs mubritinib 21\% O}_2} = 0.0114$, **$p_{\text{control 1\% O}_2 \text{ vs mubritinib 1\% O}_2} = 0.0043$. (**H**) A schematic diagram of the in vivo intracranial tumour assay involving the GFP hypoxia reporter-expressing BTSC73 is presented. GFP hypoxia reporter-expressing BTSC73 were intracranially implanted into RAGγ2C⁻/⁻, followed by treatment of mice with vehicle control or mubritinib (6 mg/kg) 3 times per week (Monday, Wednesday and Friday). (**I**) Representative images and quantifications of GFP staining of sections from GFP hypoxia reporter-expressing BTSC73 xenografts are shown. Scale bar = 1 mm, scale bar inset = 50 μm. The GFP-positive cells were quantified with Fiji software. Data are presented as the mean ± SEM, $n = 3$ mice. Unpaired two-tailed t test. **$p = 0.0088$. (**J–M**) BTSC73 (**J, K**) or BTSC53 (**l, M**) tumourspheres were treated with 500 nM mubritinib, IR 2 Gy, or combination of both for 3 days and subjected to labelling with H₂DCFDA and MitoSOX followed by live confocal imaging (**J and l**) or flow cytometric analysis (**K, M**). Representative confocal images of the H₂DCFDA (green) and MitoSOX (red) probes are shown. Nuclei were stained by Hoechst. Scale bar = 100 μm (**J, l**). The percentages of H₂DCFDA or MitoSOX positive cells are shown (**K, M**). Data are presented as the mean ± SEM, $n = 3$ independent biological experiments. One-way ANOVA followed by Tukey's test. BTSC73 (**K**): **$p_{\text{mitoSOX (control vs mubritinib + IR)}} = 0.0013$, **$p_{\text{mitoSOX (mubritinib vs mubritinib + IR)}} = 0.0048$, **$p_{\text{mitoSOX (IR vs mubritinib + IR)}} = 0.0019$, ***$p_{\text{H}_2\text{DCFDA (control vs mubritinib+IR)}} = 3.3e-5$, ***$p_{\text{H}_2\text{DCFDA (mubritinib vs mubritinib+IR)}} = 0.0002$, ***$p_{\text{H}_2\text{DCFDA (IR vs mubritinib+IR)}} = 6.8e-5$. BTSC53 (**M**): **$p_{\text{mitoSOX (control vs mubritinib)}} = 0.0026$, *$p_{\text{mitoSOX (control vs IR)}} = 0.0221$, **$p_{\text{mitoSOX (control vs mubritinib + IR)}} = 5.6e-5$, *$p_{\text{mitoSOX (mubritinib vs mubritinib + IR)}} = 0.0158$, **$p_{\text{mitoSOX (IR vs mubritinib + IR)}} = 0.0019$, ***$p_{\text{H}_2\text{DCFDA (control vs mubritinib+IR)}} = 0.0003$, **$p_{\text{H}_2\text{DCFDA (mubritinib vs mubritinib+IR)}} = 0.0059$, **$p_{\text{H}_2\text{DCFDA (IR vs mubritinib+IR)}} = 0.0032$. (**N**) Representative images of GB tumour sections from BTSC73 xenografts from mice treated with mubritinib, IR, or combination of mubritinib and IR and subjected to staining using OxyIHC oxidative stress detection kit are shown. Scale bar = 50 μm. OxyIHC was quantified with Fiji software. Data are presented as the mean ± SEM, $n = 3$ mice. One-way ANOVA followed by Tukey's test. ***$p_{\text{control vs mubritinib + IR}} = 0.0001$, **$p_{\text{mubritinib vs mubritinib + IR}} = 0.0011$, **$p_{\text{IR vs mubritinib + IR}} = 0.0025$. (**O**) Schematic illustration showing that mubritinib alleviates hypoxia and enhances ROS generation in response to IR. Source data are available online for this figure.

Fluorescent imaging via live confocal microscopy revealed poor oxygen diffusion, in the hypoxic core of the BTSC tumourspheres cultured in normoxia, as indicated by high green fluorescence (Figs. 6A,C and EV5A). To better model tissue hypoxia in vitro, we set up BTSC tumourspheres cultures under hypoxic conditions (1% O₂), and found that low O₂ levels caused a significant increase in hypoxia probe fluorescent labelling, as demonstrated by confocal imaging and flow cytometry analysis in multiple BTSCs (Figs. 6A–D and EV5B–E). Interestingly, in both normoxic and hypoxic settings, treatment with mubritinib strongly reduced the fluorescent labelling (Figs. 6A–D and EV5A–E). This suggests an increase in the oxygen available for diffusion within the tumoursphere upon mubritinib treatment.

To further confirm these results, we genetically engineered BTSCs expressing an oxygen-dependent GFP reporter by transducing them with a vector containing d2EGFP driven by hypoxia responsive elements (HREs) (Fig. 6E). Mubritinib treatment reduced GFP, indicating increased oxygen availability as demonstrated by live confocal imaging and flow cytometry analysis (Fig. 6F,G). Then, we leveraged this hypoxia GFP reporter to determine whether mubritinib mitigates hypoxia and increases oxygen availability in GB tissues in an in vivo setting. Therefore, we conducted intracranial tumour assays by implanting GFP-reporter-expressing BTSC73 cells into the brains of mice. The mice were then administered either a vehicle control or mubritinib (Fig. 6H). At the end of the assay, brain sections from both groups were subjected to GFP staining as an indicator of hypoxia. Interestingly, we found that mubritinib reduced hypoxia in GB tumours in vivo (Fig. 6I), suggesting an increase in oxygen availability in mubritinib-treated tumours.

Next, we sought to determine whether the increase in oxygen availability results in enhanced ROS generation in response to IR. Therefore, we assessed ROS production following combination treatment with mubritinib and IR, using H₂DCFDA and MitoSOX Red probes to detect cellular ROS and mitochondrial superoxides,

respectively. We found that the combination of mubritinib and IR led to a significant increase in ROS levels compared to either treatment alone, in multiple patient-derived BTSCs tested in vitro (Figs. 6J–M and EV5F,G). To assess ROS generation in GB tumours in vivo, we subjected brain tumour xenografts from animals treated with mubritinib alone, IR alone, or the combination of both to staining using the OxyIHC oxidative stress detection kit. Our results revealed significantly higher ROS levels in GB tumours receiving combined IR and mubritinib treatment compared to either mubritinib or IR monotherapy (Fig. 6N).

To further demonstrate that mubritinib alleviates hypoxia and sensitizes BTSCs to IR, we evaluated whether mubritinib enhances the response to IR under hypoxic conditions. We performed a series of experiments in which BTSCs were irradiated and cultured under low oxygen conditions (1% O₂). We found that under hypoxic conditions, mubritinib was capable to enhance ROS generation (Fig. EV5H,I) and to sensitizing BTSCs to IR, as demonstrated by the ELDA (Fig. EV5J). Altogether, these data suggest that OXPHOS inhibition by mubritinib results in greater oxygen availability and enhanced ROS generation to improve the response of BTSCs and GB tumours to IR (Fig. 6O).

## Combining mubritinib with TMZ chemotherapy effectively suppresses GB growth

To further determine the clinical relevance of mubritinib treatment in GB tumours, we explored the potential of a combined treatment involving mubritinib and TMZ. To begin with, we conducted in vitro assays in which BTSCs were treated with 20 or 500 nM mubritinib, 5 μM TMZ, or a combination of both. Seven days following treatment, cells were subjected to cell viability assays. Treatment with either 500 nM mubritinib alone or TMZ alone reduced the number of live cells to ~35% or ~50% of the control in the two BTSC lines tested (#53 and #73). Interestingly, a more efficient decrease in cell viability to ~15% was observed when

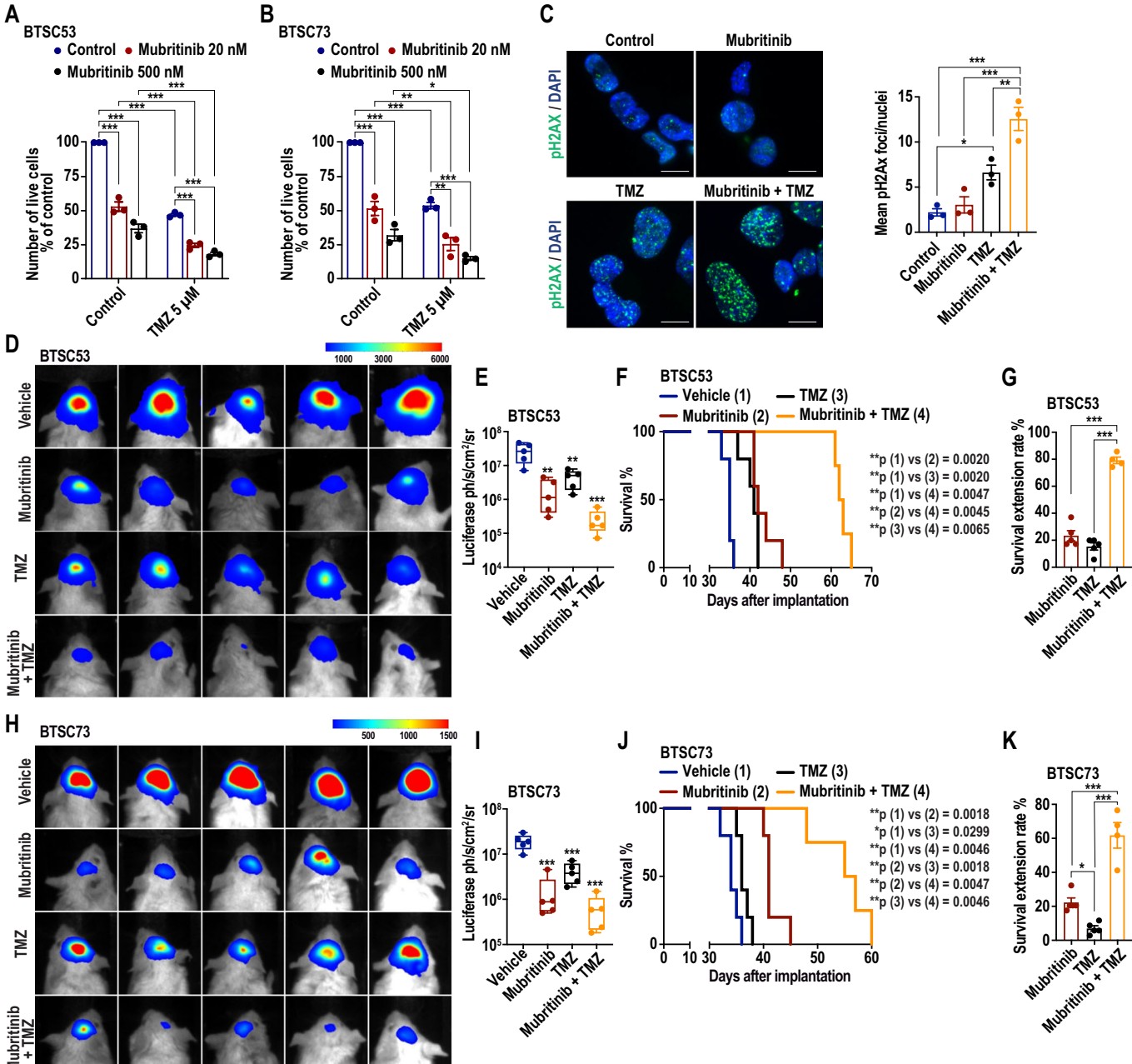

BTSCs were treated with a combination of TMZ and mubritinib (Fig. 7A,B). Having demonstrated that mubritinib downregulates DNA repair pathways in the RNA-seq data, and since the induction of DNA lesions is a key mechanism by which TMZ exerts its effects, we assessed DNA damage following treatment of BTSCs with TMZ, mubritinib, or a combination of both. Immunofluorescence staining and quantification revealed a significant increase in the number of pH2AX foci per nucleus following combined treatment with TMZ and mubritinib, compared to either treatment alone (Fig. 7C).

Having shown that mubritinib enhances the effect of TMZ in vitro, we next aimed to determine the combinatorial benefit of mubritinib and TMZ in vivo using patient-derived xenograft models. First, we transplanted BTSC53 transduced with luciferase into the brains of immunocompromised mice. Mice bearing BTSC53 were treated with one of the four arms: vehicle control, mubritinib monotherapy (6 mg/kg), TMZ monotherapy (1 cycle of 5 days of 1 mg/kg), or a combination of mubritinib and TMZ. Compared to vehicle control, a significant reduction in tumour size, as assessed by luciferase imaging, was observed in the mubritinib-treated group or TMZ-treated group (Fig. 7D,E). Importantly, tumour growth was most efficiently inhibited and markedly decreased in the group receiving the combination of mubritinib and TMZ (Fig. 7D,E). Furthermore, the survival of animals receiving the combination of TMZ and mubritinib was significantly extended compared to the animals that received mubritinib or

**Figure 7. Mubritinib sensitizes BTSCs and GB tumours to TMZ.**

(A, B) BTSC53 (A) and BTSC73 (B) were treated with 20 nM or 500 nM mubritinib, 5 µM TMZ or a combination of mubritinib and TMZ and subjected to live cell counting by PI staining followed by flow cytometry after 7 days of treatment. Data are presented as the mean ± SEM, $n = 3$ independent biological experiments. One-way ANOVA followed by Tukey's test. BTSC53 (A): ***$p_{control \; vs \; mubritinib \; 20} = 2.5e-8$, ***$p_{control \; vs \; mubritinib \; 500} = 9.3e-10$, ***$p_{control \; vs \; TMZ} = 6.73e-9$, ***$p_{TMZ \; vs \; TMZ \; + \; mubritinib \; 20} = 7.6e-5$, ***$p_{TMZ \; vs \; TMZ \; + \; mubritinib \; 500} = 5.9e-6$, ***$p_{mubritinib \; 20 \; vs \; TMZ \; + \; mubritinib \; 20} = 6.3e-6$, ***$p_{mubritinib \; 500 \; vs \; TMZ \; + \; mubritinib \; 500} = 0.0004$. BTSC73 (B): ***$p_{control \; vs \; mubritinib \; 20} = 5.3e-6$, ***$p_{control \; vs \; mubritinib \; 500} = 1.2e-7$, ***$p_{control \; vs \; TMZ} = 8.6e-6$, *$p_{TMZ \; vs \; TMZ \; + \; mubritinib \; 20} = 0.0010$, ***$p_{TMZ \; vs \; TMZ \; + \; mubritinib \; 500} = 5.1e-5$, **$p_{mubritinib \; 20 \; vs \; TMZ \; + \; mubritinib \; 20} = 0.0020$; *$p_{mubritinib \; 500 \; vs \; TMZ \; + \; mubritinib \; 500} = 0.0417$. (C) Representative immunofluorescence images and quantification of phosphorylated histone H2A (pH2AX) (green) in BTSC73 treated with vehicle control, mubritinib, TMZ, or mubritinib + TMZ are shown. Nuclei were stained with DAPI (blue). Quantification was performed with Fiji software. Scale bar = 10 µm. Data are presented as the mean ± SEM, $n = 3$ independent biological experiments. One-way ANOVA followed by Tukey's test. *$p_{control \; vs \; TMZ} = 0.0358$, ***$p_{control \; vs \; mubritinib \; + \; TMZ} = 0.0002$, ***$p_{mubritinib \; vs \; mubritinib \; + \; TMZ} = 0.0003$, **$p_{TMZ \; vs \; mubritinib \; + \; TMZ} = 0.0073$. (D–K) Luciferase-expressing BTSC53 (D–G) or BTSC73 (H–K) were intracranially injected into RAGγ2C$^{-/-}$ mice. The mice were randomized into 4 groups: vehicle control, mubritinib (6 mg/kg 3 times per week), TMZ (1 cycle of 5 days of 1 mg/kg), or combination of mubritinib (6 mg/kg 3 times per week) and TMZ (1 cycle of 5 days of 1 mg/kg). The first mubritinib injection was administered on day 11 after surgery, 24 h prior the start of TMZ treatment. Bioluminescence images (D, H) and quantification of luciferase activity (E, I) 4 weeks after implantation are presented. Data are presented as box plots showing 25th and 75th percentiles (box), median (centre line), minima and maxima (whiskers), $n = 5$ mice. One-way ANOVA followed by Tukey's test. BTSC53 (E): **$p_{vehicle \; vs \; mubritinib} = 0.0011$, **$p_{vehicle \; vs \; TMZ} = 0.0029$, ***$p_{vehicle \; vs \; mubritinib \; + \; TMZ} = 0.0006$. BTSC73 (I): ***$p_{vehicle \; vs \; mubritinib} = 1.7e-5$, ***$p_{vehicle \; vs \; TMZ} = 0.0001$, ***$p_{vehicle \; vs \; mubritinib \; + \; TMZ} = 1e-5$. KM was plotted to evaluate mice lifespan in each group (F and J) (log-rank test $n \geq 4$ mice), and survival extensions were calculated (G, K). Data are presented as the mean ± SEM, $n \geq 4$ mice. One-way ANOVA followed by Tukey's test. BTSC53 (G): ***$p_{mubritinib \; vs \; mubritinib \; + \; TMZ} = 2.3e-7$, ***$p_{TMZ \; vs \; mubritinib \; + \; TMZ} = 6e-8$. BTSC73 (K): *$p_{mubritinib \; vs \; TMZ} = 0.0461$, ***$p_{mubritinib \; vs \; mubritinib \; + \; TMZ} = 9.5e-5$, ***$p_{TMZ \; vs \; mubritinib \; + \; TMZ} = 4e-6$. Source data are available online for this figure.

TMZ alone (Fig. 7F,G). This advantage in efficiently reducing tumour growth and improving overall survival was also observed in the group receiving the combination of mubritinib and TMZ using another patient-derived BTSC (#73) xenograft model (Fig. 7H–K). Our data suggest that mubritinib ameliorates the curative effect of TMZ and underscores the promising therapeutic potential of effectively targeting GB by combining mubritinib and TMZ.

## Mubritinib does not induce damage to healthy cells and is well tolerated

Having established that mubritinib impairs BTSC growth and tumourigenesis both in vitro and in vivo, this raised the question of mubritinib's impact on non-oncogenic normal cells. To address this question, we employed hNPCs, human hepatocytes (HepaSH), and murine post-mitotic primary mouse neurons (Fig. 8A). First, we compared the effect of mubritinib to that of TMZ, the primary chemotherapeutic agent for GB in the clinic, on the viability of non-oncogenic healthy hNPCs. Strikingly, phase-contrast images of hNPC neurospheres showed that mubritinib, at doses effective against BTSCs, had no significant effect on hNPCs, whereas TMZ was severely toxic to hNPCs (Fig. 8B).

To further demonstrate this, we treated hNPCs with either TMZ or mubritinib and subjected them to an Annexin V/PI flow cytometry assay 7 days post-treatment. We found that, at concentrations where mubritinib inhibited the proliferation of BTSCs, there was no significant decrease in cell viability and no induction of cell death or early apoptosis in mubritinib-treated hNPCs (Fig. 8C,D). The lack of toxicity of mubritinib on hNPCs was further confirmed using calcein-AM/DAPI double staining, marking live and dead cells, respectively (Appendix Fig. S2A–C). In contrast, TMZ profoundly reduced the number of viable hNPCs by 90% and induced a 36% increase in the number of dead cells (Fig. 8C,D). These data suggest that, mubritinib impairs selectively BTSCs without affecting the normal hNPCs. This could be explained by the fact that hNPCs have distinct metabolic profile compared to BTSCs, where they rely on glycolysis rather than OXPHOS (Beckervordersandforth et al, 2017; Khacho et al, 2016; Zheng et al, 2016). In support, we found that, similar to mubritinib, the OXPHOS inhibitor, rotenone, had no effect on the number of live hNPCs, while 2-Deoxy-D-glucose, a competitive glycolysis

inhibitor, significantly reduced the number of live hNPCs (Appendix Fig. S2D).

Next, we evaluated the toxicity of mubritinib on hepatocytes and neurons. Phase-contrast examinations showed no morphological alterations in mubritinib-treated human hepatocytes, HepaSH, compared to those treated with the hepatotoxic drug bosentan (Fig. 8E). In addition, cell viability assays revealed no significant decrease in the viability of HepaSH (Fig. 8F) or mouse neurons (Fig. 8G) following mubritinib treatment.

Toxicity in normal brain cells has been reported with many chemotherapeutic agents. For instance, the anti-GB drugs carmustine has been shown to impact normal brain cells and increase cell death in the lateral subventricular zone (SVZ), where NPCs reside (Dietrich et al, 2006). We, therefore, assessed whether mubritinib treatment alters healthy NPCs in vivo. We found that mubritinib had no apparent impact on the percentage of SOX2-positive NPCs in the SVZ (Fig. 8H,I). Furthermore, we performed immunostaining for CC3 in the brains of mubritinib-treated mice and found no increase in apoptosis compared to that in vehicle-treated mice (Appendix Fig. S2E). Taken together, the in vitro and in vivo data show that mubritinib impairs BTSCs without causing significant damage to normal cells.

Next, we assessed the tolerability of mubritinib in vivo. We evaluated the systemic toxicity of mubritinib after 3 months (3 times/week) of repeated mubritinib treatment (6 mg/kg) by performing blood biochemical analysis and behavioural studies on mubritinib-treated mice (Fig. 8J). Body weight assessments indicated that mubritinib was well-tolerated at the tested dose without causing any weight loss (Fig. 8K). Through blood biochemical analysis, we examined liver function through measuring the aspartate amino transferase (AST), alanine amino transferase (ALT), alkaline phosphatase (ALP), total protein, albumin and glucose levels (Fig. 8L); kidney function through measuring the creatinine, uric acid and urea levels (Fig. 8M); and pancreas function by measuring amylase level (Fig. 8N). No significant changes in the levels of these markers were observed in the mubritinib-treated mice compared to the vehicle control mice. Additionally, no difference in LDH levels was observed, indicating no induction of any tissue damage following mubritinib treatment (Fig. 8O). Lactic acidosis is a frequently reported side effect and a major concern that has led to the termination of many clinical trials

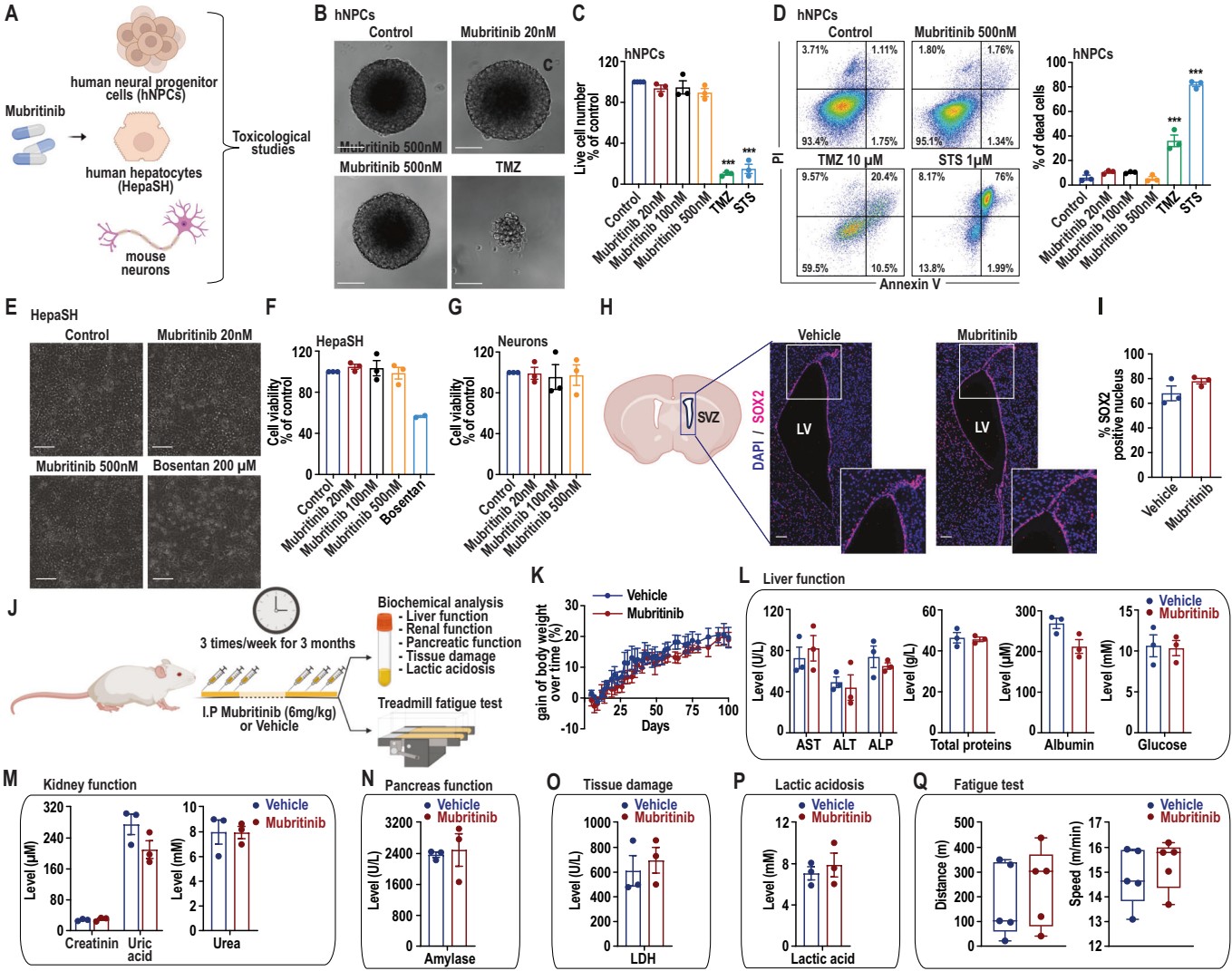

**Figure 8. Mubritinib does not impact normal non-oncogenic cells and has a safe profile in vivo.**

(A) Schematic diagram of the in vitro analysis of the effects of mubritinib on normal cells. (B) Representative phase-contrast images of human neural progenitor cells (hNPCs) treated with 20 nM mubritinib, 500 nM mubritinib, or 10 µM TMZ for 7 days. Scale bars = 100 µm. (C, D) hNPCs were treated with increasing concentration of mubritinib (0–500 nM) or 10 µM TMZ for 7 days followed by live cell counting (C) and Annexin V/PI double staining (D). Representative flow cytometry plots and the percentages of dead cells (PI positive) are shown (D). 1 µM staurosporine (STS) is used as positive control. Data are presented as the mean ± SEM, $n = 3$ independent biological experiments. One-way ANOVA followed by Dunnett's test vs vehicle control. Live cell number (C): $p_{\text{control vs mubritinib 20}} = 0.6277$, $p_{\text{control vs mubritinib 100}} = 0.7447$, $p_{\text{control vs mubritinib 500}} = 0.1854$, ***$p_{\text{control vs TMZ}} = 5e{-}10$, ***$p_{\text{control vs STS}} = 1e{-}9$. Dead cells (D): $p_{\text{control vs mubritinib 20}} = 0.5555$, $p_{\text{control vs mubritinib 100}} = 0.6174$, $p_{\text{control vs mubritinib 500}} = 0.9989$, ***$p_{\text{control vs TMZ}} = 4.5e{-}6$, ***$p_{\text{control vs STS}} = 1e{-}10$. (E, F) HepaSH human hepatocytes were treated with mubritinib or the hepatotoxic drug bosentan, as a positive control. Representative phase-contrast images are shown (E). Scale bars = 100 µm. Cytotoxicity was measured by MTT (F). Data are presented as the mean ± SEM, $n \geq 2$ independent biological experiments. One-way ANOVA followed by Dunnett's test vs vehicle control. $p_{\text{control vs mubritinib 20}} = 0.8229$, $p_{\text{control vs mubritinib 100}} = 0.9219$, $p_{\text{control vs mubritinib 500}} = 0.9933$. (G) Cell viability of mouse cortical neurons was measured by MTT after mubritinib treatment. Data are presented as the mean ± SEM, $n = 3$ independent biological experiments. One-way ANOVA followed by Dunnett's test vs vehicle control. $p_{\text{control vs mubritinib 20}} = 0.9997$, $p_{\text{control vs mubritinib 100}} = 0.9660$, $p_{\text{control vs mubritinib 500}} = 0.9925$. (H, I) Representative immunofluorescence images of SOX2 in the SVZ of mouse brain treated with vehicle control or mubritinib (H). Scale bars = 100 µm. The number of SOX2-positive cells was quantified with Fiji software (I). Data are presented as the mean ± SEM, $n = 3$ mice. Unpaired two-tailed t test. $p = 0.2026$. (J) A schematic diagram of the in vivo toxicological studies is presented. Biochemical analysis and treadmill fatigue tests were performed on RAGγ2C$^{-/-}$ mice treated for 3 months with vehicle control or mubritinib (6 mg/kg) 3 times per week (Monday, Wednesday and Friday). (K) The body weights of the mice are presented. Data are presented as the mean ± SEM, $n = 5$ mice. Two-way ANOVA followed by Sidak's test. $p = 0.4354$. (L–P) Biochemical analyses to evaluate liver (L), kidney (M), pancreatic function (N), tissue damage (O) and lactic acidosis (P) were performed. Data are presented as the mean ± SEM, $n = 3$ mice. Unpaired two-tailed t test. $p_{\text{AST}} = 0.5737$, $p_{\text{ALT}} = 0.7168$, $p_{\text{ALP}} = 0.4601$, $p_{\text{total proteins}} = 0.7949$, $p_{\text{albumin}} = 0.0531$, $p_{\text{glucose}} = 0.8712$, $p_{\text{creatin}} = 0.5971$, $p_{\text{uric acid}} = 0.1361$, $p_{\text{urea}} = 0.9663$, $p_{\text{amylase}} = 0.7867$, $p_{\text{LDH}} = 0.6259$, $p_{\text{lactic acid}} = 0.5728$. (Q) A treadmill fatigue test was performed. The speed and distances ran by the mubritinib-treated and vehicle control groups were measured. Data are presented as box plots showing 25th and 75th percentiles (box), median (centre line), minima and maxima (whiskers), $n = 5$ mice. Unpaired two-tailed t test. $p_{\text{distance}} = 0.5490$, $p_{\text{speed}} = 0.4946$. Source data are available online for this figure.

of OXPHOS inhibitors (Yap et al, 2023). Thus, we measured the level of lactic acid and found that mubritinib did not induce increase in the levels of lactic acid (Fig. 8P). Next, we performed a non-voluntary preclinical test of fatigue-like behaviour, the treadmill fatigue test, on mice following three months of repeated mubritinib treatment. After two training sessions, the fatigue test was performed. Interestingly, we did not detect any fatigue-like behaviour in the mubritinib-treated mice compared to the control mice. In fact, mubritinib-treated mice ran at comparable speeds and distances to those in the control group (Fig. 8Q). In summary, these data indicate that at effective anti-GB doses, mubritinib has a well-tolerated and safe profile and could represent a promising therapeutic agent for treating GB tumours.

## Discussion

GB is the most aggressive and deadly brain tumour with a dismal prognosis despite multimodal therapies (Van Meir et al, 2010). GB tumours exhibit remarkable cellular heterogeneity, with a population of BTSCs at the apex of the differentiation hierarchy (Lathia et al, 2015). BTSCs display potent tumour-initiating capacity and promote malignant behaviours associated with disease progression and relapse (Bao et al, 2006; Guryanova et al, 2011; Lathia et al, 2015). Therefore, targeting BTSCs is crucial for improving GB treatment and overcoming therapeutic resistance. In this study, we establish for the first time that mubritinib is a promising drug for targeting cancer stem cells in GB. We employed murine-derived and multiple patient-derived BTSCs with different genetic backgrounds and found that mubritinib suppressed BTSC stemness pathways, self-renewal and proliferation. Mechanistically, by employing mitochondrial respiration assays and rescue experiments involving ectopic expression of the *Saccharomyces cerevisiae NDI1* gene, we provide compelling evidence that mubritinib acts on complex I of the ETC to impair mitochondrial respiration, BTSC growth and GB progression. Through pharmacokinetic studies, we established that mubritinib is a brain-penetrant drug. Using patient-derived BTSC orthotopic xenografts and syngeneic murine GB preclinical models, we established that mubritinib delays GB tumourigenesis, sensitizes GB response to radiotherapy and chemotherapy and expands the lifespan of GB tumour-bearing animals. Importantly, biochemical, toxicological and behavioural studies have shown that mubritinib is safe. Taken together, these data reveal that mubritinib is a potent drug that can impair BTSCs and suppress GB tumours as a stand-alone therapy or in combination with IR or TMZ.

Mubritinib, also known as TAK165, is a small molecule that was developed in the early 2000s and was initially reported as a specific ERBB2 inhibitor (Nagasawa et al, 2006). However, later studies demonstrated that gastric cancer cells that exhibit dramatic reductions in the expression of various tyrosine kinase receptors (TKRs), including HER2/ERBB2, are resistant to TKR inhibitors but sensitive to mubritinib. This finding suggested that mubritinib acts via alternative mechanisms that do not involve the inhibition of HER2/ERBB2 (Grygielewicz et al, 2016). In a more recent study, Baccelli and colleagues demonstrated that mubritinib selectively inhibits the growth of a subset of AML cells that rely on OXPHOS by inhibiting complex I of the ETC (Baccelli et al, 2019).

Prior to our study, the impact of mubritinib on BTSCs and GB tumour progression had not been explored. Here, we provide robust evidence that mubritinib alters BTSC growth, stemness, and GB progression by impairing complex I activity and subsequently inhibiting OXPHOS.

We tested mubritinib on a panel of patient-derived BTSCs with diverse genetic backgrounds and transcriptional profiles, demonstrating its effectiveness across all tested cell lines. Notably, we observed variable sensitivity among BTSCs to mubritinib, which correlated with their respiratory capacity. To further understand this variability, we analysed the transcriptional profiles of BTSCs to assess whether sensitivity to mubritinib was linked to specific transcriptional subtypes. Using the Wang classification of GB subtypes (Wang et al, 2017), we found no correlation between sensitivity to mubritinib and either the mesenchymal or classical subtypes. However, the proneural subtype exhibited a negative correlation with mubritinib sensitivity, suggesting that BTSCs with proneural signature may be less susceptible to OXPHOS inhibition. By analysing the correlation between sensitivity to mubritinib and the recently identified metabolic-related transcriptional GB subtypes (Garofano et al, 2021), we observed a potential positive correlation with the mitochondrial GB subtype (Garofano et al, 2021). This subtype, which relies exclusively on OXPHOS for energy production, demonstrates marked vulnerability to OXPHOS inhibitors (Garofano et al, 2021). In line with this, our findings suggest that mubritinib may preferentially target the mitochondrial subtype, although further studies with a larger panel of samples are required to conclusively establish this relationship.

An important feature of BTSCs is their metabolic plasticity, which allows them to shift between metabolic pathways to survive nutrient deprivation or metabolic stress (Dando et al, 2015; Peiris-Pages et al, 2016; Snyder et al, 2018; Vlashi et al, 2011). However, certain genetic mutations impair the metabolic fitness of BTSCs and increase their sensitivity to OXPHOS inhibitors. For instance, homozygous deletion of the key glycolytic enzyme enolase 1, which occurs in 3.3% of GB patients, reduces the capacity for compensatory glycolysis and increases sensitivity to the complex I inhibitor IACS-010759 (Molina et al, 2018). Oncogenic gene fusion of *FGFR3-TACC3* has been described in 3% of human GB cases (Singh et al, 2012) and has been shown to activate OXPHOS and mitochondrial biogenesis and confer sensitivity to inhibitors of oxidative metabolism (Frattini et al, 2018). This suggests that mubritinib could be more effective on GBs harbouring these genetic alterations.

Radioresistance has emerged as one of the major obstacles in GB therapy. BTSCs are central to conferring resistance to therapy and have been shown to be more radio-resistant than their non-stem cell counterparts in GB (Bao et al, 2006; Diehn et al, 2009; Lathia et al, 2015). We previously showed that increased cellular respiration and spare respiratory capacity in BTSCs confer resistance to IR therapy (Sharanek et al, 2020). In line with these studies, we showed that BTSCs upregulate OXPHOS in response to IR. This suggests that one intrinsic mechanism by which BTSCs may resist the IR insults is by enhancing their mitochondrial respiration, and thus targeting OXPHOS could sensitize BTSCs to IR.

Tumour hypoxia presents a significant challenge in radiology, as it reduces the effectiveness of irradiation. This is mainly because irradiation efficacy is highly dependent on the availability of oxygen

for ROS generation and stabilization of DNA-damage. Therefore, strategies to alleviate hypoxia are at the forefront to improve radiotherapy. One emerging strategy to overcome radioresistance in hypoxic tumour regions is to target mitochondrial respiration. OXPHOS inhibition results in a reduction in oxygen consumption in well-oxygenated tumours and, therefore, increases oxygen availability for diffusion into hypoxic regions to improve the radiation response (Benej et al, 2018; Zannella et al, 2013). In this study, we provided convincing in vitro and in vivo experimental evidence that mubritinib increases GB oxygenation, resulting, when combined with IR, in increased ROS generation and enhanced DNA damage, therefore improving the IR efficacy.

Repurposing metformin as a cancer treatment is already being tested in a range of clinical trials for a variety of cancers, and its combination with TMZ and IR is being investigated as a therapeutic avenue for GB patients (Mazurek et al, 2020). However, it did not confer a clinical benefit in patients with recurrent or refractory GB (Yoon et al, 2023). Furthermore, the combination of metformin with chemoradiation did not improve the overall or progression-free survival of patients with non-small cell lung cancer (Skinner et al, 2021). Given the low potency of metformin as an OXPHOS inhibitor, the outcome of these clinical trials is not surprising. Our findings show that mubritinib is a potent OXPHOS inhibitor in GB, accumulates significantly in the brain, and sensitizes GB tumours to IR and TMZ. Advancing these combinations into clinical trials could hold promise for improving GB therapy, if successful.

Drugs targeting mitochondrial metabolism are emerging as promising antitumour therapeutics in preclinical models. However, in the last decades, the clinical development of agents targeting metabolism has proven challenging. A few clinical trials on OXPHOS inhibitors, such as BAY87-2243, ASP4132 and IACS-010759, have been terminated due to dose-limiting toxicity and the narrow therapeutic window of these agents (Janku et al, 2022; Xu et al, 2020; Yap et al, 2023). Other clinical trials have failed due to the lack of anti-tumour effects (Alistar et al, 2017). In a very recent clinical trial, IACS-010759, was tested in patients with advanced solid tumours and AML (Yap et al, 2023). Patients who were administered the drug developed unacceptable toxicities, including elevated blood lactate and peripheral neuropathy, resulting in termination of the clinical trial (Yap et al, 2023). At lower doses, this drug was not effective. Importantly, Yap et al showed that IACS-010759 induces behavioural and physiological changes indicative of peripheral neuropathy in mouse models (Yap et al, 2023), suggesting that preclinical models could predict these drug-induced injuries. In light of the outcomes of these clinical trials, efforts should be directed towards thorough early evaluation of the safety of anticancer agents in preclinical context. Our biochemical assays showed that chronic mubritinib administration does not induce lactic acidosis or any significant changes. Furthermore, chronic mubritinib treatment did not induce behavioural changes or fatigue in the animals. Interestingly, the substantial accumulation of mubritinib in the brain suggested its efficacy at low doses and therefore minimized side effects. Therefore, our data with mubritinib breaks down the recent bleak view of targeting mitochondrial respiration as an anti-cancer strategy, and encourages to further thoroughly evaluate the pharmacokinetics, pharmacodynamics and safety profiles of this drug.

In summary, our study identified mubritinib as a brain-penetrant drug with a safe profile and robust anti-GB activity in preclinical models. Given that mubritinib has already undergone a phase I clinical trial in the context of ERBB2+ solid tumours, our work encourages and warrants future investigations for clinical translation and repurposing of mubritinib for better management of GB tumours in combination with current standard of care.

# Methods

**Reagents and tools table**

| Reagent/Resource | Reference or Source | Identifier or Catalog Number |
|---|---|---|
| **Experimental models** | | |
| RAGγ2C$^{-/-}$ (M. musculus) | Animal facility of Talence, University of Bordeaux, France | NA |
| C57BL/6N (M. musculus) | Charles River Laboratories | C57BL/6N |
| Patient-derived brain tumour stem cells: #12, #25, #50, #53, #73, #100, #119, #147, #198 | Provided by Dr. Samuel Weiss and Dr. Artee Luchman, University of Calgary, Canada | NA |
| Patient-derived brain tumour stem cells: P3 | Provided by Prof. Rolf Bjerkvig, University of Bergen, Norway | NA |
| Human HepaSH hepatocytes | Biopredic International, France | NA |
| mGB2 | Provided by Dr. Peter Angel, German Cancer Research Center, Germany | NA |
| ReNcell® CX Human Neural Progenitor Cell Line | Sigma-Aldrich | #SCC007 |
| **Plasmids** | | |
| pLenti6.3/V5 NDI1 | Provided by Dr. Joseph Marszalek, University of Texas, United States | NA |
| pLenti6.3/V5 | Provided by Dr. Joseph Marszalek, University of Texas, United States | NA |
| 5HRE/GFP | Addgene | #46926 |
| **Antibodies** | | |
| IR-Dye 680 goat anti-mouse IgG secondary antiboby | LI-COR Biosciences | #926-68020 |
| IR-Dye 800 donkey anti-mouse IgG secondary antiboby | LI-COR Biosciences | #926-32212 |
| IR-Dye 680 donkey anti-rabbit IgG secondary antiboby | LI-COR Biosciences | #926-68023 |
| IR-Dye 800 goat anti-rabbit IgG secondary antiboby | LI-COR Biosciences | #926-32211 |
| Notch1 (D1E11) XP® rabbit mAb | Cell Signaling Technology | #3608 |
| Olig2 (EPR2673) rabbit mAb | Abcam | #ab109186 |
| Sox2 (E-4) mouse mAb | Santa Cruz | #sc-365823 |
| AMPKα1 rabbit pAb | ABclonal | #A1229 |

| Reagent/Resource | Reference or Source | Identifier or Catalog Number |
|---|---|---|
| CDKN1B/p27KIP1 rabbit pAb | ABclonal | #A0290 |
| Phospho-AMPKα (Thr172) (40H9) rabbit mAb | Cell Signaling Technology | #2535 |
| Cyclin D1 (EPR2241) rabbit mAb | Abcam | #134175 |
| GFAP (GA5) mouse mAb | Cell Signaling Technology | #3670 |
| Phospho-RB1 (Ser807/811) rabbit pAb | Proteintech | #30376-1-AP |
| α-Tubulin mouse mAb | Sigma-Aldrich | #T5168 |
| Vinculin mouse mAb | Invitrogen | #MA511690 |
| Cleaved Caspase-3 (Asp175) rabbit mAb | Cell Signaling Technology | #9661 |
| Ki67 (SP6) rabbit mAb | Abcam | #ab16667 |
| GFP rabbit pAb | Torrey Pines | #TP401 |
| SOX2 mouse mAb | Proteintech | #66411-1 |
| Goat anti-rabbit IgG (H + L) Cross-Adsorbed Secondary Antibody, Alexa Fluor™ 488 | Invitrogen | #A11008 |
| Goat anti-mouse IgG (H + L) Cross-Adsorbed Secondary Antibody, Alexa Fluor™ 594 | Invitrogen | #A11005 |
| Goat anti-rabbit biotinylated antibody | Dako | #E0432 |
| **Oligonucleotides and other sequence-based reagents** | | |
| ON TARGET-plus SMART pool human NDUFS7 siRNA | Dharmacon | #L-031021-00-0005 |
| ON TARGET-plus non-targeting pool | Dharmacon | #D-001810-10-05 |
| qPCR primers | This paper | See Methods |
| **Chemicals, Enzymes and other reagents** | | |
| DMEM/F12 media | Gibco | #21331020 |
| Neurobasal media | Gibco | #21103049 |
| GlutaMAX | Fisher Scientific | #35050061 |
| HEPES | Sigma-Aldrich | #51558 |
| B-27 supplement minus vitamin A | Gibco | #12587010 |
| B-27 supplement | Gibco | #17504044 |
| Heparin | Sigma-Aldrich | #H3149 |
| EGF | Peprotech | #AF-100-15 |
| bFGF | Peprotech | #AF-100-18B |
| Penicillin–streptomycin | Gibco | #15140122 |
| HepaSH recovery medium | Biopredic International | #MIL130 |
| HepaSH seeding medium | Biopredic International | #MIL221 |
| HepaSH culture medium | Biopredic International | #MIL222 |
| Carboxyfluorescein diacetate succinimidyl ester | Invitrogen | #65-0850-85 |
| Neuron isolation kit | Miltenyi | #130-115-389 |
| Poly-D-lysine | Gibco | #A3890401 |

| Reagent/Resource | Reference or Source | Identifier or Catalog Number |
|---|---|---|
| Laminin | Sigma-Aldrich | #L2020 |
| Papain from Carica papaya | Sigma-Aldrich | #76216 |
| Accutase | Corning | #25-058-CI |
| Mubritinib | MedChemExpress | #HY-13501 |
| Temozolomide | TargetMol | #T1178 |
| Thiazolyl blue tetrazolium bromide | Sigma-Aldrich | #M5655 |
| 4-(trifluorométhoxy) phenylhydrazone carbonyle cyanide (FCCP) | Sigma-Aldrich | #C2920 |
| TACS annexin V-FITC apoptosis detection kit | R&D Systems | #4830-01-K |
| Calcein-AM | Invitrogen | #C1430 |
| Bradford | Sigma-Aldrich | #B6916 |
| Click-iT EdU cell proliferation kit | Invitrogen | #C10340 |
| FxCycle PI/RNase staining solution | Molecular Probes | #F10797 |
| 2′,7′-dichlorodihydrofluorescein diacetate ($H_2DCFDA$) | Invitrogen | #D399 |
| MitoSOX™ Mitochondrial Superoxide Indicators | Invitrogen | #M36008 |
| Hypoxia Green Reagent | Thermo Scientific | #H20035 |
| All-In-One 5X RT MasterMix | abm | #G492 |
| EurobioGreen qPCR Mix Hi-ROX | Eurobio Scientific | #GAEMMX02H0T |
| D-luciferin | AAT Bioquest | #AAT-12507 |
| ProLong Diamond Antifade Mountant | Molecular Probes | #P36961 |
| VECTASTAIN Elite ABC reagent | Vector Laboratories | #PK-6100 |
| OxyIHC oxidative stress detection kit protocol | Millipore | #S7450 |
| **Software** | | |
| FlowJo 10.8.1 | BD Biosciences | NA |
| Fiji 2.1.0/1.53o | ImageJ | NA |
| Image Studio Lite 5.2.5 | LI-COR | NA |
| R 4.4.1 | R | NA |
| M3 vision | Biospace Lab | NA |
| Prism 10.3.1 | GraphPad | NA |
| **Other** | | |
| Resipher real-time cell analyser | Lucid Scientific | NA |
| Oroboros Oxygraph-2K | Oroboros Instruments | NA |
| Microbeam Technology Irradiation System | XenX Xstrahl | NA |
| PhotonIMAGER in vivo imaging system | Biospace Lab | NA |

## Animals

All animal experiments were conducted in accordance with institutional guidelines and approved by the local ethics committee (agreement number: C335222). The C57BL/6N mice were purchased from Charles River Laboratories. The RAGγ2C$^{-/-}$ mice used in this study were born, housed, and treated in the animal facility of Talence (Animalerie Mutualisée de Talence, University of Bordeaux, France). The housing room temperature and relative humidity were adjusted to $22.0 \pm 2.0\,°C$ and $55.0 \pm 10.0\%$, respectively. The light/dark cycle was adjusted to 12 h lights-on and 12 h lights-off. Autoclaved water and irradiated food pellets were given ad libitum.

## Patient-derived and murine BTSC cultures

All patient-derived cell lines used were generated prior to our study (Cusulin et al, 2015; Kelly et al, 2009; Sakariassen et al, 2006). Human BTSCs 12, 25, 50, 53, 73, 75, 100, 119, 147 and 198 were generated from patient GB tumours by Dr. Samuel Weiss's team at the University of Calgary following written informed consent from patients and approval from the University of Calgary Ethics Review Board as previously described (Bahia et al, 2023; Cusulin et al, 2015; Kelly et al, 2009). The P3 line was derived from a GB tumour as previously described (Sakariassen et al, 2006). Cells were characterized for major mutations in GB including EGFRvIII, p53, PTEN, and IDH1 status (Appendix Table S1). The murine GB cell line mGB2 was kindly provided by Dr. Peter Angel (Costa et al, 2021). Prior to use, BTSCs were recovered from cryopreservation in 10% DMSO and cultured in low attachment flasks as spheres in BTSC complete media composed of DMEM/F12 media (Gibco, #21331020) supplemented with 5 mM HEPES (Sigma-Aldrich, #51558), 2 mM GlutaMAX (Gibco, #35050061), 1X B-27 supplement (Gibco, #12587010), 2 µg/ml heparin (Sigma-Aldrich, #H3149), 20 ng/ml EGF (Peprotech, #AF-100-15), 20 ng/ml bFGF (Peprotech, #AF-100-18B), and 100 U/ml penicillin‒streptomycin (Gibco, #15140122). All cell lines were tested negative for mycoplasma. All cells were kept in a humidified 5% $CO_2$ incubator at 37 °C.

## Differentiation of BTSCs

For BTSC differentiation, cells were cultured in BTSC differentiation media, composed of DMEM/F12 (Gibco, #21331020) supplemented with 5 mM HEPES (Sigma-Aldrich, #51558), 2 mM GlutaMAX (Gibco, #35050061), 100 U/ml penicillin‒streptomycin (Gibco, #15140122), and 10% foetal bovine serum (FBS) for 14 days. After differentiation, cells were assessed for stemness (SOX2 and Olig2) and differentiation (GFAP) marker expression by Western blotting. To compare sensitivity to mubritinib between BTSCs and their differentiated progeny, treatments were performed using complete BTSC media for 4 days, followed by live cell counting through PI cell viability flow cytometry assay.

## CFSE proliferation assay

BTSCs proliferation over time was evaluated by flow cytometry using carboxyfluorescein diacetate succinimidyl ester (CFSE) (Invitrogen, #65-0850-85). Briefly, BTSCs were washed twice with 1X PBS and incubated with 1 µM CFSE in 1X PBS at room temperature for 10 min, protected from light. Then, the CFSE solution was diluted five times with BTSC complete medium, and the cells were incubated on ice for 5 min. Cells were then washed three times with BTSC complete medium. Immediately after staining, a T0 sample was analysed by flow cytometry. For each experimental condition, $3 \times 10^4$ cells were plated and incubated for various time points to monitor cell proliferation. At each time point, CFSE fluorescence intensity was measured using an Accuri C6 flow cytometer. FlowJo software was used to analyse the data and generate population growth curve by dividing the CFSE mean fluorescence at T0 by the CFSE mean fluorescence at each time point.

## Human neural progenitor cell cultures

Human neural progenitor cell cultures (hNPCs), ReNcell CX line, derived from the cortical region of human foetal brain tissue (Sigma-Aldrich, #SCC007) were cultured either in 3D spheres in low-attachment flasks or in 2D monolayers on Matrigel-coated plates or flasks. hNPCs were maintained in DMEM/F12 media (Gibco, #21331020) supplemented with, 5 mM HEPES (Sigma-Aldrich, #51558), 2 mM GlutaMAX (Gibco, #35050061), 1X B-27 supplement (Gibco, #12587010), 2 µg/ml heparin (Sigma-Aldrich, #H3149), 20 ng/ml EGF (Peprotech, #AF-100-15), 20 ng/ml bFGF (Peprotech, #AF-100-18B), and 100 U/ml penicillin‒streptomycin (Gibco, #15140122).

## Human HepaSH hepatocyte cultures

Human HepaSH hepatocytes (HepaSH) were prepared at the Central Institute for Experimental Animals (Kawasaki, Japan) as previously described (Uehara et al, 2023). HepaSH cells derived from one human donor were used in this study. The vial content was transferred to a 50 ml tube containing 40 ml of PBS and centrifuged at $200 \times g$ for 2 min at 4 °C. The cell pellet was then resuspended in recovery medium (Biopredic International, #MIL130) and centrifuged at $200 \times g$ for 2 min at 4 °C. The cell pellet was then resuspended in seeding medium (Biopredic International, #MIL221) composed of Williams E GlutaMAX™ supplemented with 100 U/mL penicillin, 100 µg/mL streptomycin, 4 µg/mL bovine insulin and 10% foetal calf serum. HepaSH cells were then plated on collagen I-coated plates. After 24 h, the medium was discarded, and the HepaSH cells were maintained in culture medium (Biopredic International, #MIL222) composed of Williams E GlutaMAX™ supplemented with 100 U/mL penicillin, 100 µg/mL streptomycin, 4 µg/mL bovine insulin and 50 µM hydrocortisone. The media were renewed every 2 days. HepaSH cells, media and supplements were kindly provided by Biopredic International.

## Primary mouse cortical neuron cell cultures

Mouse cortical neurons were prepared from E17.5 C57BL/6N mouse embryos. Briefly, brains were isolated from embryos under a dissecting microscope, and the midbrain and thalamic tissues were gently removed to leave an intact hemisphere containing the cortex and hippocampus. The cortex was then dissected and dissociated using a papain solution (0.5 mM EDTA, 1 mM L-cysteine and

0.4 mg/mL papain (7 units of papain/mL) (Sigma-Aldrich, #76216)) at 37 °C in a water bath for 25 min. Neuron purification was performed using a neuron isolation kit (Miltenyi, #130-115-389) according to the manufacturer's instructions. After purification, neurons were plated at a density of $1 \times 10^5$ cells per cm$^2$ on plates coated with 50 μg/mL poly-D-lysine (Gibco, #A3890401) and 1 μg/mL laminin (Sigma-Aldrich, #L2020) in 2 mL of neurobasal media (Gibco, #21103049) supplemented with 1X B-27 (Gibco, #17504044), 100 U/mL penicillin–streptomycin (Gibco, #15140122), and 2 mM GlutaMAX (Gibco, #51558). Half of the media was replaced with fresh media every 2–3 days.

## Ectopic expression of NDI1

The pLenti6.3/V5 NDI1 and pLenti6.3/V5 plasmids were kindly provided by Dr. Joseph Marszalek (Molina et al, 2018). HEK293T cells were transfected with 75 μg of lentiviral plasmid (pLenti6.3/V5 NDI1 or pLenti6.3/V5), 50 μg of PAX2 packaging plasmid and 20 μg of SD11 viral envelope plasmid to produce the viruses. A total of $1 \times 10^5$ BTSCs were transduced in a 6-well plate with the virus. After overnight incubation, the virus was washed off, and BTSCs were replated in BTSC media. Transduced cells were selected through growth in 2.5 μg/ml blasticidin.

## Knockdown of NDUFS7 gene

Transient KD of NDUFS7 was achieved using siRNA approach using ON TARGET-plus SMART pool human NDUFS7 siRNA (Dharmacon, #L-031021-00-0005). For control siRNA ON TARGET-plus non-targeting pool (Dharmacon, #D-001810-10-05) was used. siRNA (100 nM) were nucleofected into BTSCs ($10^6$ cells). Twenty-four-hour after nucleofection, BTSCs were cultured in absence or presence of mubritinib for 7 days in BTSC media at 37 °C in a humidified atmosphere of 5% $CO_2$.

## Treatment of BTSCs

BTSCs were dissociated into single-cell suspensions using Accutase solution (Corning, #25-058-CI), plated in low-attachment plates or flasks, and treated with mubritinib (MedChemExpress, #HY-13501) at the time of plating at the indicated concentration or with vehicle (0.1% DMSO) for the indicated time points. For combinational treatments, cells were pre-treated for 4 h with mubritinib before exposure to IR or treatment with TMZ.

## Resipher real-time bioenergetic analysis

To measure the oxygen consumption rate (OCR), an indicator of mitochondrial metabolic activity, a Resipher real-time cell analyser (Lucid Scientific) was used. BTSCs were chemically dissociated using Accutase prior to plating. Cells were plated at a concentration of $1 \times 10^5$ cells/well in Nunc 96-well plates (Fischer Scientific, #10212811) in BTSC media. The plates were then incubated at 37 °C in a humidified atmosphere of 5% $CO_2$ for 4 h. Following incubation, BTSCs were placed on the Resipher system for real-time monitoring of the OCR in the absence or presence of 3 μM carbonyl cyanide-4-(trifluoromethoxy) phenylhydrazone (FCCP) to measure the basal and maximal respiration, respectively. For the mubritinib treatment conditions, following plating and a 4 h-incubation, mubritinib was added in the absence or presence of 3 μM FCCP, and the cells were subjected to OCR monitoring.

## High-resolution respirometry

Cellular oxygen uptake was quantified by high-resolution respirometry using an Oroboros® Oxygraph-2K (Oroboros Instruments, Innsbruck, Austria). Briefly, 24 h following treatment with mubritinib or vehicle control, BTSCs were chemically dissociated using Accutase, counted and suspended in BTSC medium at a concentration of $2 \times 10^6$ cells/ml. Five hundred microliters of the cell suspension was transferred into the Oroboros chamber to measure the $O_2$ concentration. The cell suspensions were continuously stirred at 750 rpm. Cellular respiration was quantified in terms of oxygen flux based on the rate of change in the $O_2$ concentration in the chambers in the absence or presence of FCCP to calculate the maximal respiration.

## MTT assay

MTT (3-(4,5-dimethylthiazol-2-yl)-2,5-diphenyl tetrazolium bromide) assays were performed to evaluate the cytotoxicity of mubritinib (Sharanek et al, 2019). Briefly, human hepatocytes and mouse neurons were seeded in 96-well plates and treated with various concentrations of mubritinib in triplicate for 4 days. After removing the culture medium, 100 μl of serum-free medium containing 0.5 mg/ml MTT (Sigma-Aldrich, #M5655) was added to each well and incubated for 2 h at 37 °C. The water-insoluble formazan was dissolved in 100 μl of dimethyl sulfoxide, and the absorbance was measured at 570 nm.

## Live cell counting

BTSCs or hNPCs were dissociated into single-cell suspensions using Accutase. Cells were seeded at a density of $3 \times 10^4$ cells/well in low-attachment 6-well plates and treated with mubritinib. The cell number was evaluated 7 days post-plating by PI cell viability flow cytometry using an Accuri C6 flow cytometer. The area under curve (AUC) of dose-response (0 to 100 nM) of mubritinib was calculated and a sensitivity score was defined using the formula:

$$Sensitivity\ score = 1 - \frac{AUC_x - AUC_{min}}{AUC_{max} - AUC_{min}}$$

where $AUC_x$ represents the AUC of the tested cell line, $AUC_{min}$ represents the AUC corresponding to 100% response and $AUC_{max}$ represents the AUC corresponding to 0% response. A sensitivity score close to 1 indicates high sensitivity, while a score close to 0 indicates low sensitivity.

## Cell death

Cell death in BTSCs or hNPCs was determined using a TACS annexin V-FITC apoptosis detection kit (R&D Systems, #4830-01-K) according to the manufacturer's protocol. Briefly, $3 \times 10^4$ cells/well were seeded in low-attachment 6-well plates and treated with mubritinib or vehicle control. After 7 days of treatment, co-staining with TACS annexin V-FITC and PI was performed on single cell suspension following the manufacturer's instructions. Single cell

suspension was resuspended in 50 μL of the Annexin V incubation reagent containing 0.5 μL of Annexin V and 5 μL of PI. The fluorescence was analysed by flow cytometry (Accuri C6 flow cytometer). Data were analysed using FlowJo software. Forward and side scatter density plots were used to exclude debris. Doublets were excluded by gating the events in forward scatter height versus forward scatter area. Density plots showing annexin V (X-axis) and PI (Y-axis) staining were generated. The quadrant gating was adjusted according to the negative control (unstained cells). Apoptotic cells positive for Annexin V can be seen in the lower right quadrant, and dead cells positive for both Annexin and PI can be seen in the upper right quadrant. Healthy cells negative for both stains are visualized in the lower left quadrant.

## Calcein AM cell viability assay

hNPCs were plated in 24-well Matrigel-coated plates at a density of $4 \times 10^4$ cells/well and treated with mubritinib. Four days after treatment, the number of live cells was estimated using calcein-AM (Invitrogen, #C1430)/DAPI double staining. The cells were incubated at 37 °C for 30 min with 0.5 μM calcein-AM and 0.1 μg/mL DAPI. Images were acquired using a 4X objective on an Eclipse Ti Nikon microscope coupled with NIS analysis software and a Hamamatsu Digital CCD C10600-10B camera. A minimum of 4 random fields were imaged and counted for each biological replicate. The number of live and dead cells was quantified with Fiji.

## Ionizing radiation

For measurement of the OCR following IR, BTSCs were dissociated into single cell suspension using Accutase. BTSCs were plated in low-attachment cell culture plates and then irradiated with 2 or 4 Gy using a Microbeam Technology Irradiation System (XenX Xstrahl®). The OCR was assessed 16 h post-irradiation using a Resipher Real-time Cell Analyzer (Lucid Scientific).

## Preparation of nuclear fractions

Three days following mubritinib treatment, BTSCs were collected and washed with 1X cold PBS. The cells were resuspended in 100 μl of ice-cold buffer composed of 10 mM HEPES, 60 mM KCl, 1 mM EDTA, 0.075% (v/v) NP-40, 1 mM dithiothreitol and 1 mM phenylmethylsulfonyl fluoride, adjusted to pH 7.6 and supplemented with protease and phosphatase inhibitors (Thermo Scientific, #A32959). The cells were then maintained on ice for 10 min. The mixture was then centrifuged to pellet the nuclei at $750 \times g$ for 5 min. The pelleted nuclei were washed once with 100 μl of the same buffer without NP-40. After washing, the supernatants were discarded and the pellets containing nuclei were resuspended in RIPA lysis buffer containing protease and phosphatase inhibitors, and sonicated briefly to shear genomic DNA and homogenize the lysate for immunoblotting.

## Immunoblotting and antibodies

Total proteins were harvested in RIPA lysis buffer containing protease and phosphatase inhibitors (Thermo Scientific, #A32959). The protein concentration was determined by Bradford assay (Sigma-Aldrich, #B6916), after which the samples were subjected to SDS–PAGE and electroblotted onto nitrocellulose membranes. The membranes were blocked in 5% bovine serum albumin in TBST before being sequentially probed with primary antibodies and IR-Dye 680 (LI-COR Biosciences, #926-68020 and #926-68023)- or IR-Dye 800 (LI-COR Biosciences, #926-32212 and #926-32211)-labelled secondary antibodies. Target proteins were visualized using an Odyssey infra-red scanner (LI-COR). The densitometry of proteins was quantified using Image Studio Lite Software and normalized to tubulin or vinculin. The following antibodies were used: Notch1 (1:1000, Cell Signaling Technology, #CST3608), Olig2 (1:2000, Abcam, #ab109186), SOX2 (1:200, Santa Cruz, #sc-365823), total AMPKα1 (1:1000, ABclonal, #A1229), p27$^{Kip1}$ (1:1000, ABclonal, #A0290), phospho-AMPKα (Thr172) (1:1000, Cell Signaling Technology, #2535), cyclin D1 (1:2000, Abcam, #134175), GFAP (1:2000, Cell Signaling Technology, #3670), p-Rb (1:1000, Proteintech, #30376-1-AP), α-tubulin (1:10,000, Sigma-Aldrich, #T5168) and vinculin (1:5000, Invitrogen, #MA511690).

## Extreme limiting dilution analysis

Decreasing numbers of live BTSCs per well (dose: 12, 6, 3 and 1) were plated in a low attachment 96-well plate using a BD FACSMelody™ Cell Sorter, with a minimum of 12 wells/dose, and treated with mubritinib or vehicle control. Twenty-one days after plating, the presence of spheres in each well was recorded and analysis was performed using software available at http://bioinf.wehi.edu.au/software/elda/ to determine the percentage of stem cell frequency (Hu and Smyth, 2009).

## EdU proliferation assay

BTSCs were dissociated into single cell suspension using Accutase, counted and plated at a density of $3 \times 10^4$ cells/well in low-attachment 6-well plates. At the time of plating, BTSCs were treated with mubritinib or vehicle control for 4 days. Then, EdU was added to the culture for 1 h. BTSCs were dissociated into single cell suspension and $3 \times 10^4$ cells were plated on 48-well plates coated with poly-D-lysine (Fischer Scientific, # 16021412). After the cells adhered to the plate, they were washed with 1X PBS, fixed, permeabilized and stained using the Click-iT EdU cell proliferation kit (Invitrogen, #C10340) according to the manufacturer's protocol. Images were acquired using a 10X objective on an Eclipse Ti Nikon microscope coupled with NIS analysis software and a Hamamatsu Digital CCD C10600-10B camera. The proportion of cells that incorporated EdU was determined as the ratio of EdU-positive cells to the total number of cells.

## Cell cycle analysis

BTSCs were dissociated into single cell suspension using Accutase and $3 \times 10^5$ cells were plated in T25 flasks and treated with mubritinib. After 24 h, the cells were dissociated into single-cell suspensions, harvested and fixed with 70% ethanol overnight at 4 °C. The cells were washed with 1X PBS and stained with FxCycle PI/RNase staining solution (Molecular Probes, #F10797) (Sharanek et al, 2021). The fluorescence was analysed by flow cytometry (Accuri C6 flow cytometer). The fractions of G0/G1-, S- and

G2-phase cells were determined using the Watson pragmatic algorithm of FlowJo software.

## Intracellular ROS production in vitro

ROS generation was measured using the probe 2′,7′-dichlorodihydrofluorescein diacetate ($H_2DCFDA$) (Invitrogen, #D399). Upon cleavage of the acetate groups by intracellular esterases and oxidation, the non-fluorescent $H_2DCFDA$ is converted to the highly fluorescent 2′,7′-dichlorofluorescein. Mitochondrial superoxide levels were measured using MitoSOX red (Invitrogen, #M36008). Following 3 days of treatment with mubritinib and irradiation under normoxic (21% $O_2$) or hypoxic (1% $O_2$), BTSC tumourspheres were loaded with 2 µM $H_2DCFDA$ or 1 µM MitoSOX red diluted in BTSC media. After 30 min of incubation (37 °C, 5% $CO_2$), BTSC tumourspheres were subjected to live imaging using a C2si confocal Nikon microscope. For flow cytometric analysis, after the 30 min incubation with probes, BTSCs were chemically dissociated using Accutase and the fluorescence was assessed by flow cytometry using an Accuri C6 flow cytometer. Data were analysed using the FlowJo software.

## Assessment of hypoxia in vitro

Hypoxia was assessed using the membrane-permeant hypoxia green reagent (Thermo Scientific, #H20035). Following 3 days of treatment with mubritinib under normoxic (21% $O_2$) or hypoxic (1% $O_2$) conditions, BTSC tumourspheres were loaded with 2 µM of the probe diluted in BTSC media. After 40 min of incubation (37 °C, 5% $CO_2$), BTSC tumourspheres were subjected to live confocal imaging using a C2si confocal Nikon microscope. For flow cytometric analysis, after the 40 min incubation with the probe, BTSCs were chemically dissociated using Accutase and the fluorescence was assessed by flow cytometry using an Accuri C6 flow cytometer. Data were analysed using the FlowJo software.

## Assessment of hypoxia using the GFP hypoxia reporter in vitro

The cDNA of the d2EGFP driven by the 5HRE-hCMVmp promoter, which consists of five copies of a 35-bp fragment from the hypoxia-responsive element (HRE) of the human VEGF gene and a human cytomegalovirus minimal promoter (hCMVmp) (Addgene, #46926), was subcloned into the lentiviral backbone, and lentiviral particles were prepared. A total of $1 \times 10^5$ BTSCs were transduced in a 6-well plate with the viral particles. After overnight incubation, the virus was washed off, and the BTSCs were replated in BTSC media. Following 3 days of mubritinib treatment under either normoxic (21% $O_2$) or hypoxic (1% $O_2$) conditions, BTSC tumourspheres expressing the hypoxia-responsive GFP reporter were subjected to live imaging using a C2si confocal Nikon microscope. For flow cytometric analysis, BTSCs were chemically dissociated using Accutase, and the fluorescence was assessed by flow cytometry using an Accuri C6 flow cytometer. Data were analysed using FlowJo software.

## Gene expression analysis

Total RNAs were isolated from cells using TRIzol reagent (Invitrogen) according to the manufacturer's instructions. RNAs were then subjected to reverse transcription using the 5X All-In-One RT MasterMix cDNA synthesis system (abm, #G492). Quantitative real-time PCR was performed using the fluorescent dye SYBR Green (Eurobio Scientific, #GAEMMX02H 0T). mRNA expression levels were then normalized to the housekeeping gene beta-glucuronidase (GUSB) or tubulin using the following primers:

*NDUFS7*-Fwd: GAGGTGTCCATCAGAGCGTG
*NDUFS7*-Rev: AGCTTGGCCACCACATACTC
*NDI1*-Fwd: TGCACCAGTTGGGACAGTAG
*NDI1*-Rev: AGCTGGCTAACGGCAGATAA
*GUSB*-Fwd: GCGTTCCTTTTGCGAGGAGA
*GUSB*-Rev: GGTGGTATCAGTCTTGCTCAA
*Tubulin*-Fwd: GAGTGCATCTCCATCCACGTT
*Tubulin*-Rev: TAGAGCTCCCAGCAGGCATT

## Whole-transcriptome analyses (RNA-seq)

Total RNA from BTSC25, BTSC53, BTSC73, BTSC147 and P3 was extracted with the Qiagen RNeasy Mini Kit according to the manufacturer's protocol. A DNase treatment of RNA has been added before RNA Cleanup using the RNase-Free DNase set from Qiagen following the manufacturer's instructions. Total RNA quality was checked using a Fragment Analyzer Systems (Agilent Technologies), RQN were between 8.3 and 10. One hundred nanograms of total RNA were converted into Stranded mRNA Library for mRNA-sequencing by using the Illumina Stranded mRNA Ligation kit according to Illumina's instructions. Briefly, poly(A) tailed RNA are captured by Oligo (dT) beads. The purified poly(A) RNA are fragmented and reverse transcribed into strand complementary cDNA. Then 3′end are adenylated for preparing fragment for the dual indexing adaptor ligation. Purified products are selectively amplified for library generation. All libraries have been quantified using a Qubit HS dsDNA kit (Thermo Fisher Scientific) and the library quality check has been performed using a High Sensitivity NGS Fragment Analysis Kit on a Fragment Analyzer System (Agilent Technologies). Indexed libraries were normalized and pooled to be sequenced on an Illumina Novaseq 6000 sequencer according to manufacturer's instructions with targeted conditions of 2 × 75 bp and 25 M reads/sample. Publicly available raw RNA-seq data for BTSC12, BTSC50, BTSC75 (Cusulin et al, 2015) were downloaded from NCBI Sequence Read Archive (accession number SRP057855) and converted into fastq files using *sra_toolkit* version 3.0.0.

## RNA-seq data preprocessing and differential analysis

Preprocessing steps consisted of (i) trimming of low quality reads, (ii) quality control, (iii) subsequent read mapping and (iv) count matrix construction. *fastp* version 0.23.4 and *MultiQC* version 1.9 were used for (i) and (ii), respectively. For mapping, *STAR* version 2.7.9a was used, with the Gencode release 45 reference human genome. A count matrix for all samples was generated retaining only uniquely mapped reads. All downtreams analyses were performed on the R software, version 4.4.1.

## Gene set variation analysis

The raw count matrix of BTSC12, 25, 50, 53, 73, 75, 147 and P3 was normalized using the function from the *DESeq2* package (Love et al,

2014) after removing genes with less than 10 mapped reads across all samples. A batch effect correction was performed between the BTSC12, 50, 75 and other cell lines using the ComBat function from the *sva* package (Hanzelmann et al, 2013; Leek et al, 2024). The list of genes of Wang and Garofano signatures were downloaded from corresponding original publications (Garofano et al, 2021; Wang et al, 2017), and then used for Gene Set Variation Analysis (GSVA). An enrichment score for each signature was computed among the cell lines using the gsva function from the *GSVA* package (Hanzelmann et al, 2013). Heatmaps for GSVA scores were then generated using the *pheatmap* package (Fig. EV2A,B).

## Differential expression analysis

Differential expression analysis between mubritinib against control conditions was performed on triplicates, using the raw count matrix and *DESeq2*, on genes with more than 10 mapped reads in at least 3 samples. Correspondence between Ensembl gene ID and hgnc symbols was done using the *biomaRt* package (Durinck et al, 2009). Statistical significance was established using the Wald test that yielded the *p*-values, further corrected for multiple testing using the Benjamini-Hochberg method. Genes were considered to be significantly differentially expressed if adjusted *p*-value < 0.05.

## Gene set enrichment analysis

Differentially expressed genes between control and mubritinib-treated cells were ranked by $-\log10(pvalue)*sign(log2FC)$. This ranked list of genes was subjected to GSEAusing the GSEA function of the *clusterProfiler* package (Xu et al, 2024). Reference gene sets from the Reactome pathway database were collected from the Molecular Signature Database (MSigDB) using the *msigdbr* package (Dolgalev, 2022). The output of GSEA analysis provided the Normalized Enrichment Score (NES), *p*-value and *p*-value adjusted for false discovery rate (p.adjust) for each analysed pathway. p.adjust < 0.05 was considered as significant. GSEA plots were generated using the gseaplot2 function of the *enrichplot* package (Yu and Gao, 2024).

## Quantification of mubritinib in the plasma and brain of mice

Following intraperitoneal injection of mice with mubritinib, brains and plasma were collected at different time points. One hundred microliters of plasma or 100 mg of brain tissue were ground with 100 μL of distilled water using a homogenizer. Then, 600 μL of acetonitrile with [$^{13}$C, $^{2}$H$_3$]-sorafenib (Darmstadt, Germany) were added and the samples were minced again. The samples were then centrifuged at 10,000 rpm for 5 min. Supernatants were collected and subjected to liquid chromatography/tandem mass spectrometry analysis. Liquid chromatography was performed on an Acquity UPLC system (Waters, USA) linked to MassLynx software. An Acquity Cortecs C18 + 1.6 μm 2.1 × 50 mm column (Waters, USA) was used. Mobile phase A was distilled water with 1% acetic acid and 0.1% formic acid, and mobile phase B was acetonitrile. The following gradient was applied: starting with 80% A/20% B mobile phases. B was increased to 40% from 0.2 to 0.4 min, to 60% from

0.4 to 0.6 min, to 80% from 0.6 to 0.8 min and to 90% from 0.8 to 1 min, maintained at 90% during 0.75 min, and then reduced to 20% from 2.1 min until the end of the analysis. The flow rate was 0.3 ml/min.

An Acquity TQD detector (Waters, Milford, USA) operated with an electrospray ionization (ESI) source in positive ion mode, using nitrogen as the nebulization and desolvation gas. The ESI ion source was operated at 150 °C, with the desolvation temperature at 400 °C, cone gas flow adjusted at 50 L/h, desolvation flow at 1000 L/h, and capillary voltage set up at 3.0 kV. MS collision was carried out by argon at $3 \times 10^{-3}$ mBar. Mass spectrometer settings are shown in Appendix Table S2.

## Stereotaxic injections and bioluminescent imaging

For intracranial injections, $3 \times 10^5$ patient-derived BTSCs were stereotactically implanted into the right striata (2 mm lateral to the bregma, 1 mm ventral and 3 mm from the pial surface) of 8-week-old male RAGγ2C$^{-/-}$ mice. For assessment of hypoxia using the GFP hypoxia reporter in vivo, BTSC73-expressing the GFP hypoxia reporter were stereotactically implanted into the right striata of 8-week-old male RAGγ2C$^{-/-}$ mice. For the syngeneic model, $5 \times 10^5$ luciferase-expressing murine mGB2 cells were injected into the right striata (2 mm lateral to bregma, 1 mm ventral to bregma and 3 mm from the pial surface) of 8-week-old male C57BL/6N mice. Mice were randomly assigned to the treatment or vehicle group. Seven days post BTSCs injection, the mice were intraperitoneally injected with 6 mg/kg mubritinib in a 5% DMSO and corn oil suspension or vehicle control 3 days a week (Monday, Wednesday and Friday) until the mice were sacrificed. To examine tumour volume, the animals were intraperitoneally injected with 200 μL of 15 mg/mL D-luciferin (AAT Bioquest, #AAT-12507), anaesthetized via isoflurane inhalation, and subjected to bioluminescence imaging using the PhotonIMAGER in vivo imaging system (Biospace Lab). All bioluminescence data were collected and analysed using M3 vision software. For KM survival plots, mice were collected when they showed signs of tumour-related illness.

## Combination of mubritinib and IR in vivo

Two treatment protocols were implemented to evaluate the combination of mubritinib and IR.

In the first protocol, 7 days after intracranial injection of BTSCs, mice were intraperitoneally injected with 6 mg/kg mubritinib in a 5% DMSO and corn oil suspension, or a vehicle control, three times a week (Monday, Wednesday, and Friday) until sacrifice. Mice received 2 Gy of localized IR on days 12 and 16 post-BTSC73 cell implantation, using the Microbeam Technology Irradiation System (XenX Xstrahl®).

In the second protocol, mubritinib administration began 11 days after intracranial injection of BTSCs. Twenty-four hours after the first mubritinib treatment (on day 12 post BTSC intracranial injection), and again on day 16, mice received 2 Gy of localized IR.

## Combination of mubritinib and TMZ in vivo

Eleven days post BTSCs intracranial stereotactic implantation, mice were randomly assigned to the treatment to one of the four

groups: vehicle control, mubritinib monotherapy (6 mg/kg), TMZ monotherapy (1 cycle of 5 days of 1 mg/kg), or a combination of mubritinib (6 mg/kg), and TMZ (1 cycle of 5 days of 1 mg/kg). The first mubritinib injection was started 24 h prior to TMZ.

## Treadmill fatigue test

After 3 months of repeated mubritinib treatment, mice were familiarized and trained to the treadmill apparatus by 2 training sessions on 2 separate days, as previously described (Dougherty et al, 2016). On the first day of training, mice were allowed to freely explore the treadmill for 3 min, after which the treadmill was turned on at a speed of 3.0 m/min to allow the mice to begin walking. After ensuring that all mice began walking, a 10 min training session was started using the following parameters: the start speed was set at 8 m/min; after 5 min, the speed was increased to 9 m/min; after 7 min, the speed was increased to 10 m/min, and the session was stopped at 10 min. On the second day of training, a 15 min training session was performed using the following parameters: the start speed was set at 10 m/min; after 5 min, the speed was increased to 11 m/min; after 10 min, the speed was increased to 12 m/min, and the treadmill was stopped after 15 min. For the test, the electric grid shockers that serve to encourage running of the mice were turned on, and the test was performed as previously described (Dougherty et al, 2016). At exhaustion, the time and speed were recorded, and the distance traversed was calculated. Exhaustion was defined as the point at which mice stopped running and remained on the electric grid shockers for more than 10 s. During the training sessions and the test, the treadmill angle of inclination was set to 5°.

## Immunohistochemical staining and analyses

Immunohistochemical stainings of brain tissue sections were performed following paraformaldehyde (PFA) fixation, paraffin embedding and sectioning. Briefly, mice were subjected to transcardiac perfusion with 4% PFA diluted in PBS, after which the brains were collected and incubated for 24 h in 4% PFA. The brains were then washed, dehydrated and embedded in paraffin. Five-micron-thick sections were cut using a microtome. The sections were then deparaffinized and subjected to antigen retrieval with Tris-EDTA buffer (pH 9). Slides were then permeabilized with 0.4% Triton-X100 and blocked with 5% normal donkey serum in PBS-Tween (0.05%). The slides were then incubated with the following primary antibodies diluted in PBS-Tween (0.05%) overnight at 4 °C: anti-Olig2 (1:100, Abcam, #ab109186), anti-cleaved caspase 3 (1:100, Cell Signaling Technology, #9661), anti-Ki67 (1:100, Abcam, #ab16667), anti-pH2AX (1:200, Cell Signaling, #9718), anti-GFP (1:1000, Torrey Pines, #TP401) and anti-SOX2 (1:100, Protein-tech, #66411-1). The slides were then washed and incubated with fluorescence-conjugated secondary antibodies (Invitrogen) at RT for 1 h, counterstained with 5 μg/ml DAPI, mounted using ProLong Diamond Antifade Mountant (Molecular Probes, # P36961), and imaged using a slide scanner (Hamamatsu Nanozoomer 2.0HT). For chromogenic detection, following primary antibody incubation, slides were washed and incubated

with 3% $H_2O_2$ for 10 min. Then, slides were washed and incubated with the secondary goat anti-rabbit biotinylated antibody (1:300, Dako, #E0432) for 1 h at room-temperature. After 3 washes, slides were incubated with the VECTASTAIN Elite ABC reagent (Vector Laboratories, #PK-6100) and DAB according to the manufacturer's protocol. Slides were conterstained with hematoxylin, mounted and scanned with a slide scanner system. For quantification of SOX2, Olig2 and Ki67 in tumours, the percentage of positive cells was determined using Fiji and normalized to the number of nuclei (minimum $18 \times 10^3$ nuclei per mouse, $n = 3$ mice). For quantification of SOX2 positive NPCs the number of SOX2 positive cells in SVZ was quantified using Fiji and normalized to the number of nuclei in the SVZ ($n = 3$ mice). For quantification of pH2AX, CC3 and GFP, the area of positive signal was determined using Fiji and normalized to the whole tumour area ($n = 3$ mice).

To estimate the tumour volume, 5 μm coronal sections were collected every 100 μm over a 600 μm span of the anterior-posterior axis of the brain ($n \geq 3$ mice). Hematoxylin-eosin staining was performed, and whole sections of the brain were imaged using a slide scanner (Hamamatsu Nanozoomer 2.0HT). Tumour volume was calculated using the following equation: $(L/3*(A1+ \sqrt{(A1*A2)} + A2)$. L is the distance between two sections, and A is the respective tumour area measured on each section. The volumes between each section were added to obtain the tumour volume at a distance of 600 μm.

## In vivo oxidative stress detection

To measure oxidative stress in the brain tumour tissues, freshly isolated brains, were fixed in Methacarn fixative solution (10% glacial acetic acid, 30% trichloromethane, 60% methanol) at 4 °C overnight. The brains were then washed, dehydrated and embedded in paraffin. Five-micron-thick sections were cut using a microtome. The sections were then deparaffinized and subjected to OxyIHC staining according to the OxyIHC oxidative stress detection kit protocol (Millipore, #S7450). The slides were then mounted using VectaMount® mounting media and whole sections of the brain were imaged using a slide scanner (Hamamatsu Nanozoomer 2.0HT). For quantifications, the area of positive signal was quantified using Fiji normalized by the tumour area.

## Statistical analysis

The experiments were randomized, and investigators were not blinded to allocation during experiments and outcome assessment. No data were excluded from the analyses. Statistical analysis was performed using ANOVA and Student's t test with the aid of GraphPad software 10. Two-tailed unpaired t tests were used to compare two conditions. One-way or two-way ANOVA with Tukey, Dunnett or Sidak post hoc test were used for analysing multiple groups. Data are shown as the mean and standard error of the mean (mean ± SEM). The log-rank test was used for statistical analysis of the Kaplan–Meier survival plot. $p$-values equal to or less than 0.05 were considered significant and are marked with an asterisk on the histograms. $p$-values less than 0.05 are denoted by *, $p$-values less than 0.01 are denoted by **, and $p$-values less than 0.001 are denoted by *** on the histograms.

## The paper explained

### Problem

Current standard therapies for glioblastoma, a highly aggressive brain tumour, include surgical excision, radiotherapy, and chemotherapy with temozolomide (TMZ). These treatments have however shown limited efficacy, largely due to the presence of brain tumour stem cells (BTSCs), which possess self-renewal properties. A significant challenge is to develop safe, brain-penetrant treatments that can suppress BTSCs and/or sensitize their response to current standard-of-care therapies.

### Results

Using multiple patient-derived BTSC models, we report that a drug called mubritinib effectively impairs BTSC stemness, self-renewal capacity, and growth. Mechanistic studies revealed that mubritinib targets complex I of the electron transport chain, disrupting mitochondrial respiration. Additionally, we found that mubritinib interferes with the AMPK/p27[Kip1] pathway, resulting in cell-cycle arrest in BTSCs. In vivo, mubritinib efficiently crosses the blood-brain barrier and accumulates in the brain, leading to tumour growth suppression and increased lifespan in treated patient-derived xenografted mice and syngeneic preclinical mouse models. Moreover, mubritinib improves oxygen diffusion in tumours by inhibiting mitochondrial oxygen consumption, which ultimately increases glioblastoma tumours' sensitivity to radiotherapy. We also demonstrated that mubritinib enhances the DNA-damaging effects of TMZ, making glioblastoma tumours more responsive to this treatment. Finally, in vitro and in vivo toxicological and behavioural studies revealed that mubritinib is well-tolerated and spares normal cells from damage.

### Impact

This work highlights mitochondrial respiration as a critical metabolic dependency and an exploitable vulnerability in BTSCs. Our findings show that mubritinib efficiently crosses the blood-brain barrier, impairs glioblastoma stem cells and tumours with minimal side effects, positioning it as a promising candidate for glioblastoma treatment. The combinatorial benefits of mubritinib with standard of care therapies further underscore its potential clinical impact.

## Data availability

RNA-seq data generated in this study are available in GEO database under the accession number GSE253362 and GSE284766.

The source data of this paper are collected in the following database record: biostudies:S-SCDT-10_1038-S44321-025-00195-6.

## Peer review information

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

## Acknowledgements

This work was supported by grants from Fondation de France N° 00130891 to ABu, and N° 00130896 to AS; Cancéropôle GSO N° 2023-E5 to AS; Association pour la Recherche sur les Tumeurs Cérébrales N° 283008 to ABu; Institut National du Cancer PLBIO N° 227441 and N° 241284 to TD; and SIRIC BRIO-Commucan and plan-cancer N°C24009GS to ABi. AS and ABu were supported by a fellowship from the Fondation de France. AS is also a recipient of the Marie Skłodowska-Curie Actions fellowship. CT is a recipient of the NewMoon PhD fellowship (Bordeaux University) and Region Nouvelle-Aquitaine. ML was supported by a PhD fellowship from the Fondation ARC (grant MD-DOC). We thank the staff (Marie-Alix Derieppe, Julie Martineau, Marcia Campistron and Marie-Paule Algeo) at the animal core facility "Animalerie Mutualisee de Talence", for assisting with the studies involving mice. We thank Atika Zouine, Vincent Pitard and Jean-Michel Griffon for technical assistance at the Flow cytometry facility, UAR 3427, INSERM US 05, Univ. Bordeaux, F-33000 Bordeaux, France. We thank the administrative staff at INSERM, UMR1312 (Stéphanie Lannelongue and Delphine Galin-Chené) for administrative assistance. We thank the authors of Molina et al (2018) for providing the NDI1 plasmids. We thank Dr. Peter Angel for providing the mGB2 cell line. We thank Dr. Deborah Tribouillard-Tanvier for assisting with the treadmill studies. We thank Dr. Arnaud Mourier for assistance and access to the Oroboros apparatus. We are grateful to Biopredic International (Saint

Grégoire, France) and the Central Institute for Experimental Animals (Kawasaki, Japan) for kindly providing the HepaSH cells. We thank Biopredic International for providing all the required media and supplements for the HepaSH cells.

## Author contributions

**Audrey Burban**: Conceptualization; Data curation; Formal analysis; Supervision; Funding acquisition; Validation; Investigation; Visualization; Methodology; Writing—original draft; Project administration; Writing—review and editing. **Cloe Tessier**: Data curation; Validation; Investigation; Methodology; Writing—review and editing. **Mathieu Larroquette**: Data curation; Investigation; Methodology; Writing—review and editing. **Joris Guyon**: Data curation; Investigation; Methodology; Writing—review and editing. **Cloe Lubiato**: Validation; Investigation; Methodology. **Mathis Pinglaut**: Validation; Investigation. **Maxime Toujas**: Formal analysis; Investigation; Writing—review and editing. **Johanna Galvis**: Data curation; Investigation. **Benjamin Dartigues**: Data curation; Investigation. **Emmanuelle Georget**: Validation; Investigation. **H Artee Luchman**: Resources. **Samuel Weiss**: Resources. **David Cappellen**: Resources. **Nathalie Nicot**: Data curation. **Barbara Klink**: Data curation. **Macha Nikolski**: Data curation. **Lucie Brisson**: Resources; Writing—review and editing. **Thomas Mathivet**: Resources; Writing—review and editing. **Andreas Bikfalvi**: Conceptualization; Supervision; Funding acquisition; Visualization; Project administration; Writing—review and editing. **Thomas Daubon**: Conceptualization; Supervision; Funding acquisition; Visualization; Project administration; Writing—review and editing. **Ahmad Sharanek**: Conceptualization; Data curation; Formal analysis; Supervision; Funding acquisition; Validation; Investigation; Visualization; Methodology; Writing—original draft; Project administration; Writing—review and editing.

Source data underlying figure panels in this paper may have individual authorship assigned. Where available, figure panel/source data authorship is listed in the following database record: biostudies:S-SCDT-10_1038-S44321-025-00195-6.

## Disclosure and competing interests statement
The authors declare no competing interests.

# Expanded View Figures

**Figure EV1.  Mubritinib decreases the proliferation of patient- and murine-derived BTSCs without inducing cell death.**   ▶

(A) Basal mitochondrial respiration, maximal mitochondrial respiration and spare respiratory capacity (SRC) were measured in BTSC73 following mubritinib treatment using the high-resolution respirometer Oroboros. Data are presented as the means ± SEM, $n = 3$ independent biological experiments. One-way ANOVA followed by Dunnett's test vs control. $**p_{basal\ (20)} = 0.0012$; $***p_{basal\ (100)} = 2.9e{-}5$; $***p_{basal\ (500)} = 1.1e{-}5$, $*p_{maximal\ (20)} = 0.0228$, $***p_{maximal\ (100)} = 0.0009$, $***p_{maximal\ (500)} = 0.0006$, $*p_{SRC\ (100)} = 0.0488$, $*p_{SRC\ (500)} = 0.0377$. (B) The murine glioblastoma cells, mGB2, were exposed to increasing concentrations of mubritinib (0 to 500 nM), followed by live cell counting. Data are presented as the means ± SEM, $n = 3$ independent biological experiments. One-way ANOVA followed by Dunnett's test vs control. $***p_{mubritinib\ 100} = 0.0009$, $***p_{mubritinib\ 500} = 4.4e{-}5$. (C) The percentages of dead cells (PI positive) were measured by PI staining followed by flow cytometry in BTSCs treated with increasing concentrations of mubritinib for 7 days. Data are presented as the means ± SEM, $n = 3$ independent biological experiments. One-way ANOVA followed by Dunnett's test vs control. $p$-values are higher than 0.05 for all conditions. (D) The percentages of dead cells (PI positive) and early apoptotic cells (Annexin V positive and PI negative) were measured by Annexin V/PI double staining followed by flow cytometry in BTSCs treated with increasing concentrations of mubritinib for 7 days. (E–G) Pearson correlation analysis was performed between basal OCR and the percentage of live cells in BTSC following 7 days treatment with mubritinib at 20 nM (E) 100 nM (F) and 500 nM (G). (H) Pearson correlation analysis was performed between maximal OCR and sensitivity score to mubritinib following 7 days of treatment. (I–K) mRNA levels of *NDI1* gene were assessed by RT-qPCR in BTCS53 (I), BTSC73 (J) and BTSC147 (K) transduced with control or NDI1 vector. Data are presented as the means ± SEM, $n = 3$ independent biological experiments. (L–N) Basal OCR was measured by Resipher system in BTSC53 (L), BTSC73 (M) and BTSC147 (N) expressing the control (CTL) vector or NDI1 following treatment with vehicle control or 500 nM mubritinib. Data are presented as the means ± SEM, $n = 3$ independent biological experiments. Two-way ANOVA followed by Tukey's test. BTSC53 (L): $***p_{CTL\ vs\ mubritinib} = 1.8e{-}12$, $p_{NDI1\ vs\ NDI1\ +\ mubritinib} = 0.1702$. BTSC73 (M): $***p_{CTL\ vs\ mubritinib} = 4.9e{-}13$, $p_{NDI1\ vs\ NDI1\ +\ mubritinib} = 0.6586$. BTSC147 (N): $***p_{CTL\ vs\ mubritinib} = 1e{-}10$, $p_{NDI1\ vs\ NDI1\ +\ mubritinib} = 0.0828$. (O) Proliferation curve was generated using the CFSE assay in BTSC73 expressing the CTL vector or NDI1 following treatment with vehicle control or 500 nM mubritinib. Data are presented as the means ± SEM, $n = 3$ independent biological experiments. Two-way ANOVA followed by Tukey's test. $*p_{3d\ (CTL\ vs\ mubritinib)} = 0.0267$, $*p_{4d\ (CTL\ vs\ mubritinib)} = 0.0265$, $*p_{5d\ (CTL\ vs\ mubritinib)} = 0.0312$, $**p_{6d\ (CTL\ vs\ mubritinib)} = 0.0011$, $p_{NDI1\ vs\ NDI1\ +\ mubritinib} > 0.05$ at all the time points. (P, Q) mRNA levels of *NDUFS7* gene were assessed by RT-qPCR in BTCS73 (P), BTSC53 (Q) electroporated with either siCTL or si*NDUFS7*. Data are presented as the means ± SEM, $n = 3$ independent biological experiments. Unpaired two-tailed t test. $***p_{BTSC73} = 5e{-}8$, $***p_{BTSC53} = 4.8e{-}6$. (R, S) BTSC73 (R) and BTSC53 (S) were electroporated with either siCTL or si*NDUFS7* and treated with 500 nM of mubritinib for 7 days, followed by live cell counting. Data are presented as the means ± SEM, $n = 3$ independent biological experiments. One-way ANOVA followed by Tukey's test. BTSC73 (R): $***p_{siCTL\ vs\ siCTL\ +\ mubritinib} = 1.6e{-}6$, $***p_{siCTL\ vs\ siNDUFS7} = 5.3e{-}6$, $p_{siNDUFS7\ vs\ siNDUFS7\ +\ mubritinib} = 0.0690$. BTSC53 (S): $***p_{siCTL\ vs\ siCTL\ +\ mubritinib} = 3e{-}5$, $***p_{siCTL\ vs\ siNDUFS7} = 6.8e{-}6$, $p_{siNDUFS7\ vs\ siNDUFS7\ +\ mubritinib} = 0.5667$. (T) BTSCs and differentiated progeny counterparts (Diff) were subjected to immunoblotting using the antibodies indicated on the blots. Vinculin is used as loading control. (U–X) The number of live stem and differentiated cells from BTSC12 (U), BTSC53 (V), BTSC73 (W) and BTSC147 (X) was measured following 500 nM mubritinib treatment. Data are presented as the means ± SEM, $n = 3$ independent biological experiments. One-way ANOVA followed by Tukey's test. BTSC12 (U): $***p_{stem\ control\ vs\ mubritinib} = 0.0002$, $*p_{diff\ control\ vs\ mubritinib} = 0.0426$, $**p_{stem\ mubritinib\ vs\ diff\ mubritinib} = 0.0078$. BTSC53 (V): $***p_{stem\ control\ vs\ mubritinib} = 9.4e{-}5$, $**p_{stem\ mubritinib\ vs\ diff\ mubritinib} = 0.0010$. BTSC73 (W): $***p_{stem\ control\ vs\ mubritinib} = 0.0001$, $*p_{diff\ control\ vs\ mubritinib} = 0.0322$, $**p_{stem\ mubritinib\ vs\ diff\ mubritinib} = 0.0049$. BTSC147 (X): $***p_{stem\ control\ vs\ mubritinib} = 5.1e{-}5$, $***p_{stem\ mubritinib\ vs\ diff\ mubritinib} = 0.0007$.

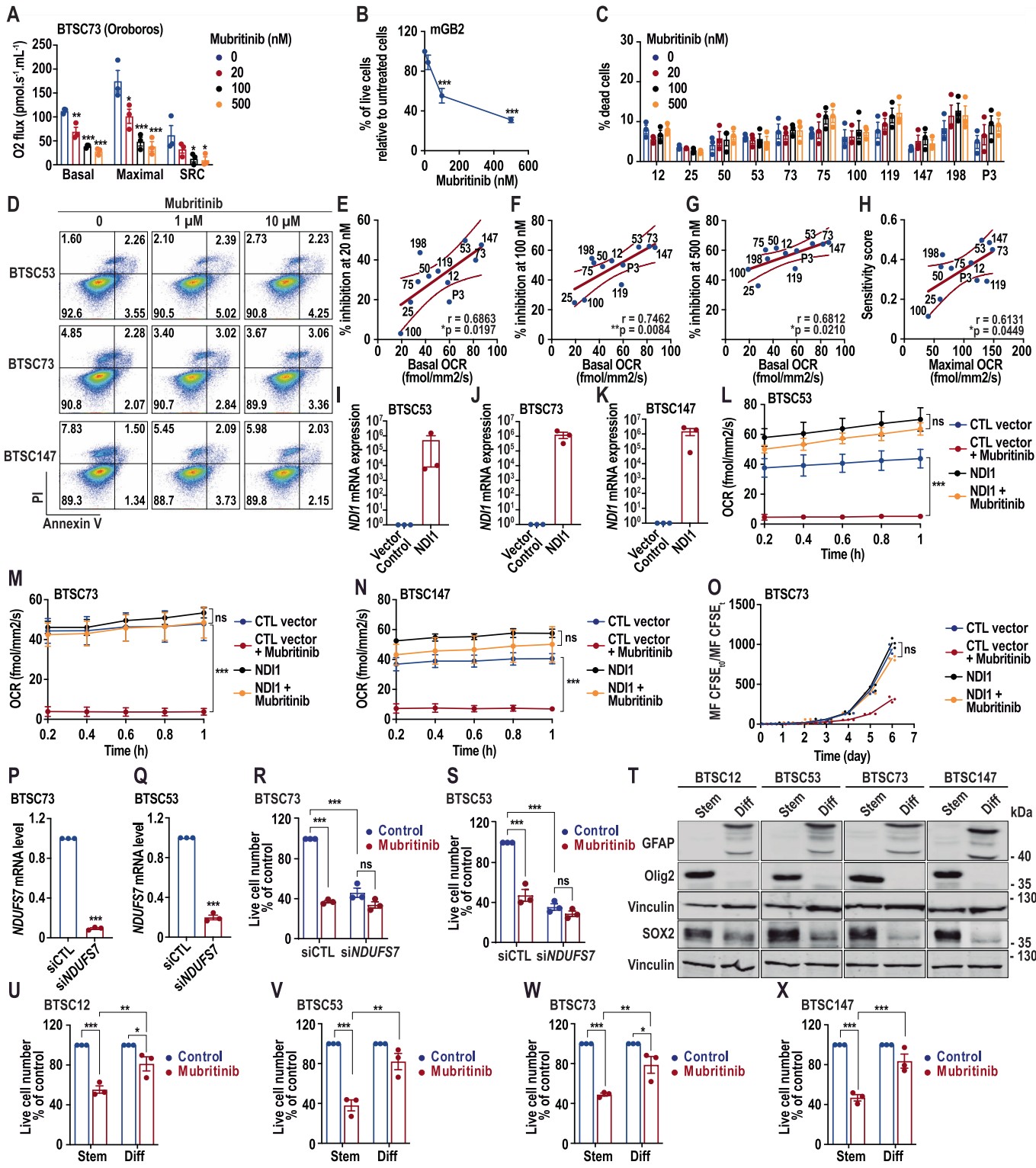

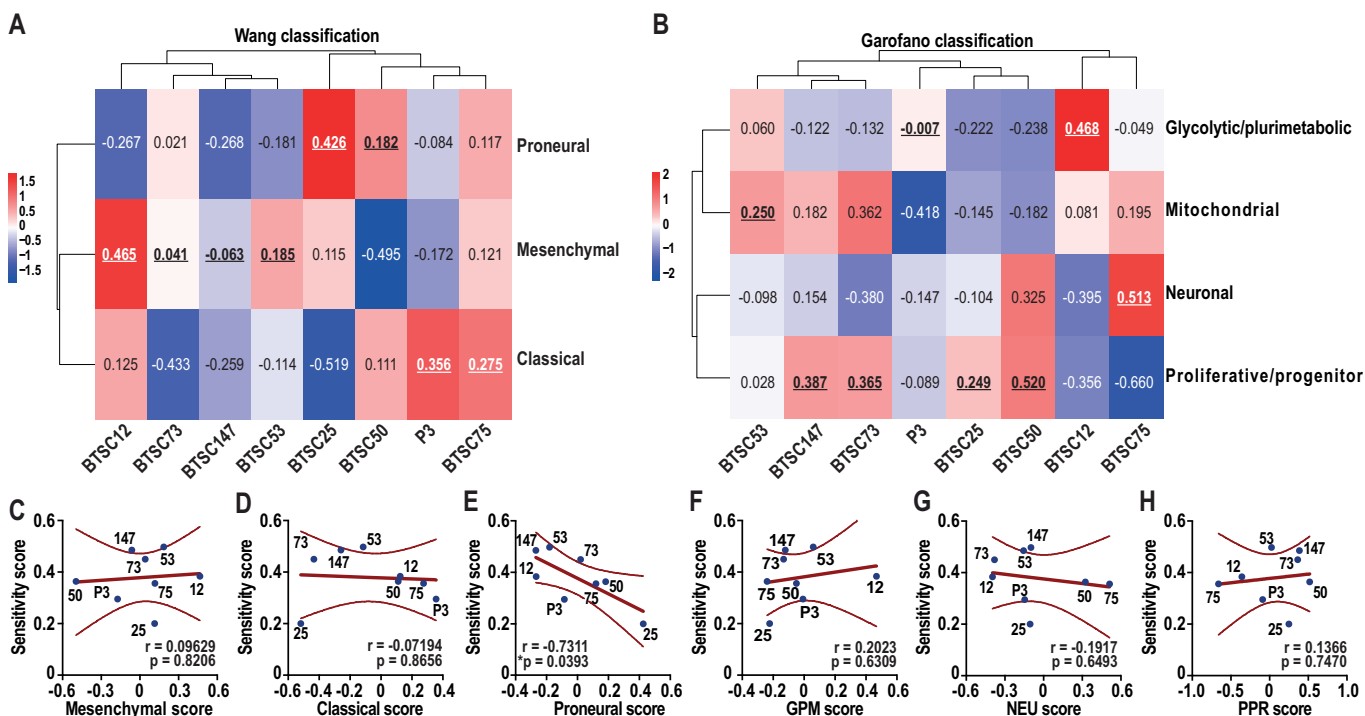

**Figure EV2.  Transcriptional classification scores of BTSCs and correlation analysis with sensitivity to mubritinib.**

(**A, B**) Heatmaps illustrating the hierarchical clustering of genes related to Wang (**A**) and Garofano (**B**) glioblastoma transcriptional subtypes. Dominant transcriptional subtype score is underlined and highlighted in bold. (**C–H**) Pearson correlation analysis was performed between the BTSC sensitivity score to mubritinib following 7 days of treatment and mesenchymal (**C**), classical (**D**), proneural (**E**), glycolytic/plurimetabolic (GPM) (**F**), neuronal (NEU) (**G**) and proliferative/progenitor (PPR) (**H**) transcriptional subtype signature scores.

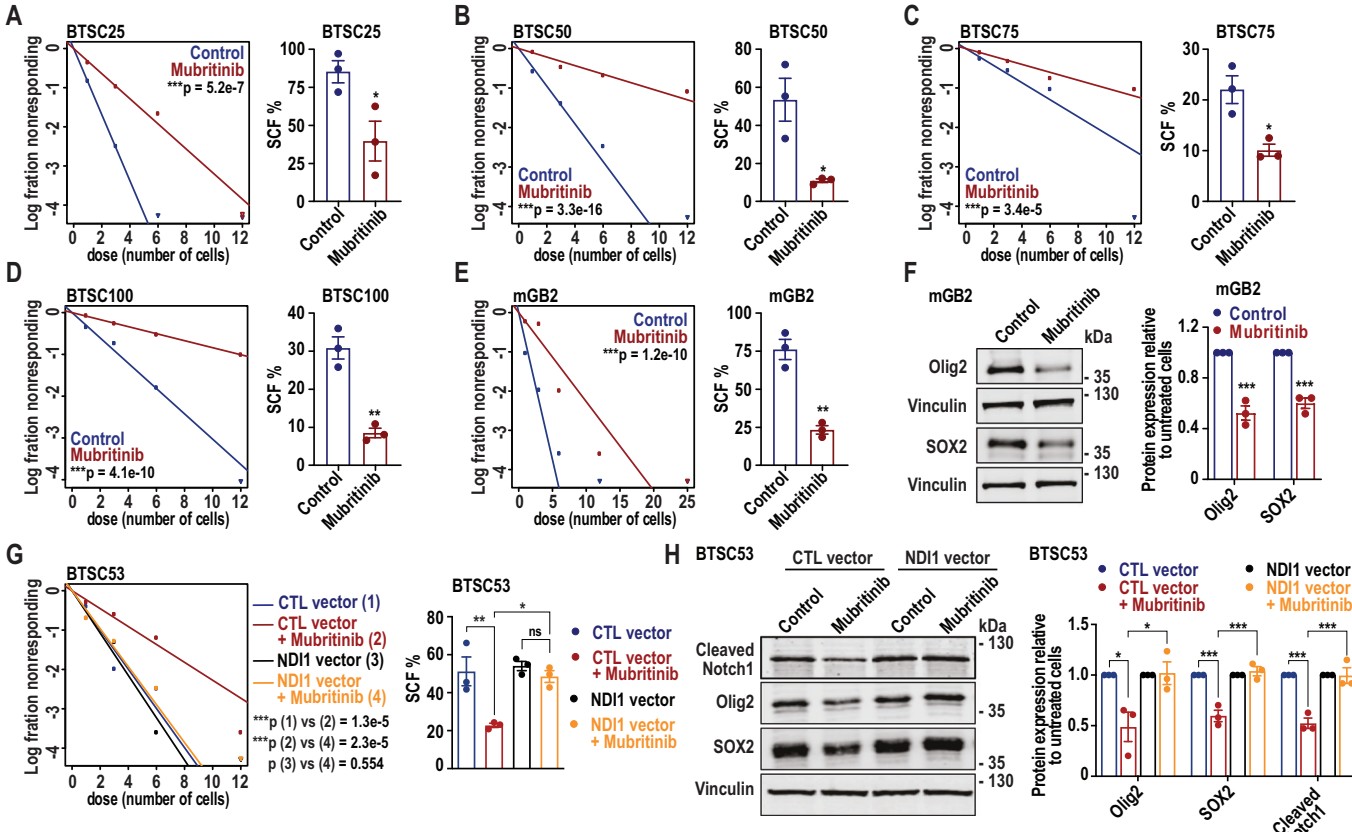

**Figure EV3. Validation of the effect of mubritinib on the stemness of murine-derived and additional patient-derived BTSCs.**

(A–D) BTSC25 (A), BTSC50 (B), BTSC75 (C) and BTSC100 (D) were subjected to ELDA to estimate the SCF, 21 days following treatment with 500 nM mubritinib or vehicle control. Data are presented as the means ± SEM, $n = 3$ independent biological experiments. Chi-square test for ELDA plots and unpaired two-tailed t test for SCF were used. $*p_{BTSC25} = 0.038$, $*p_{BTSC50} = 0.0196$, $*p_{BTSC75} = 0.016$, $**p_{BTSC100} = 0.0022$. (E) The murine-derived BTSCs (mGB2) were subjected to ELDA to estimate the SCF, 7 days following treatment with 500 nM mubritinib or vehicle control. Data are presented as the means ± SEM, $n = 3$ independent biological experiments. Chi-square test for ELDA plots and unpaired two-tailed t test for SCF were used. $**p_{mGB2} = 0.0018$. (F) mGB2 cells were treated for 4 days with 500 nM mubritinib or vehicle control and subjected to immunoblotting using the antibodies indicated on the blots. Vinculin was used as loading control. Densitometric quantifications of Olig2 and SOX2 protein levels normalized to their corresponding loading controls are presented. Data are presented as the means ± SEM, $n = 3$ independent biological experiments. One-way ANOVA followed by Dunnett's test vs vehicle control. $***p_{Olig2} = 0.0003$, $***p_{SOX2} = 0.0007$. (G) BTSC53 transduced with the control (CTL) or NDI1 vector were treated with vehicle control or 500 nM mubritinib and subjected to ELDA to estimate the SCF, 21 days following treatment. Data are presented as the means ± SEM, $n = 3$ independent biological experiments. Chi-square test for ELDA plots and one-way ANOVA followed by Tukey's test for SCF were used. $**p_{CTL vs mubritinib} = 0.0069$, $*p_{mubritinib vs NDI1 + mubritinib} = 0.0121$, $p_{NDI1 vs NDI1 + mubritinib} = 0.8008$. (H) BTSC53 transduced with the CTL or NDI1 vector were treated with vehicle control or 500 nM mubritinib for 4 days and subjected to immunoblotting using the antibodies indicated on the blots. Vinculin was used as loading control. Densitometric quantifications of cleaved Notch1, Olig2 and SOX2 protein levels normalized to their corresponding loading controls are presented. Data are presented as the means ± SEM, $n = 3$ independent biological experiments. One-way ANOVA followed by Tukey's test. $*p_{Olig2 (CTL vs mubritinib)} = 0.0174$, $*p_{Olig2 (Mubritinib vs NDI1 + mubritinib)} = 0.0142$, $p_{Olig2 (NDI1 vs NDI1 + mubritinib)} = 0.9986$, $***p_{SOX2 (CTL vs mubritinib)} = 0.0002$, $***p_{SOX2 (mubritinib vs NDI1 + mubritinib)} = 0.0001$, $p_{SOX2 (NDI1 vs NDI1 + mubritinib)} = 0.8669$, $***p_{Cleaved Notch1 (CTL vs mubritinib)} = 0.0004$, $***p_{Cleaved Notch1 (mubritinib vs NDI1 + mubritinib)} = 0.0004$, $p_{Cleaved Notch1 (NDI1 vs NDI1 + mubritinib)} > 0.9999$.

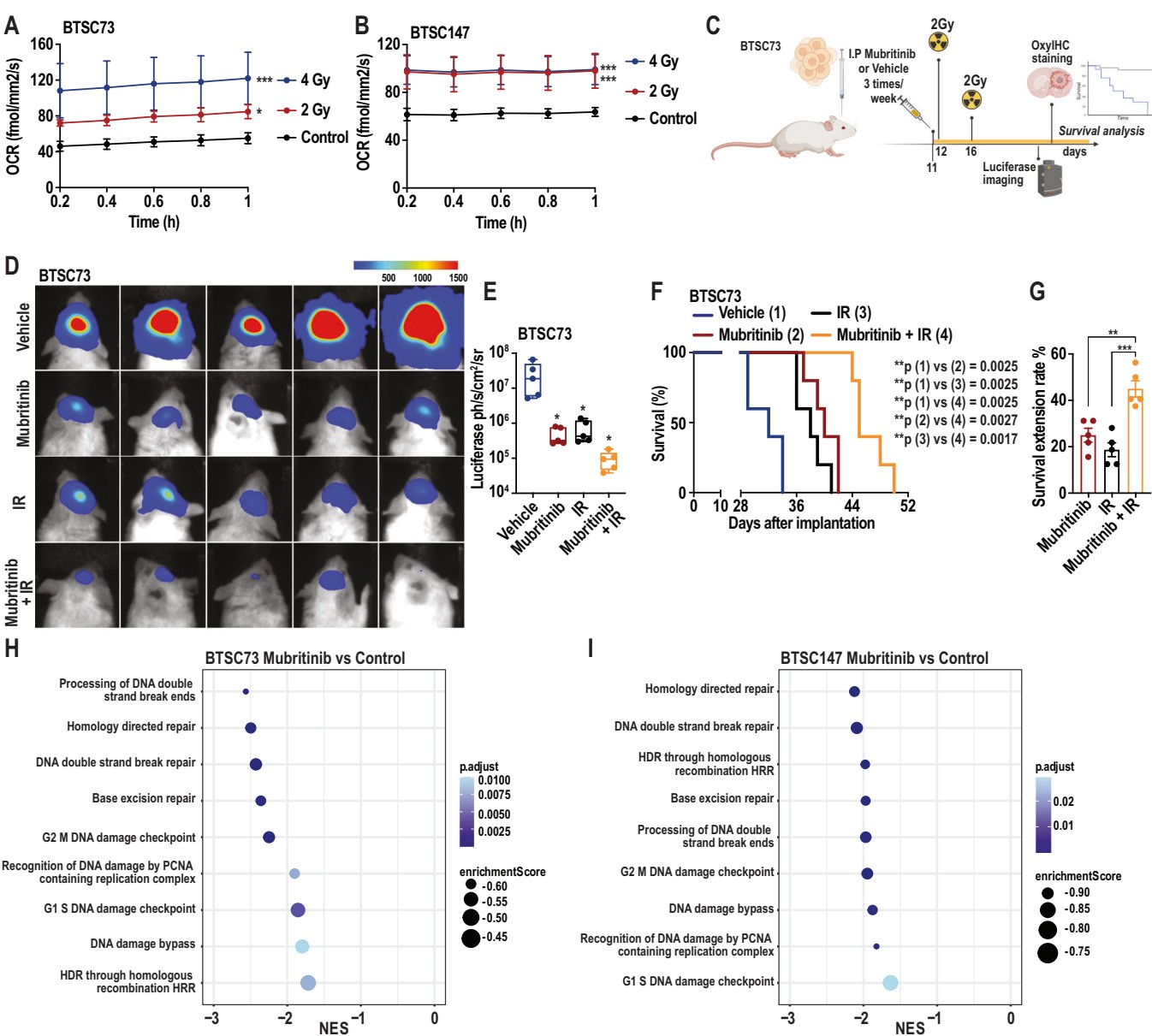

**Figure EV4. Mubritinib sensitizes GB tumours to ionizing radiation.**

(A, B) 2 Gy or 4 Gy irradiated BTSC73 (A) BTSC147 (B) were subjected to real-time Resipher analysis to measure the basal OCR 16 h following irradiation. Data are presented as the means ± SEM, $n = 3$ independent biological experiments. Two-way ANOVA followed by Dunnett's test vs vehicle control. BTSC73 (A): $*p_{control\ vs\ 2\ Gy} = 0.0357$, $***p_{control\ vs\ 4\ Gy} = 6e-6$. BTSC147 (B): $***p_{control\ vs\ 2\ Gy} = 6.8e-6$, $***p_{control\ vs\ 4\ Gy} = 3.3e-6$. (C) Schematic diagram of the experimental procedure in which luciferase-expressing BTSC73 cells were intracranially injected into RAGγ2C$^{-/-}$ mice. 11 days after implantation, the mice were randomized into 2 groups: vehicle control or mubritinib (6 mg/kg). Mice were treated 3 times per week (Monday, Wednesday and Friday). For IR groups, 24 h after the first mubritinib injection, mice received IR (2 Gy). On day 16, mice were subjected to another cycle of IR at 2 Gy. (D, E) Bioluminescence images (D) and quantification of luciferase activity (E) are presented. Data are presented as box plots showing 25th and 75th percentiles (box), median (centre line), minima and maxima (whiskers), $n = 5$ mice. One-way ANOVA followed by Tukey's test. $*p_{vehicle\ vs\ mubritinib} = 0.0244$, $*p_{vehicle\ vs\ IR} = 0.0258$, $*p_{vehicle\ vs\ mubritinib\ +\ IR} = 0.0222$. (F) KM survival plot was graphed to evaluate mice lifespan in each group, mice were collected at end stage (log-rank test, $n = 5$ mice). (G) Survival extensions of mice bearing BTSC73-derived tumours treated with mubritinib, IR, or mubritinib + IR relative to those treated with the vehicle control were calculated. Data are presented as the means ± SEM, $n = 5$ mice. One-way ANOVA followed by Tukey's test. $**p_{mubritinib\ vs\ IR\ +\ mubritinib} = 0.0020$, $***p_{IR\ vs\ IR\ +\ mubritinib} = 0.0002$. (H, I) Gene set enrichment analysis of deregulated genes in BTSC73 (H) and BTSC147 (I) treated with 500 nM of mubritinib for 24 h demonstrates enrichment of gene sets corresponding to DNA repair pathways. Permutation test was used to calculate $p$-values, which were then corrected using the Benjamini-Hochberg method to obtain adjusted $p$-values (p.adjust).

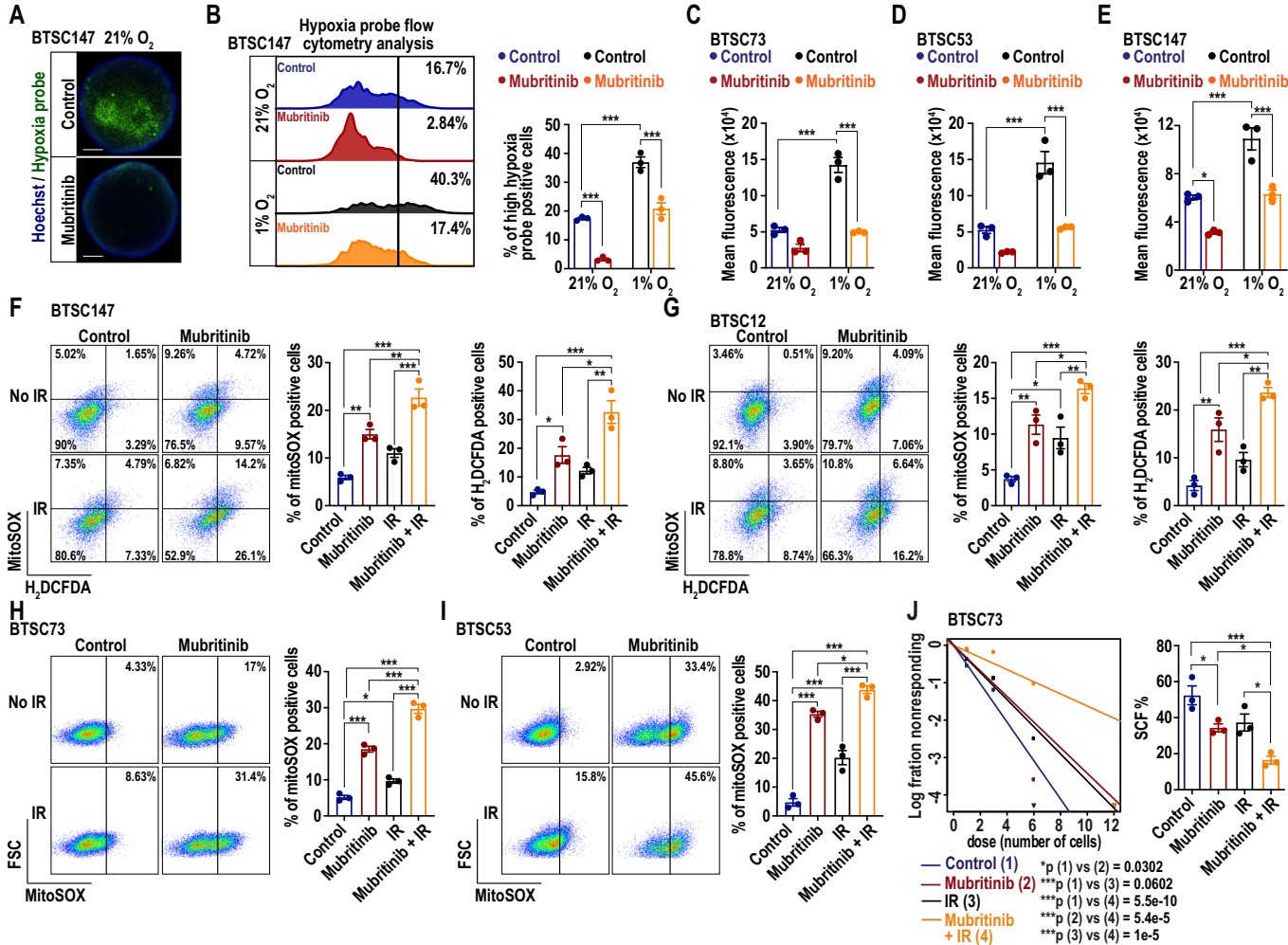

**Figure EV5.   Mubritinib alleviates tumour hypoxia and enhances oxidative stress to sensitize GB tumours to IR.**

(A) BTSC147 tumourspheres were treated with 500 nM mubritinib and subjected to labelling with hypoxia probe (green) followed by live fluorescent imaging. Nuclei were stained by Hoechst. Scale bar = 100 μm. (B) BTSC147 tumourspheres were treated with 500 nM mubritinib under normoxic (21% $O_2$) or hypoxic (1% $O_2$) conditions and subjected to labelling with hypoxia probe followed by flow cytometric analysis. The percentages of high hypoxia probe-positive cells are shown. Data are presented as the means ± SEM, $n = 3$ independent biological experiments. One-way ANOVA followed by Tukey's test. ***$p_{control\ 21\%\ O_2\ vs\ mubritinib\ 21\%\ O_2}$ = 0.0005, ***$p_{control\ 21\%\ O_2\ vs\ control\ 1\%\ O_2}$ = 7.4e−7, ***$p_{control\ 1\%\ O_2\ vs\ mubritinib\ 1\%\ O_2}$ = 0.0002. (C–E) BTSC73 (C), BTSC53 (D) and BTSC147 (E) tumourspheres were treated with 500 nM mubritinib under normoxic or hypoxic conditions and subjected to labelling with hypoxia probe followed by flow cytometric analysis. The mean fluorescence of hypoxia probe is shown. Data are presented as the means ± SEM, $n = 3$ independent biological experiments. One-way ANOVA followed by Tukey's test. BTSC73 (C): ***$p_{control\ 21\%\ O_2\ vs\ control\ 1\%\ O_2}$ = 2.8e−5, ***$p_{control\ 1\%\ O_2\ vs\ mubritinib\ 1\%\ O_2}$ = 2.2e−5. BTSC53 (D): ***$p_{control\ 21\%\ O_2\ vs\ control\ 1\%\ O_2}$ = 0.0002, ***$p_{control\ 1\%\ O_2\ vs\ mubritinib\ 1\%\ O_2}$ = 0.0002. BTSC147 (E): *$p_{control\ 21\%\ O_2\ vs\ mubritinib\ 21\%\ O_2}$ = 0.0148, ***$p_{control\ 21\%\ O_2\ vs\ control\ 1\%\ O_2}$ = 0.0006, ***$p_{control\ 1\%\ O_2\ vs\ mubritinib\ 1\%\ O_2}$ = 0.0009. (F, G) BTSC147 (F) and BTSC12 (G) tumourspheres were treated with 500 nM mubritinib, IR 2 Gy, or combination of both for 3 days and subjected to labelling with $H_2$DCFDA and MitoSOX followed by flow cytometric analysis. The percentages of $H_2$DCFDA or MitoSOX positive cells are presented. Data are presented as the means ± SEM, $n = 3$ independent biological experiments. One-way ANOVA followed by Tukey's test. BTSC147 (F): **$p_{mitoSOX\ (control\ vs\ mubritinib)}$ = 0.0023, ***$p_{mitoSOX\ (control\ vs\ mubritinib\ +\ IR)}$ = 3.1e−5, **$p_{mitoSOX\ (mubritinib\ vs\ mubritinib\ +\ IR)}$ = 0.0065, ***$p_{mitoSOX\ (IR\ vs\ mubritinib\ +\ IR)}$ = 0.0004. *$p_{H_2DCFDA\ (control\ vs\ mubritinib)}$ = 0.0287, ***$p_{H_2DCFDA\ (control\ vs\ mubritinib\ +\ IR)}$ = 0.0002, *$p_{H_2DCFDA\ (mubritinib\ vs\ mubritinib\ +\ IR)}$ = 0.0129, **$p_{H_2DCFDA\ (IR\ vs\ mubritinib\ +\ IR)}$ = 0.0020. BTSC12 (G): **$p_{mitoSOX\ (control\ vs\ mubritinib)}$ = 0.005, *$p_{mitoSOX\ (control\ vs\ IR)}$ = 0.0237, ***$p_{mitoSOX\ (control\ vs\ mubritinib\ +\ IR)}$ = 0.0002, *$p_{mitoSOX\ (mubritinib\ vs\ mubritinib\ +\ IR)}$ = 0.0453, **$p_{mitoSOX\ (IR\ vs\ mubritinib\ +\ IR)}$ = 0.0089. **$p_{H_2DCFDA\ (control\ vs\ mubritinib)}$ = 0.0040, ***$p_{H_2DCFDA\ (control\ vs\ mubritinib\ +\ IR)}$ = 0.0001, *$p_{H_2DCFDA\ (mubritinib\ vs\ mubritinib\ +\ IR)}$ = 0.0386, **$p_{H_2DCFDA\ (IR\ vs\ mubritinib\ +\ IR)}$ = 0.0013. (H, I) BTSC73 (H) and BTSC53 (I) tumourspheres were treated with 500 nM mubritinib, IR 2 Gy, or combination of both for 3 days under hypoxic conditions (1% $O_2$) and subjected to labelling with MitoSOX followed by flow cytometric analysis. The percentages of MitoSOX positive cells are presented. Data are presented as the means ± SEM, $n = 3$ independent biological experiments. One-way ANOVA followed by Tukey's test. BTSC73 (H): ***$p_{control\ vs\ mubritinib}$ = 1.7e−5, *$p_{control\ vs\ IR}$ = 0.0226, ***$p_{control\ vs\ mubritinib\ +\ IR}$ = 1.3e−7, ***$p_{mubritinib\ vs\ mubritinib\ +\ IR}$ = 6.4e−5, ***$p_{IR\ vs\ mubritinib\ +\ IR}$ = 7.7e−7. BTSC53 (I): ***$p_{control\ vs\ mubritinib}$ = 4.5e−6, ***$p_{control\ vs\ IR}$ = 0.0006, ***$p_{control\ vs\ mubritinib\ +\ IR}$ = 6.6e−7, *$p_{mubritinib\ vs\ mubritinib\ +\ IR}$ = 0.0241, ***$p_{IR\ vs\ mubritinib\ +\ IR}$ = 3.2e−5. (J) BTSC73 treated with 500 nM mubritinib, irradiated with 2 Gy, and subjected to ELDA, under hypoxic condition (1% $O_2$), to estimate the SCF. Data are presented as the means ± SEM, $n = 3$ independent biological experiments. Chi-square test for ELDA plots and one-way ANOVA followed by Tukey's test for SCF were used. *$p_{control\ vs\ mubritinib}$ = 0.0419, ***$p_{control\ vs\ mubritinib\ +\ IR}$ = 0.0007, *$p_{mubritinib\ vs\ mubritinib\ +\ IR}$ = 0.0435, *$p_{IR\ vs\ mubritinib\ +\ IR}$ = 0.0206.

