## [Peer Review File · EMBO Molecular Medicine]

Exploiting metabolic vulnerability in glioblastoma using a brain-penetrant drug with a safe profile

Audrey Burban, Cloe Tessier, Mathieu Larroquette, Joris Guyon, Cloe Lubiato, Mathis Pinglout, Maxime Toujas, Johanna Rodriguez, Benjamin Dartigues, Emmanuelle Georget, H Artee Luchman, Samuel Weiss, David Cappellen, Nathalie Nicot, Barbara Klink, Macha Nikolski, Lucie Brisson, Thomas Mathivet, Andreas Bikfalvi, Thomas Daubon, and Ahmad Sharanek

Corresponding author(s): Ahmad Sharanek (ahmad.charanek@u-bordeaux.fr) , Andreas Bikfalvi (andreas.bikfalvi@u-bordeaux.fr), Thomas Daubon (thomas.daubon@u-bordeaux.fr)

Review Timeline:

Submission Date:	15th Mar 24
Editorial Decision:	20th Mar 24
Revision Received:	20th Nov 24
Editorial Decision:	16th Dec 24
Revision Received:	4th Jan 25
Editorial Decision:	12th Jan 25
Revision Received:	13th Jan 25
Accepted:	14th Jan 25

Editor: Lise Roth

Transaction Report:

20th Mar 2024

Dear Dr. Sharanek,

Thank you for the submission of your manuscript to our editorial offices. I have now had the opportunity to read it, together with the referees' reports and your rebuttal letter, and to discuss them with the other members of our editorial team.

We agree that the study fits the scope of the journal, and we appreciate that you are willing to address the referees' concerns. We thus encourage you to submit a revised version of your manuscript, including the modifications and revisions described in your point-by-point letter.

Acceptance of the manuscript will entail a second round of review. EMBO Molecular Medicine encourages a single round of revision only and therefore, acceptance or rejection of the manuscript will depend on the completeness of your responses included in the next, final version of the manuscript. For this reason, and to save you from any frustrations in the end, I would strongly advise against returning an incomplete revision.

When submitting your revised manuscript, please carefully review the instructions that follow below. Failure to include requested items will delay the evaluation of your revision:

We require:

- 1) A .docx formatted version of the manuscript text (including legends for main figures, EV figures and tables). Please make sure that the changes are highlighted to be clearly visible.
- 2) Individual production quality figure files as .eps, .tif, .jpg (one file per figure). For guidance, download the 'Figure Guide PDF' (<https://www.embopress.org/page/journal/17574684/authorguide#figureformat>).
- 3) At EMBO Press we ask authors to provide source data for the main figures. Our source data coordinator will contact you to discuss which figure panels we would need source data for and will also provide you with helpful tips on how to upload and organize the files.
- 4) A .docx formatted letter INCLUDING the reviewers' reports and your detailed point-by-point responses to their comments. As part of the EMBO Press transparent editorial process, the point-by-point response is part of the Review Process File (RPF), which will be published alongside your paper.
- 5) A complete author checklist, which you can download from our author guidelines (<https://www.embopress.org/page/journal/17574684/authorguide#submissionofrevisions>). Please insert information in the checklist that is also reflected in the manuscript. The completed author checklist will also be part of the RPF.
- 6) It is mandatory to include a 'Data Availability' section after the Materials and Methods. Before submitting your revision, primary datasets produced in this study need to be deposited in an appropriate public database, and the accession numbers and database listed under 'Data Availability'. Please remember to provide a reviewer password if the datasets are not yet public (see <https://www.embopress.org/page/journal/17574684/authorguide#dataavailability>). In case you have no data that requires deposition in a public database, please state so in this section. Note that the Data Availability Section is restricted to new primary data that are part of this study.
- 7) For data quantification: please specify the name of the statistical test used to generate error bars and P values, the number (n) of independent experiments (specify technical or biological replicates) underlying each data point and the test used to calculate p-values in each figure legend. The figure legends should contain a basic description of n, P and the test applied. Graphs must include a description of the bars and the error bars (s.d., s.e.m.). Please provide exact p values.
- 8) Our journal encourages inclusion of *data citations in the reference list* to directly cite datasets that were re-used and obtained from public databases. Data citations in the article text are distinct from normal bibliographical citations and should directly link to the database records from which the data can be accessed. In the main text, data citations are formatted as follows: "Data ref: Smith et al, 2001" or "Data ref: NCBI Sequence Read Archive PRJNA342805, 2017". In the Reference list, data citations must be labeled with "[DATASET]". A data reference must provide the database name, accession

number/identifiers and a resolvable link to the landing page from which the data can be accessed at the end of the reference. Further instructions are available at .

9) We replaced Supplementary Information with Expanded View (EV) Figures and Tables that are collapsible/expandable online. A maximum of 5 EV Figures can be typeset. EV Figures should be cited as 'Figure EV1, Figure EV2' etc... in the text and their respective legends should be included in the main text after the legends of regular figures.

10) The paper explained: EMBO Molecular Medicine articles are accompanied by a summary of the articles to emphasize the major findings in the paper and their medical implications for the non-specialist reader. Please provide a draft summary of your article highlighting

11) For more information: There is space at the end of each article to list relevant web links for further consultation by our readers. Could you identify some relevant ones and provide such information as well? Some examples are patient associations, relevant databases, OMIM/proteins/genes links, author's websites, etc...

12) Author contributions: CRediT has replaced the traditional author contributions section because it offers a systematic machine readable author contributions format that allows for more effective research assessment. Please remove the Authors Contributions from the manuscript and use the free text boxes beneath each contributing author's name in our system to add specific details on the author's contribution. More information is available in our guide to authors.

13) Disclosure statement and competing interests: We updated our journal's competing interests policy in January 2022 and request authors to consider both actual and perceived competing interests. Please review the policy <https://www.embopress.org/competing-interests> and update your competing interests if necessary.

14) Every published paper now includes a 'Synopsis' to further enhance discoverability. Synopses are displayed on the journal webpage and are freely accessible to all readers. They include a short stand first (maximum of 300 characters, including space) as well as 2-5 one-sentences bullet points that summarizes the paper. Please write the bullet points to summarize the key NEW findings. They should be designed to be complementary to the abstract - i.e. not repeat the same text. We encourage inclusion of key acronyms and quantitative information (maximum of 30 words / bullet point). Please use the passive voice. Please attach these in a separate file or send them by email, we will incorporate them accordingly.

15) As part of the EMBO Publications transparent editorial process initiative (see our Editorial at <http://embomolmed.embopress.org/content/2/9/329>), EMBO Molecular Medicine will publish online a Review Process File (RPF) to accompany accepted manuscripts.

In the event of acceptance, this file will be published in conjunction with your paper and will include the anonymous referee reports, your point-by-point response and all pertinent correspondence relating to the manuscript. Let us know whether you agree with the publication of the RPF and as here, if you want to remove or not any figures from it prior to publication. Please note that the Authors checklist will be published at the end of the RPF.

I look forward to receiving your revised manuscript.

Yours sincerely,

Lise Roth

Please use this link to login to the manuscript system and submit your revision:
<https://embomolmed.msubmit.net/cgi-bin/main.plex>

***** Reviewer's comments *****

Rev_Com_number: RC-2024-02361

New_manu_number: EMM-2024-19652-T

Corr_author: Sharanek

Title: Exploiting metabolic vulnerability in glioblastoma using a brain-penetrant drug with a safe profile

Manuscript number: RC-2024-02361

Corresponding author(s): Ahmad Sharaneq, Thomas Daubon, Andreas Bikfalvi

1. General Statements [optional]

We thank the referee for their thorough evaluation and appreciation of our manuscript, noting it is well-written, organized, and that the data are clearly presented and convincing. We are grateful for their interest in our work and for finding that our study demonstrates promising potential and could have clinical translational impact in glioblastoma treatment. We appreciate the referee's constructive comments and suggestions. In the revised manuscript, we have addressed each of their points by conducting additional experiments and making the necessary revisions to the text.

2. Description of the planned revisions

Reviewer #1 (Evidence, reproducibility and clarity (Required)):

The authors report the use of mubritinib, a drug targeting complex I of the mitochondrial electron transport chain, to halt the proliferation of brain tumor stem cells (BTSCs) isolated as neurospheres from glioblastomas. They demonstrate that mubritinib crosses the blood-brain barrier, and show that this drug delays GBM tumorigenesis and extends lifespan in mouse models that were generated by transplantation of human BTSCs or mouse cell lines. They also provide evidence that this potentially harmful drug is well-tolerated by mice. The ability of mubritinib (initially conceived as a ERBB2 inhibitor) to block complex I of the electron transport chain was previously identified in acute myeloid leukemia, where this drug proved to selectively inhibit a subset of cases relying on oxidative phosphorylation (OXPHOS). However, the use of mubritinib is novel in GBM, a lethal malignancy for which a few therapeutic options are available, and no substantial progress has been reported since 2005. In addition, the authors show that accumulation of mubritinib in the brain tissue allows for the use of a reduced dose of mubritinib, thereby reducing the risk of the deleterious effects that blunt enthusiasms about targeting of mitochondrial respiration.

However, this manuscript presents significant weaknesses that are detailed below. In particular, the models used appear insufficient to support the conclusion, stated in the abstract, that the drug can be effective in GBMs with different oncogenic mutations. Based on the provided evidence, the claim that 'mubritinib potently impairs stemness and growth of patient-derived BTSCs harboring different oncogenic mutations' should be removed, or supported by using BTSCs that adequately represent the major GBM subtypes identified by genetic and transcriptional analysis. Specifically, BTSCs harboring EGFR amplification (displayed by 40% of GBMs) would need to be added. Equally, it is inappropriate to conclude (Discussion, page 14) that the models used, displaying EGFR mutations, are representative of widespread EGFR

alterations: indeed the EGFR alteration observed in 40-50% of patients is EGFR amplification, whose pathogenic effects can not be recapitulated by the EGFR mutation harbored in the models used. If models harboring EGFR amplification are unavailable to these authors, making unrealistic to repeat the experiments in at least two different EGFR-amplified BTSCs in the timeframe allowed for revision, not only the claim that mubritinib inhibits BTSCs harboring different oncogenic mutations should be removed, but lack of experiments in EGFR-amplified BTSCs should be discussed as a limitation of this study. In addition, as detailed below, in vitro experiments should be expanded to corroborate mechanistic aspects of the drug, safety should be better demonstrated and some aspects of in vivo experiments should be clarified.

Overall, the methodology seems sufficiently detailed to allow reproduction. The experiments are adequately repeated and the statistical analysis is appropriate, but in one case detailed below.

We thank the reviewer for their careful evaluation, positive feedback on our manuscript, and for the many constructive comments and suggestions. We would like to note that three of the cell lines included in the initial version of the manuscript are EGFR-amplified. We fully agree with the reviewer's recommendation to add additional cell lines to strengthen the paper. In the revised manuscript, we have included four additional cell lines (BTSC75, BTSC100, BTSC119, and BTSC198). This brings the total number of patient-derived BTSC lines in the study to 11, five of which harbor EGFR amplification. The genetic and transcriptional characteristics of these cell lines are summarized in **Table EV1** and **New Figs EV1a-b**, respectively. As suggested by the reviewer, we have removed the statement "Mubritinib has an effect on BTSCs harboring different oncogenic mutations" from the revised manuscript.

Major Point N.1

Figure 2a and 2c. Although statistically significant, the effect of mubritinib at 20 nM is biologically of limited significance in a subset of BTSCs, where the drug reduces viability by less than 25% (Fig. 2a). Therefore, the correlation between oxygen consumption rate (OCR) and mubritinib sensitivity (% of live cells), shown in Fig. 2c, should be presented for all mubritinib doses, and particularly for the dose of 500 nM, which is utilized in subsequent experiments.

Additionally, it would be useful to display the correlation not only between viability and basal OCR but also with maximal OCR. This analysis could identify varying levels of sensitivity to ECT inhibition, aligning with the expectation that different BTSCs may exhibit varying degrees of dependency on mitochondrial OXPHOS. This would suggest that different GBMs may require different dosages of the drug. In all the experiments presented in the manuscript, the authors use the lines exhibiting the highest mubritinib sensitivity (BTSC 53, BTSC73), which might not be representative of all GBMs. This selection bias need explicit clarification in the text.

As suggested by the reviewer, we performed additional analysis of the correlation between the oxygen consumption rate (OCR) and mubritinib sensitivity (% of live cells) for all mubritinib concentrations. We observed a positive correlation between the OCR and sensitivity to mubritinib at all the concentration tested (**New Figs EV2e-g**). Also, to better take into account all the concentrations of mubritinib, we determined the area under curve (AUC) in the dose-response curve for each cell line and calculated the sensitivity score. We performed correlation

analysis between the sensitivity score and OCR. This analysis confirmed the positive correlation between the OCR and sensitivity to mubritinib (**New Fig 1k**). Furthermore, as requested, we assessed the correlation between the sensitivity score and maximal OCR, which also showed a positive correlation (**New Fig EV2h**).

In response to the reviewer's concern about potential selection bias due to the use of the most sensitive cell lines (BTSC53 and BTSC73), we expanded our analysis in the revised manuscript by including additional cell lines. In addition to BTSC53 and BTSC73, we performed ELDA analysis on 8 other patient-derived BTSC lines (BTSC147, BTSC12, BTSC119, P3, BTSC25, BTSC50, BTSC75 and BTSC100) and a murine derived BTSC line (mGB2) (**New Figs 3a-f and EV3a-e**). We assessed the stemness markers on 6 patient-derived BTSC lines (BTSC53, BTSC73, BTSC147, BTSC12, BTSC119, P3) and the murine derived BTSC line, mGB2 (**New Figs 3g-l and EV3f**). These experiments validated the impact of mubritinib on BTSC maintenance using multiple BTSC models. We also confirmed the increase in ROS generation following combined treatment with mubritinib and IR using BTSC12 and BTSC147 (**New Figs EV5f-g**), in addition to BTSC73 and 53 (**New Figs 6j-m**). We also include BTSC147 and mGB2 to assess the impact of mubritinib on GB growth *in vivo* (**Figs 4d-i**). All these data are now included in the revised manuscript.

Major Point N.2.

In Fig. 2b, the statistics is significantly biased as it is calculated based on technical replicates, rather than on a significant number of independent models featuring either wild-type or mutated EGFR. Presented in this manner, this analysis is unacceptable.

Additionally, as noted previously, the models used in this manuscript do not represent the overall GBM genetics, particularly due to lack of EGFR amplified models, which correspond to 40% of cases, and cannot be recapitulated by EGFR mutations. In general, the number of models is too small to draw any conclusion regarding the relationship between genetics and mubritinib sensitivity (including conclusions concerning TP53, or the MGMT status, shown in supplementary figures 1b-c). If the authors intend to claim that GBMs are sensitive to mubritinib independently of the genetic status, they should repeat their experiments by using models representative of the major genetic/transcriptional subtypes of GBMs, by clearly characterizing and identifying them in the experiments: e.g. 'classical/EGFR-amplified'; 'mesenchymal'; 'proneural/PDGFR amplified'. Otherwise (more realistically), it is suggested to remove claims that mubritinib is effective independently of genetic alterations throughout the manuscript (including the abstract and the discussion).

In the revised version of the manuscript, we have added four additional cell lines to the study. In total, the revised manuscript includes a murine-derived BTSC and 11 patient-derived BTSC lines, five of which harbor EGFR amplification.

As advised by the reviewer, we have removed the analyses (including the previous Fig. 2b) and deleted the statement "mubritinib is effective independently of genetic alterations" throughout the manuscript.

As requested by reviewers 1 & 2, we performed analysis where we attempted to correlate the BTSC sensitivity to mubritinib to the transcriptional subtype signatures. Using RNA-seq data

from BTSCs, we analysed the gene expression profiles across different BTSCs and generated classification scores based on the Wang GB signatures (classical, proneural, and mesenchymal) (**New Figs EV1a and EV1c-e**). No statistically significant correlations emerged between sensitivity to mubritinib and the mesenchymal or the classical signature, but we observed a negative correlation with the proneural signature (**New Figs EV1e**). We performed similar analysis with the Garofano pathway-based transcriptional GB signatures (proliferative/progenitor, neuronal, mitochondrial and glycolytic/plurimetabolic) (**New Figs 1s, EV1b and EV1 f-h**). We found a positive correlation between sensitivity to mubritinib and the mitochondrial subtype signature (**New Fig 1s**), a GB transcriptional subtype that was identified to rely on OXPHOS (PMID: 33681822).

However, we clearly state in the text that “a study with a larger panel of samples will need to be conducted to conclusively establish whether sensitivity to mubritinib is linked to any specific transcriptional GB subtype.”

Major Point N.3

Fig. 2k-l present a transcriptional analysis with questionable representativeness as it is performed on the single line BTSC147 from a recurrent GBM, which is unlikely to represent primary GBMs. The analysis appears overly descriptive and fails to add significant information beyond the observation that mubritinib induces a proliferative arrest, as assessed in biological experiments. Additionally, the claim that the Neftel 'Neural Progenitor Cell' signature is altered in a biologically significant manner after only 24 hours of mubritinib treatment seems questionable. As such, this analysis should be moved to supplementary information. A more intriguing alternative would be to compare groups of BTSCs that exhibit high or low sensitivity to mubritinib and attempt to identify gene sets that can correlate with and possibly contribute to explain differences in drug sensitivity.

In the revised manuscript, we performed RNA-seq analysis following mubritinib treatment on an additional BTSC line (BTSC73), which is derived from primary glioblastoma. We conducted GSEA analysis on both BTSC73 and BTSC147 and found that genes related to the cell cycle were among the top downregulated pathways in both cell lines (**New Figs. 2a-c and Appendix fig S1a-e**). This RNA-seq analysis highlighted the effect of mubritinib on the cell cycle and led to the identification of the AMPK/CyclinD1/p27^{kip1} axis (**New Figs 2f-j and Appendix fig S1g**). In addition, analysis of RNA-seq data revealed that mubritinib significantly downregulates DNA repair pathways (**New Figs EV4h-i**), providing rationale to assess the enhanced DNA damage in response to TMZ and IR. We have included these RNA-seq data in Fig 2 and, EV5 and Appendix fig S1, while the remaining RNA-seq results for mubritinib-treated BTSC73 and BTSC147 are now included in the supplementary data (**Dataset EV1-4**) as requested by the reviewer.

As suggested by the reviewer, we also performed additional analysis comparing gene signatures between high and low mubritinib-sensitive BTSCs. We found a significant correlation between the mitochondrial GB signature (PMID: 33681822) and sensitivity to mubritinib (**New Fig 1s**). Specifically, the BTSCs that were highly sensitive to mubritinib showed higher

expression of the mitochondrial signature. This GB subtype has been shown to rely exclusively on OXPHOS for energy production and to exhibit marked vulnerability to OXPHOS inhibitors (PMID: 33681822).

Major point N. 4

Figure 3. LDA need to be measured at longer timepoints (14-21 days vs. 7 days shown).

As requested by the reviewer, all ELDA experiments were repeated and measured at a later time point (21 days). We also performed ELDA analysis at 21 days on an additional set of patients derived-BTSCs, bringing the total number of BTSC lines analyzed to 11. The new data are now included in the revised manuscript (**New Figs 3a-f and EV3a-d**).

Major point N. 5

Supplementary Figure 3c aims to demonstrate that neural stem cells are unaffected by mubritinib merely by showing stem markers in western blots, which is insufficient. To provide convincing evidence, an LDA should be performed using human Neural Progenitor Cells.

As requested by the reviewer, in addition to the analysis of stem cell markers by Western blot (**Fig 3p**), we performed ELDA at 21 days on human neural progenitor cells (hNPCs) following treatment with mubritinib. Interestingly, we found that, unlike the BTSCs, hNPCs were not affected by mubritinib treatment (**New Fig 3o**). This may be explained by the distinct metabolic profile of hNPCs, which rely on glycolysis rather than OXPHOS (PMID: 27282387, PMID: 28334613, PMID: 27237737). In support, we found that, similar to mubritinib, the OXPHOS inhibitor rotenone had no effect on the number of live hNPCs, while 2-Deoxy-D-glucose (2-DG), a competitive glycolysis inhibitor, significantly reduced the number of live hNPCs (**New Appendix fig S2d**). These data are included in the revised version.

Major Point N. 6

Page 9. The mechanistic nexus between OXPHOS inhibition and radiosensitization described by the authors remains unclear. In particular, the link between enrichment in OXPHOS proteins observed in recurrent vs. primary GBMs on the one hand, and downregulation of homologous recombination related-pathways on the other hand is difficult to grasp. The authors should endeavor to more clearly explain how mubritinib can interfere with the adaptive response to ionizing radiation, thereby providing a rationale for experiments combining the two treatments. As noted in the discussion (page 14), 'targeting mitochondrial respiration is an emerging strategy to overcome radioresistance in the tumour hypoxic areas'. Thus, a plausible mechanism of mubritinib-induced radiosensitization may involve reducing oxygen consumption, thereby leaving more oxygen available for diffusion and improving radiation response (by increasing generation of reactive oxygen species). To provide convincing mechanistic evidence, the authors should include in vitro experiments assessing the ability of mubritinib to radiosensitize BTSC in both normoxic and hypoxic conditions. LDA or radiobiological clonogenic assays showing the effect of combination treatment on stem cell frequency are recommended.

In the revised manuscript, we performed both *in vitro* and *in vivo* studies to better explore the link between the inhibition of OXPBOS and radiosensitization. We tested the hypothesis that mubritinib alleviates tumour hypoxia and enhances oxidative stress to sensitize GB tumours to IR. By inhibiting OXPBOS, mubritinib reduces oxygen consumption within the tumour, allowing more oxygen to diffuse into hypoxic regions. This, in turn, enhances ROS production and improves the tumour's response to IR (PMID: 30201710; PMID: 30201710).

To begin, we investigated whether mubritinib reduces hypoxia and increases oxygen availability in BTSC tumourspheres using a hypoxia probe *in vitro*. In the first experimental setup, BTSCs were labeled with the hypoxia probe under normoxic conditions (21% O₂), which highlighted the hypoxic core of the tumoursphere (**New Figs 6a, 6c, and EV5a**). Mubritinib treatment significantly reduced the fluorescent labeling (**New Figs 6a-d and EV5a-e**), indicating increased oxygen diffusion within the tumoursphere.

To better model tissue hypoxia, we next placed BTSC tumourspheres in a hypoxia chamber (1% O₂). Under these conditions, we observed a marked increase in fluorescence labeling throughout the tumoursphere compared to normoxia (**New Figs 6a-d and EV5b-e**). Mubritinib treatment again reduced hypoxia probe labeling, suggesting greater oxygen availability (**New Figs 6a-d and EV5b-e**). We further confirmed these effects using a GFP hypoxia reporter construct, which showed reduced hypoxia and increased oxygen availability *in vitro* upon mubritinib treatment (**New Figs 6e-g**).

Next, to assess whether mubritinib reduces hypoxia in GB tumours *in vivo*, we utilized the GFP hypoxia reporter and employed intracranial tumour assays in which BTSC73 cells were transduced with the hypoxia GFP-reporter and implanted into the brains of mice. Then, mice received either vehicle control or mubritinib. At the end of the assay, the brains from both groups were subjected to GFP staining, as an indicator of hypoxia (**New Fig 6h**). Interestingly, we found that mubritinib was also able to alleviate hypoxia in GB tumours *in vivo* (**New Fig 6i**), suggesting more oxygen availability for diffusion in GB tumours.

We then examined whether the increased oxygen availability with mubritinib treatment results in enhanced ROS generation when combined IR. We used the H₂DCFDA and MitoSOX Red probes to detect cellular ROS and mitochondrial superoxides, respectively. We found that combined treatment with IR and mubritinib induced significantly greater ROS generation compared to either treatment alone in multiple patient-derived BTSCs *in vitro* (**New Figs 6j-m and EV5f-g**). To assess ROS generation in GB tumours *in vivo*, we subjected GB tumour xenografts from animals treated with mubritinib alone, IR alone, or the combination of both to staining with the OxyIHC oxidative stress detection kit. Our results revealed significantly higher ROS levels in GB tumours receiving combined IR and mubritinib treatment compared to either monotherapy (**New Fig 6n**).

Additionally, as suggested by the reviewer, we assessed ROS generation and radiosensitization effects under hypoxic conditions (1% O₂). Combined mubritinib and IR treatment significantly increased ROS generation in BTSCs compared to either treatment alone under hypoxia (**New Figs EV5h-i**). Importantly, ELDA analysis revealed that mubritinib was able to sensitize BTSCs to IR under hypoxic conditions (**New Fig EV5j**). Altogether, these data indicate that OXPHOS inhibition by mubritinib alleviates hypoxia, increases oxygen availability, and enhances ROS generation, ultimately improving the response of BTSCs and GB tumours to IR.

As recommended by the reviewer, we performed ELDA experiments to assess the effect of combining IR and mubritinib on stem cell frequency. Our assays revealed that the combination treatment reduced stem cell frequency more effectively than IR or mubritinib alone. Interestingly, ELDA analysis showed that mubritinib radiosensitizes BTSCs in both normoxic (**New Figs 5a-b**) and hypoxic conditions (**New Fig EV5j**). These data are now included in the revised version.

Major Point N. 7

Fig. 5d. Concerning the scheme of *in vivo* treatment, it is unclear why irradiation is administered 5 days after the beginning of mubritinib treatment, considering that mubritinib reaches its peak brain concentration much earlier, as shown in Fig. 4d.

Furthermore, if mubritinib alone is effective against the tumor, comparing tumors that have been treated with IR alone at the same time-point as those treated with IR + mubritinib seems inappropriate. This is because, in the latter scenario, IR is applied to tumors that are likely reduced in volume compared to those treated with vehicle prior to IR. This discrepancy could introduce bias in the evaluation of the combined treatment's efficacy.

To address this comment, we repeated the *in vivo* tumour assay and we changed the treatment scenario in which we started the irradiation only 24 h after the start of mubritinib treatment (at the concentration peak of mubritinib in the brain), as suggested by the reviewer. We found that, similar to the previous treatment scenario (where irradiation began 5 days after starting mubritinib), a 24-hour pre-treatment with mubritinib was sufficient to sensitize GB tumours to IR. Regardless of the treatment protocol, the combination of IR and mubritinib showed a benefit in delaying GB tumours and extending the lifespan of the animals. These data have been included in the revised manuscript (**New Figs EV4c-g**).

Major Point N. 8.

Figure 6b. The methodology employed to measure the effect of mubritinib on human neural progenitor cells should be the same as that used for BTSC. Moreover, a positive control (a treatment inducing death, such as bosentan for hepatocytes) needs to be added.

As suggested by the reviewer, in the revised version we employed the annexin V/PI double staining followed by flow cytometry (the same methodology used for BTSCs), to evaluate the impact of mubritinib on human neural progenitor cells (hNPCs). Interestingly, we compared the side effects of mubritinib with that of TMZ on hNPCs. While mubritinib has no impact on the cell

viability, TMZ was highly toxic to these normal cells (**New Figs 8b-d**). As recommended, we also included Staurosporine A as a positive control (**New Figs 8c-d**). These results are now included in the figures and results section of the revised manuscript.

Minor points

Minor point N.1

Please use consistent units (nM or uM) for mubritinib.

nM is now used throughout the manuscript.

Minor point N.2

ND1 expression in transduced cells should be shown by western blot (in Supplementary Figures).

Antibodies against NDI1 are not routinely used; they can be obtained from only two suppliers but require a lead time of approximately 22 weeks. As an alternative approach to validate ectopic expression of NDI1, we used RT-qPCR to measure NDI1 mRNA levels (**New Figs EV2i-k**).

Reviewer #1 (Significance (Required)):

SIGNIFICANCE

This manuscript reports preclinical results on the use of mubritinib, a drug targeting the mitochondrial electron chain transport complex 1, to halt proliferation of glioblastoma (GBM) stem cells in vitro and treat experimental GBMs generated by stem cell transplantation. Given that the standard therapy for GBM still relies on a limited number of conventional options, evidence demonstrating the preclinical effectiveness of mubritinib could exert a significant translational impact. Mubritinib has not yet been proposed for GBM treatment, but there is convincing evidence that the drug may be effective in subsets of acute myeloid leukemia that rely on oxidative phosphorylation (Baccelli et al., Cancer Cell 36:84, 2019. PMID: 31287994). Data provided in this manuscript on potential effectiveness of mubritinib are overall convincing. However, in its current form, the manuscript present major limitations regarding the representativeness of models used, which do not support the claim that mubritinib could be universally useful in GBM, and regarding mechanistic aspects of mubritinib combination with radiotherapy. These and other aspects could be addressed through additional experiments. The audience interested in the reported findings includes preclinical and clinical neuro-oncologists.

My field of expertise is biology and genetics of glioblastoma stem cells and generation of in vitro and in vivo GBM preclinical models.

Reviewer #2 (Evidence, reproducibility and clarity (Required)):

In this manuscript Burban et al explore the effect of the mitochondrial oxidative phosphorylation (OXPHOS) inhibitor Mubritinib on patient derived glioblastoma stem cells and murine

xenografts. The authors first show that i) Mubritinib is an inhibitor of OXPHOS in brain tumor stem cells (BTSC), ii) that it impairs cell growth and self-renewal of patient-derived BTSC with different genetic background and iii) has an effect on the expression of genes related to stemness. In addition, the authors convincingly show that Mubritinib is a brain penetrant drug, and by transplanting luciferase expressing BTSC into the brain of immunodeficient mice they show it delays GB tumorigenesis and the animal lifespan (either alone, or more efficiently if combined with IR treatment). Finally, by performing toxicological and behavioral studies in mice models, Burban et al demonstrate that Mubritinib has a well-tolerated and safe profile and does not induce damage to healthy cells.

The manuscript is well written and organized and the data is clearly presented. The results are convincing, but a few additional experiments and controls would be beneficial to support the claims of the paper, most of which are easily addressable.

We would like to thank the reviewer for their positive feedback, particularly for the comments stating that “The manuscript is well written and organized,” “The data are clearly presented,” and “The results are convincing.” In the revised manuscript, we have addressed each of the remaining concerns, as outlined below.

Main comments:

- Finding suitable control cells for BTSC experiments is a widely acknowledged challenge in the field. However, in line with other studies, it is recommended that the authors consider using a non-oncogenic NSC as control line to demonstrate that the effects reported in Figure 1 and Figure 2 are more pronounced in BTSC compared to NSC (as it was done in Suppl Fig3).

As requested by the reviewer, we performed similar experiments on non-oncogenic human neural progenitor cells (hNPCs) as those performed for BTSCs.

First, hNPCs were treated with mubritinib and subjected to ELDA analysis. Unlike BTSCs, we found that mubritinib treatment did not affect the self-renewal or stem cell frequency (SCF) in hNPCs (**New Fig 3o**).

Second, hNPCs were subjected to Western blot analysis to assess the levels of stemness markers (Cleaved Notch1, Olig2, and SOX2), and we observed that mubritinib treatment had no effect on these markers (**New Fig 3p**).

Third, hNPCs were subjected to cell viability assay following mubritinib treatment, and found that unlike BTSCs, hNPCs were not affected by mubritinib treatment (**New Figs 8b-d and Appendix fig S2a-c**). Interestingly, we compared the effect of mubritinib to that of TMZ on the normal hNPCs. While mubritinib, at concentration where it is effective on BTSCs, showed no effects on hNPCs, TMZ was highly toxic to hNPCs (**New Figs 8b-d**). These results suggest that, unlike TMZ, mubritinib selectively targets BTSCs without affecting non-oncogenic hNPCs.

Fourth, we assessed the impact of mubritinib on normal mouse NPCs residing in the lateral subventricular zone (SVZ) *in vivo*. We found that mubritinib had no apparent effect on the number of SOX2-positive NPCs (**Fig 8h**). Additionally, immunostaining for cleaved caspase-3 (CC3) in the brains of mubritinib-treated mice showed no increase in apoptosis in NPCs

compared to vehicle-treated mice (**Appendix fig S2e**). Taken together, these *in vitro* and *in vivo* data demonstrate that mubritinib selectively impairs BTSCs without causing significant damage to healthy cells.

This differential effect could be explained by the distinct metabolic profile of hNPCs, which primarily rely on glycolysis rather than OXPHOS (PMID: 27282387, PMID: 28334613, PMID: 27237737). In support of this, we found that, similar to mubritinib, the OXPHOS inhibitor rotenone had no effect on hNPC viability, while 2-Deoxy-D-glucose (2-DG), a competitive glycolysis inhibitor, significantly reduced the number of live hNPCs (**New Appendix fig S2d**). These data are now included in the revised manuscript.

- Figure 5 presents a significant finding indicating that Mubritinib enhances the sensitivity of GB tumors to IR. Considering that Temozolomide (TMZ) is the primary chemotherapy drug for GB patients, it would be crucial to investigate the potential outcomes of a combined treatment involving Mubritinib and TMZ. This will help determine if the combination exhibits promising results, comparable to what is demonstrated in Figures 5d, 5g, and 5i for Mubritinib and IR. Such experiment would reinforce the drug's potential for clinical trials in GB treatment.

We thank the reviewer for this suggestion. In the revised manuscript, we performed both *in vivo* and *in vitro* experiments involving combination treatment with mubritinib and TMZ. Our data revealed a significant advantage of combining TMZ and mubritinib in inhibiting BTSCs *in vitro* (**New Figs 7a-b**) and GB tumours *in vivo* (**New Figs 7d-k**). Additionally, *in vivo* survival assays showed that the combination of TMZ and mubritinib significantly extended animal lifespan compared to either TMZ or mubritinib monotherapy (**New Figs 7f-g and 7j-k**).

We also assessed DNA damage in BTSCs following treatment with TMZ, mubritinib, or the combination of both using γ -H2AX immunofluorescence staining. The combined treatment led to a significant increase in the number of γ -H2AX foci per nucleus, indicating higher DNA damage compared to either treatment alone (**New Figs 7c**). These data, along with the radiosensitization effect of mubritinib, underscore its translational potential for glioblastoma treatment in combination with the current standard of care.

Minor comments:

- The authors should specify early in the paper what is the number of samples of patient-derived BTSC they use and the fact that their genetic mutations are known (this information is summarized in the supplemental table 1, but only reported later in the manuscript). This information is important and should be clearly stated at the beginning of the manuscript.

This information has now been added at the beginning of the results section, with their mutational characteristics presented in **Table EV1** and their transcriptional profiles shown in **Fig EV1a-b**.

- In Figure 2a the inhibition at 20nM is significant but not very pronounced. Based on Figure 1 I

would have expected to see a stronger effect at this concentration range. Can the authors comment/provide an explanation for this discrepancy?

In the revised version, we analyzed the number of live cells after 7 days of mubritinib treatment, instead of 4 days. In sensitive BTSCs, a 20 nM mubritinib treatment inhibited proliferation by about 50% (**New Fig 1f**). Additionally, the EdU assay showed that at 20 nM, mubritinib significantly reduced EdU incorporation by around 50% (**New Fig 1g-j**).

- The EdU incorporation experiment presented in Figure2h-l should be repeat with lower concentrations of the drug (in most of the assays the effect of Mubritinib is detectable at much lower concentrations).

As suggested by the reviewer, the EdU experiments were also performed at lower concentrations of mubritinib (i.e., 20 nM and 100 nM) and included in the revised version of the manuscript (**New Fig 1g-j**).

- Since the authors have done RNA-seq on the samples why don't they report the specific subtypes of their samples in the text and in Suppl Table 1 (Proneural, Neural, Classical or Mesenchymal) ? It is known that different molecular subtypes respond differently to treatments; therefore this information would be essential to understand if Mubritinib is effective on a wide range of GB subtypes.

As requested by reviewers 1 & 2, we performed analysis where we attempted to correlate the BTSC sensitivity to mubritinib to the transcriptional subtype signatures. Using RNA-seq data from BTSCs, we analysed the gene expression profiles across different BTSCs and generated classification scores based on the Wang GB signatures (classical, proneural, and mesenchymal) (**New Figs EV1a and EV1c-e**). No statistically significant correlations emerged between sensitivity to mubritinib and the mesenchymal or the classical signature, but we observed a negative correlation with the proneural signature (**New Figs EV1e**). We performed similar analysis with the Garofano pathway-based transcriptional GB signatures (proliferative/progenitor, neuronal, mitochondrial and glycolytic/plurimetabolic) (**New Figs 1s, EV1b and EV1 f-h**). We found a positive correlation between sensitivity to mubritinib and the mitochondrial subtype signature (**New Fig 1s**), a GB transcriptional subtype that was identified to rely on OXPHOS (PMID: 33681822).

However, we clearly state in the text that “a study with a larger panel of samples will need to be conducted to conclusively establish whether sensitivity to mubritinib is linked to any specific transcriptional GB subtype.”

- In Figure2b and Supplemental Figure1b-c : instead of correlating the effect with genetic mutations, it would be more relevant if the authors could correlate the data with the molecular subtypes inferred by RNA-seq (see my comment above)

As mentioned in the previous comment, we performed this analysis and added it to the revised version of the manuscript.

- Regarding the RNA-seq experiment the authors should report what is the percentage (and numbers) of genes that change expression. Is there for example a preference for up- or down-regulation? It would be interesting to see a Gene Ontology (GO) analysis for the up-regulated genes versus a GO analysis of the down-regulated genes to confirm that the relevant categories show dysregulation as expected (e.g. enrichment for cell cycle and stemness genes in the down-regulated list, etc).

In the revised version, we included tables with all the genes that are significantly downregulated and significantly upregulated in BTSC73 and BTSC147 (**Dataset EV1-2**). We also performed functional analysis of the downregulated and upregulated pathways by KEGG and Reactome analysis of the significantly upregulated and downregulated genes in BTSC73 and BTSC147 (**Dataset EV3-4**). The raw data are now deposited in the GEO under the number GSE253362.

- In Figure2 m-o the difference between CTL and Mubritinib treated cells do not seem substantial, although it is shown as statistically relevant. Can the authors specify the percentage to be able to better assess the differences?

As requested by the reviewer, the % are now added (**Figs 2 d-e and Appendix fig S1f** in the revised version).

- Add p-value for Figure3a-c

As requested, the p-values are added for all ELDA analysis (**New Figs 3a-f, 3m, 3o, EV3a-e, EV3g, 5a-b, and EV5j**).

- In the western blot in Supplemental Figure 3b Vinculin shows twice

These are two separate vinculin blots (loading controls) for different gels. The upper vinculin blot is the loading control for the Olig2 blot, while the lower vinculin blot served is the loading control for the SOX2 blot. The uncropped blots are shown below:

- Change" Given that Mubritinib is already completed a phase I clinical trial" into "...has completed..."

As requested, “is” was replaced by “has” in the revised version.

****Referees cross-commenting****

I agree with other reviewers that more data is needed to determine if mubritinib could be an effective treatment for various GB subtypes. The models used in this study do not encompass the full spectrum of GBM genetics. The authors should repeat the experiments using models that represent the major genetic/transcriptional subtypes of GBMs and clearly label and identify them in the study. Specifically, the authors should include models like 'classical/EGFR-amplified', 'mesenchymal', 'proneural/PDGFR amplified'. Alternatively, it is advisable to refrain from asserting that mubritinib is effective across genetic alterations in the manuscript.

Please see our response to the comment raised by the first reviewer:

In the revised version of the manuscript, we have added four additional cell lines to the study. In total, the revised manuscript includes a murine-derived BTSC and 11 patient-derived BTSC lines, five of which harbor EGFR amplification.

As advised by the reviewer, we have removed the analyses and deleted the statement “mubritinib is effective independently of genetic alterations” throughout the manuscript”.

As requested by reviewers 1 & 2, we performed analysis where we attempted to correlate the BTSC sensitivity to mubritinib to the transcriptional subtype signatures. Using RNA-seq data from BTSCs, we analysed the gene expression profiles across different BTSCs and generated classification scores based on the Wang GB signatures (classical, proneural, and mesenchymal) (**New Figs EV1a and EV1c-e**). No statistically significant correlations emerged between sensitivity to mubritinib and the mesenchymal or the classical signature, but we observed a negative correlation with the proneural signature (**New Figs EV1e**). We performed similar analysis with the Garofano pathway-based transcriptional GB signatures (proliferative/progenitor, neuronal, mitochondrial and glycolytic/plurimetabolic) (**New Figs 1s, EV1b and EV1 f-h**). We found a positive correlation between sensitivity to mubritinib and the mitochondrial subtype signature (**New Fig 1s**), a GB transcriptional subtype that was identified to rely on OXPHOS (PMID: 33681822).

However, we clearly state in the text that “a study with a larger panel of samples will need to be conducted to conclusively establish whether sensitivity to mubritinib is linked to any specific transcriptional GB subtype.”

Reviewer #2 (Significance (Required)):

Considering the limited effectiveness of existing treatments, it is crucial to explore alternative approaches to improve patient outcomes. This study demonstrates promising potential for the clinical translation of Mubritinib in GB treatment.

A major limitation of this study is the narrow numbers of patient-derived samples used and absence of a proper control cell line. Unfortunately, as evidenced by the existing literature in the field, selecting a control cell line for glioma stem cells research is challenging due to the unknown cell-of-origin for this type of tumor. In addition, all the toxicity/safety tests were performed in mice models and it is difficult to predict how this would translate into human patients. However, the fact that a phase I clinical trial has already been completed for Mubritinib (in the context of a different type of tumor) is encouraging.

Reviewer #3 (Evidence, reproducibility and clarity (Required)):

This manuscript has potential interest as a preclinical study for glioblastoma treatment. There is a substantial amount of data that is promising, but there are numerous issues that will require additional experimental effort before publication.

We would like to thank the reviewer for their insightful comments and for acknowledging that our study has “potential interest as a preclinical study for glioblastoma treatment” and that “there is a substantial amount of data that is promising.” In the revised manuscript, we have addressed each of the reviewer’s comments by performing new experiments, analyses, and/or making modifications to the text.

Major concerns:

1. The title is overly general and uninformative. The authors should include the drug name (mubritinib) and the specific tumour type (glioblastoma).

We have included “glioblastoma” and “mubritinib” in the running title as follow:

Mubritinib impairs glioblastoma tumours

We also included “glioblastoma” in the title as follow:

Exploiting metabolic vulnerability in glioblastoma using a brain-penetrant drug with a safe profile

2. The concept that mubritinib functions through metabolic effects is not surprising given recent publications (PMID: 37382244; PMID: 35429141; PMID: 33245718; PMID: 31287994) and that it impacts the blood-brain barrier (PMID: 36178590). However, there are limitations to the strength of this observation. Most of the experiments are associations between drug treatment and metabolic changes. The NDI1 partially rescue experiments in Figures 1h and 1i are nice but show that NDI1 expression itself increases OCR. This experiment is also performed over a very brief window of time. A better set of experiments would include measurement of cell number over a prolonged time course (Figure 2d has one time point) and to use a genetic targeting strategy against ETC complex 1.

We thank the reviewer for sharing these papers, some of which are cited in the manuscript. Our work is the first to show that mubritinib could be a promising strategy for targeting glioblastoma, a lethal cancer with currently limited therapeutic options. We provide a body of data demonstrating that mubritinib alters mitochondrial respiration, impairs glioblastoma stem cell growth, inhibits glioblastoma progression, and enhances the response to IR and TMZ.

In the revised manuscript, we performed additional experiments to support this conclusion. After efficient antibiotic selection of the BTSCs transduced with the CTL or NDI1 vector, we validated the ectopic expression of the NDI1 gene in BTSCs by RT-qPCR (**New Figs EV2i-k**). In the three BTSCs tested, we found that NDI1 almost completely rescues the inhibition of OXPHOS (**New Figs 1m-o and EV2l-n**) and recovers the proliferation inhibited by mubritinib after 7 days of treatment (**New Figs 1p-r**).

As suggested by the reviewer, we measured cell numbers over a prolonged time course in the rescue experiments using the CellTrace™ CFSE flow cytometry assay and generated population growth curve. As shown (**New Fig EV1o**), ectopic expression of NDI1 fully restored the proliferation inhibited by mubritinib.

We also assessed the link between OXPHOS inhibition by mubritinib and the impairment of BTSC stemness. First, we conducted ELDA analysis and found that ectopic expression of NDI1 almost completely restored the self-renewal capacity and stem cell frequency (SCF) in mubritinib-treated cells (**New Figs 3m and EV3g**). Western blot analysis showed that the decrease in stem cell markers following mubritinib treatment was rescued by the ectopic expression of the NDI1 gene (**New Fig 3n and EV3h**).

Importantly, we performed rescue experiments in an *in vivo* setting, in which CTL or NDI1-expressing BTSC73 cells were implanted intracranially into animals and subjected to mubritinib treatment. We generated Kaplan-Meier survival plots and found that ectopic expression of NDI1 nullified the impact of mubritinib on the lifespan of animals bearing GB xenografts (**New Fig 4n**). As requested, we also employed a genetic approach by silencing NDUFS7 using siRNA (**New Fig EV2p-q**). We found that KD of NDUFS7 phenocopied the inhibition of proliferation seen with mubritinib. Importantly, treating the NDUFS7 KD cells with mubritinib did not induce significant additional effects, suggesting that in the absence of complex I activity, mubritinib does not exert additional effects on BTSCs through other mechanisms (**New Fig EV2r-s**).

Altogether, these data provide evidence that mubritinib acts on complex I of the ETC to inhibit BTSCs *in vitro* and GB tumours *in vivo*.

3. The authors observe that EGFR expressing lines are more sensitive to mubritinib. As the rescue experiments are only partially effective and ERBB family members may be targeted by mubritinib, it is critical to address the effects of mubritinib on EGFR activation and perform rescue studies, as the application of mubritinib in patients may be guided by the EGFR mutational state.

We performed additional experiments to assess a possible role of EGFR in mubritinib-induced effects. We employed a genetic approach by silencing EGFR using siRNA. We then assessed the impact of mubritinib on BTSCs in the absence of EGFR. We found that, in the absence of

EGFR, mubritinib's effect on BTSCs was unchanged, suggesting that EGFR does not play a role in the observed effects of mubritinib. See figure below:

4. The differences found with NSC responses is interesting but needs to be developed. Why are there differences in mubritinib responses? For example, do NSCs not require ETC complex 1 as much?

The differences between NSC and BTSCs could be explained by difference in metabolic dependencies. To further investigate the metabolic reliance of NSCs, we compared their response to 2-deoxy-D-glucose, a glycolysis inhibitor, and Rotenone another ETC inhibitor. We found that similar to mubritinib, NSCs were not sensitive to Rotenone but they were highly sensitive to the competitive inhibitor of glycolysis 2-deoxy-D-glucose (2-DG) (**New Appendix fig S2d**). This suggests that NSCs rely on glycolysis instead of OXPHOS. In support, previous studies have shown that NSCs have a predominant glycolytic profile (PMID: 27282387, PMID: 28334613, PMID: 27237737). These data are included in the revised version.

5. I would suggest that the authors also compare sensitivity of the BTSCs (I would suggest a change in nomenclature as these are only from GB) and differentiated tumour cells to determine if the stem cells have greater dependence. Please use similar culture conditions.

As requested by the reviewer, we performed experiments where we assessed the effects of differentiated GB cells. To differentiate BTSCs, the cells were cultured in the presence of serum for 14 days. The stemness and differentiation markers were then analysed by Western blotting to validate the differentiation of BTSCs and their differentiated progeny (**New Fig EV2t**). To evaluate the sensitivity of differentiated cells to mubritinib, we treated them with the mubritinib in the same media as the BTSCs. Our data revealed, that BTSCs are significantly more sensitive to mubritinib than their differentiated counterparts (**New Figs EV2u-x**). In support to our data, previous studies reported that differentiated GB cells have a glycolytic metabolic profile and are less dependent on OXPHOS than glioblastoma stem cells (PMID: 21900605).

Regarding the reviewer's suggestion to change the nomenclature, we believe it is preferable to retain the original nomenclature for BTSCs in order to maintain consistency with previous studies, in which we and others have used the same cell lines.

6. The differences in cell cycle are useful but the mechanism is lacking. The claim that self-renewal is drastically or markedly changed is overstated. The ELDA's are not striking. There is no evidence that stemness is a direct target.

We thank the reviewer for spotting this out. In the revised manuscript we performed additional experiments to investigate the molecular mechanisms of BTSC cell cycle impairment by mubritinib.

We subjected two patient-derived BTSC lines (#73 and #147) to mRNA-seq analysis. Gene set enrichment analysis revealed downregulation of cell cycle-related pathways in both BTSC lines (**New Figs 2a-c and Appendix Fig S1a-e**). Notably, pathways associated with the AMPK/p27^{Kip1} signaling axis, such as SCF SKP2-mediated degradation of p27/p21, cyclin D1-related events in G1, and E2F-mediated regulation of DNA replication, were downregulated in mubritinib-treated BTSC73 (**New Figs 2a-c**) and BTSC147 (**New Appendix Figs S1a-e**). AMPK, a metabolic sensor activated by metabolic stress, can initiate cell cycle arrest via the kinase inhibitor protein p27^{Kip1} (PMID: 30717766). Since OXPHOS inhibitors have been reported to induce metabolic stress by disrupting the ATP/ADP ratio and activating AMPK (PMID: 20519126), we investigated whether mubritinib affects the AMPK/p27^{Kip1} pathway.

We first assessed AMPK phosphorylation by Western blotting and found that mubritinib significantly increased AMPK phosphorylation within 24 hours of treatment (**New Fig 2f**), with this effect persisting at 72 hours (**New Appendix fig S1g**). Downstream, AMPK regulates cyclin D1, which binds and sequesters the kinase inhibitor p27^{Kip1} (PMID: 19046439). Mubritinib treatment caused a reduction in cyclin D1 protein levels after 24 hours, with further decreases observed at 48 and 72 hours (**New Fig 2g**), suggesting release and nuclear localization of p27^{Kip1}.

We examined p27^{Kip1} levels in the nuclear fractions and observed increased nuclear p27^{Kip1} protein levels in mubritinib-treated BTSCs (**New Fig 2h**). p27^{Kip1} inhibits cyclin E/CDK2 kinase activity, keeping the retinoblastoma (Rb) protein unphosphorylated. When Rb is unphosphorylated, the E2F transcription factor is inhibited from transcribing genes necessary to enter the cell cycle (PMID: 10783254). Analysis of Rb phosphorylation revealed a marked decrease following mubritinib treatment (**New Fig 2i**).

Together, these findings indicate that mubritinib disrupts the AMPK/cyclin D1/p27^{Kip1}/Rb signaling cascade and impairs the cell cycle in BTSCs (**New Fig 2j**). All these data are included in **New Figure 2** and **New Appendix fig S1** of the revised manuscript.

We performed ELDA analysis in multiple patient derived BTSC (10 cell lines) and recorded the stem cell frequency after 21 days (as suggested by reviewer 1). ELDA analysis revealed 50-70% decrease in stem cell frequency (SCF) following treatment with mubritinib. Furthermore, western blot analysis of stemness markers in 6 patient-derived BTSCs and a murine glioblastoma stem cell line showed significant downregulation in protein levels of stemness markers following mubritinib treatment.

We also assessed whether mubritinib impairs BTSCs stemness through OXPHOS inhibition. We conducted ELDA analysis and found that ectopic expression of NDI1 restores almost completely, the self-renewal capacity and SCF in mubritinib treated cells (**New Figs 3m and EV3g**). Western blot analysis showed that the decrease in stem cell markers following mubritinib treatment was almost completely rescued by the ectopic expression of NDI1 gene (**New Figs 3n and EV3h**). This suggests that the impact of mubritinib on BTSCs stemness is due to inhibition of OXPHOS.

7. The *in vivo* effects on cell biology need greater analysis in mechanism. I am also not sure why the authors switch lines tested in different assays.

We conducted additional *in vivo* analyses to investigate how mubritinib affects GB tumours. Specifically, we performed Ki67 and cleaved caspase 3 (CC3) staining to evaluate its impact on proliferation and cell death, respectively. We observed a significant reduction in Ki67 levels in mubritinib-treated GB tumours compared to the vehicle control, indicating a decrease in tumour cell proliferation (**New Fig 4m**). However, CC3 staining revealed similar levels of cell death between the mubritinib and vehicle groups (**New Fig 5o**), suggesting that, like *in vitro* findings, mubritinib alone, primarily inhibits proliferation rather than inducing cell death in GB tumours *in vivo*.

We also further investigate the mechanism by which mubritinib enhances GB tumour sensitivity to IR (see our response, below, to point 8 for details).

Regarding the reviewer's comment on switching cell lines across assays, we would like to clarify that BTSC73 and BTSC53 were consistently used in all main experiments. Other cell lines were included to further validate key findings. For example, ELDA analysis was conducted on eight patient-derived BTSCs, in addition to BTSC73 and BTSC53, and the murine GB model mGB2 (**New Fig 3a-l and EV3a-f**). Additionally, stemness markers were analyzed in 7 cell lines, including BTSC73 and BTSC53. In the *in vivo* studies, while BTSC73 and BTSC53 were the primary cell lines, BTSC147 and the syngeneic GB model mGB2 were also included to strengthen our conclusions.

8. The mechanism of interaction with radiation is not developed. What is happening here? Are there changes in DNA damage repair or simply growth? This is a nice observation that could be better developed.

This point is also raised by reviewer 1.

In this revised manuscript, we conducted both *in vitro* and *in vivo* studies to further investigate the link between OXPHOS inhibition and radiosensitization. We hypothesized that mubritinib alleviates tumour hypoxia and increases oxidative stress, thereby sensitizing GB tumours to IR. Specifically, we tested the hypothesis that by inhibiting OXPHOS, mubritinib reduces oxygen consumption within the tumour, allowing more oxygen to reach hypoxic regions, which in turn

promotes ROS production and enhances the tumour's response to IR (PMID: 30201710; PMID: 30201710).

To start, we assessed whether mubritinib reduces hypoxia, thereby increasing oxygen availability in BTSC tumourspheres using a hypoxia probe *in vitro*. In our initial experimental setup, we labeled BTSCs with the hypoxia probe under normoxic conditions (21% O₂). We observed that the probe specifically marked the hypoxic core of the tumoursphere (**New Figs 6a, 6c, and EV5a**). Notably, mubritinib treatment significantly reduced fluorescence labeling (**New Figs 6a-d and EV5a-e**), indicating that mubritinib increases the oxygen available for diffusion within the tumoursphere.

To better model tissue hypoxia, we placed BTSC tumourspheres in a hypoxia chamber (1% O₂). Under these hypoxic conditions, fluorescence labeling increased throughout the tumoursphere compared to normoxic conditions (**New Figs 6a-d and EV5b-e**). We found that mubritinib reduced hypoxia probe labeling even in hypoxic condition, indicating that it improved oxygen availability within the tumoursphere (**New Figs 6a-d and EV5a-e**). We further validated the effects of mubritinib in reducing hypoxia and increasing oxygen availability *in vitro* using a GFP hypoxia reporter construct (**New Figs 6e-g**).

To evaluate whether mubritinib reduces hypoxia in GB tumours *in vivo*, we used the GFP hypoxia reporter and conducted intracranial tumour assays by implanting BTSC73 cells, transduced with the GFP-reporter, into the brains of mice. Mice received either vehicle control or mubritinib. At the end of the assay, brain samples from both groups were stained for GFP as a hypoxia indicator (**New Fig 6h**). Mubritinib treatment significantly reduced hypoxia in GB tumours *in vivo* (**New Fig 6i**), suggesting improved oxygen availability for diffusion in GB tumours.

We next assessed whether this increase in oxygen availability would enhance ROS generation in the combined treatment of mubritinib and IR compared to IR or mubritinib alone. We employed H₂DCFDA and MitoSOX Red probes to detect cellular ROS and mitochondrial superoxides, respectively. We observed that the combined IR and mubritinib treatment significantly increased ROS generation compared to either treatment alone in multiple patient-derived BTSCs *in vitro* (**New Figs 6j-m and EV5f-g**).

To evaluate ROS generation *in vivo*, we stained brain tumour xenograft sections from animals treated with mubritinib alone, IR alone, or the combination using the OxyLHC oxidative stress detection kit. Our results showed significantly higher ROS levels in tumours receiving both IR and mubritinib compared to either monotherapy (**New Fig 6n**).

At the reviewer 1 suggestion, we also examined mubritinib's ability to enhance ROS production and sensitize BTSCs to IR under hypoxic conditions (1% O₂). We found that combined mubritinib and IR treatment significantly increased ROS generation in BTSCs compared to either treatment alone under hypoxia (**New Figs EV5h-i**). ELDA analysis further confirmed that mubritinib sensitizes BTSCs to IR under hypoxic conditions (**New Fig EV5j**). Collectively, these findings suggest that OXPHOS inhibition by mubritinib alleviates hypoxia, increasing oxygen availability and enhancing ROS production, which improves the response of BTSCs and GB tumours to IR.

Since DNA damage is central to the effectiveness of radiotherapy, and to address the reviewer's question regarding potential changes in DNA damage, we investigated whether the enhanced IR sensitivity observed with mubritinib is associated with increased DNA damage. To do this, we performed immunofluorescence staining for phosphorylated histone H2A (p-H2AX), a DNA damage marker, in GB tumours. We observed a significant increase in γ -H2AX levels in GB xenografts from animals treated with the combination of IR and mubritinib compared to those receiving either IR or mubritinib alone (**New Fig 5n**).

Next, we assessed whether this increase in DNA damage leads to enhanced apoptosis in GB cells. Cleaved caspase-3 (CC3) staining of GB tumours showed higher levels of CC3 in xenografts treated with both mubritinib and IR (**New Fig 5o**), suggesting that mubritinib enhances IR sensitivity in GB tumours by increasing DNA damage and promoting apoptosis in response to IR.

Minor concerns:

1. Grammar needs attention.

The grammar was checked using the AJE Springer Nature/Curie editing tool.

2. Please remove the overuse of "strikingly", "drastically", "importantly", etc. Most of these descriptions are overstated.

As requested, the wording has been revised.

3. The number of in vivo replicates needs to be addressed.

The number of animals was determined using the resource equation approach, which provides sufficient statistical power while adhering to ethical guidelines by keeping animal numbers within a manageable range.

Based on this approach, the acceptable range of degrees of freedom (DF) for the error term in an analysis of variance (ANOVA) is between 10 and 20. The minimum/maximum number of animals for the experiment was calculated using the equation:

$$n = (\text{DF}/\text{number of experimental groups}) + 1$$

For example, to determine the effect of mubritinib on IR response (with 4 experimental groups), we used $n \geq 5$ mice. The minimum number of animals required based on the resource equation approach was 4 (number of mice per group = $(10/4) + 1 = 3.5$, rounding up to 4).

4. All gene expression data should be deposited. All raw data (numeric) should be made available.

We deposited all the gene expression data in the Gene Expression Omnibus (GEO) data repository, and the data can be accessed under the GEO number GSE253362. All the raw data are made available.

5. Please replace all normalized data with raw data. The statistical testing was likely incorrectly

performed, and this can give rise to false conclusions. I am particularly concerned about the normalization to cell numbers.

In the revised version we replaced the OCR graphs with raw data without normalization to number of cells. All the raw data used to generate the figures throughout the paper are made available.

16th Dec 2024

Dear Dr. Sharanek,

Thank you for submitting your revised study. We have now received the reports from the referees who evaluated your revised manuscript. Referee #3 was unfortunately not available, but referees #1 and #2 also checked your responses to this referee's concerns. As you will see from the reports below, they are satisfied with the revisions, and I will therefore be able to accept your manuscript once the following minor issues are addressed:

1/ Please address referee's 1 comments on further streamlining of the text.

2/ Manuscript text:

- Please note that emails bounced for Samuel Weidd (Sam.Weiss@ucalgary.ca), Johanna Galvis (joahanna.galvis@u-bordeaux.fr), Artee Luchman (Artee.Luchman@ucalgary.ca), and Joris Guyon (Joris.guyon@u_bordeaux.fr).
- Please correct the order of the manuscript sections as follows: Abstract, Keywords, Introduction, Results, Discussion, Methods, Acknowledgements, Disclosure and competing interests statement, References, Figure legends, Tables and their legends, Expanded View Figure legends.
- Please provide up to 5 keywords.
- Methods:
 - o All Materials and Methods need to be described in the main text using our 'Structured Methods' format. According to this format, the Methods section includes a Reagents and Tools Table (listing key reagents, experimental models, software and relevant equipment and including their sources and relevant identifiers) followed by a Methods and Protocols section describing the methods, ideally using a step-by-step protocol format. The aim is to facilitate adoption of the methodologies across labs. Please download and fill our Reagents and Tools Table template (.docx), which you can find in our author guidelines: <https://www.embopress.org/page/journal/14693178/authorguide#structuredmethods>.
 - When submitting your revised manuscript, please do not include the Reagents and Tools Table in the Methods section of the manuscript but upload it as a separate file choosing the file type "Reagent Table".
 - An example of a Method paper with Structured Methods can be found here: <https://www.embopress.org/doi/10.15252/msb.20178071>
 - o Please indicate whether the cells were tested for mycoplasma contamination.
 - o Please provide the strain and origin of the mice, as well as information on sex and age at time of experiments.
 - o Patient material: please include a statement confirming that written informed consent was obtained from all subjects and that the experiments conformed to the principles set out in the WMA Declaration of Helsinki and the Department of Health and Human Services Belmont Report. Please also complete the related sections in the authors' checklist.
 - o Statistics: please add a statement on inclusion/exclusion criteria and complete the checklist accordingly.
 - Data availability: thank you for depositing your sequencing data. Please provide the specific URL for GSE253362 dataset.
 - Please provide a 'Disclosure statement and competing interests' statement: We updated our journal's competing interests policy in January 2022 and request authors to consider both actual and perceived competing interests. Please review the policy <https://www.embopress.org/competing-interests> and update your competing interests if necessary.
 - Acknowledgements: The information provided in the manuscript and the submission system should match (currently, funding information is incomplete in the submission system).

3/ Figures and Appendix:

- EV Datasets: Please add the legends to the corresponding dataset files, in a separate tab/worksheet.
- Appendix: please include a table of content and page numbers and correct the nomenclature to "Appendix Table S1" and "Appendix Table S2".
- Please make sure that all figures/figure panels are referenced in the text, and in chronological order. Currently, Fig. EV1 C-H are called out after Fig. EV2.
- Please address the queries from our copy editors in the figure legends:
 1. Please note that the exact p values are not provided in the legends of figures 1a-c, f, h-j, m; 2g, i; 3k, n; 4i; 5b, d, g, k; 6b, d, k, m; 7a-b, e, b, i, k; 8c-d; EV 2a-b, l-n, p-t, v, x; EV 4a-b; EV 5b-c, f, h-i.
 2. Please indicate the statistical test used for data analysis in the legends of figures EV 4h-i.
 3. Please note that the box plots need to be defined in terms of minima, maxima, centre, bounds of box and whiskers, and percentile in the legends of figures 4e, h; 5g; 7e, i; 8q; EV 4e.

4/ Thank you for providing source data for your figures. Please also provide the completed SD checklist and upload the source data as one (zipped) file per figure.

5/ Please provide 'The paper explained': EMBO Molecular Medicine articles are accompanied by a summary of the articles to emphasize the major findings in the paper and their medical implications for the non-specialist reader. Please provide a draft summary of your article highlighting

6/ I slightly shortened your synopsis text, please let me know if you agree or amend as you see fit:

Using patient-derived and animal models, mubritinib effectively suppressed glioblastoma stem cells and tumours by targeting mitochondrial respiration. Combination with the current standard of care provided further therapeutic advantage, highlighting mubritinib's potential for glioblastoma treatment.

- Mubritinib effectively suppressed glioblastoma stem cell proliferation and self-renewal through the inhibition of mitochondrial respiration.
- A significant benefit in delaying tumour progression and extending survival was observed when mubritinib was combined with the current standard of care.
- Mubritinib alleviated tumour hypoxia, resulting in elevated ROS levels and increased DNA damage, thereby sensitizing tumours to ionizing radiation.
- Mubritinib has a well-tolerated profile and selectively targets glioblastoma stem cells while sparing normal cells.
- These findings highlight mubritinib's promising therapeutic potential as a novel treatment for glioblastoma and warrants its further clinical exploration.

Thank you for providing a nice synopsis picture. I have cropped a small portion (attached) to serve as thumbnail for the table of content on our webpage. Let me know if you agree, or provide a new eTOC (115 pixels x 70 pixels), as changes during proofing are usually not allowed.

7/ As part of the EMBO Publications transparent editorial process initiative (see our Editorial at <http://embomolmed.embopress.org/content/2/9/329>), EMBO Molecular Medicine will publish online a Review Process File (RPF) to accompany accepted manuscripts.

This file will be published in conjunction with your paper and will include the anonymous referee reports, your point-by-point response and all pertinent correspondence relating to the manuscript. Let us know whether you agree with the publication of the RPF and as here, if you want to remove or not any figures from it prior to publication.

I look forward to receiving your revised manuscript.

Yours sincerely,

Lise Roth

To submit your manuscript, please follow this link:
<https://embomolmed.msubmit.net/cgi-bin/main.plex>

***** Reviewer's comments *****

Referee #1 (Comments on Novelty/Model System for Author):

The novelty is slightly reduced by the fact that, while the focus on mitochondrial OXPHOS is timely, the concept itself builds upon established work in other cancer types.

Referee #1 (Remarks for Author):

I congratulate the authors for their endeavours to thoroughly address the points raised by Reviewers N. 1 and N. 3. The revised manuscript demonstrates significant improvements, addressing all major concerns raised in the initial review. The addition of models and experimental analysis of mubritinib response greatly enhance the robustness of the findings. The authors have also provided clearer mechanistic insights into the link between OXPHOS inhibition, hypoxia alleviation, and radiosensitization, which were initially lacking.

While the new data provide strong evidence for mubritinib's effects, the narrative linking mitochondrial reliance, transcriptional subtypes, and sensitivity to mubritinib could be streamlined for better clarity.

Referee #2 (Comments on Novelty/Model System for Author):

The technical quality and novelty is rated high due to rigorous analysis that ensures the reliability and validity of the experimental findings and because the research shows encouraging possibilities for the clinical application of a novel compound, Mubritinib in the treatment of GB. However, the medical impact is rated medium, because while the findings are relevant, their immediate application in clinical settings require further validation in human patients (which is in any case out of the scope of this paper). The model system used, cell lines and mouse models, effectively address the research questions but may not fully capture the complexities of human biology.

Referee #2 (Remarks for Author):

The authors have thoroughly addressed all of the reviewer's concerns in a comprehensive and satisfactory manner. The revisions made have significantly enhanced the quality and clarity of the manuscript, making it a valuable contribution to the field.

Rev_Com_number: RC-2024-02361

New_manu_number: EMM-2024-19652-V2

Corr_author: Sharanek

Title: Exploiting metabolic vulnerability in glioblastoma using a brain-penetrant drug with a safe profile

1/ Please address referee's 1 comments on further streamlining of the text.

This comment is now addressed in the discussion as follow:

“Prior to our study, the impact of mubritinib on BTSCs and GB tumour progression had not been explored. Here, we provide robust evidence that mubritinib alters BTSC growth, stemness, and GB progression by impairing complex I activity and subsequently inhibiting OXPHOS.

We tested mubritinib on a panel of patient-derived BTSCs with diverse genetic backgrounds and transcriptional profiles, demonstrating its effectiveness across all tested cell lines. Notably, we observed variable sensitivity among BTSCs to mubritinib, which correlated with their respiratory capacity. To further understand this variability, we analysed the transcriptional profiles of BTSCs to assess whether sensitivity to mubritinib was linked to specific transcriptional subtypes. Using the Wang classification of GB subtypes (Wang *et al.*, 2017), we found no correlation between sensitivity to mubritinib and either the mesenchymal or classical subtypes. However, the proneural subtype exhibited a negative correlation with mubritinib sensitivity, suggesting that BTSCs with proneural signature may be less susceptible to OXPHOS inhibition. By analysing the correlation between sensitivity to mubritinib and the recently identified metabolic-related transcriptional GB subtypes (Garofano *et al.*, 2021), we observed a potential positive correlation with the mitochondrial GB subtype (Garofano *et al.*, 2021). This subtype, which relies exclusively on OXPHOS for energy production, demonstrates marked vulnerability to OXPHOS inhibitors (Garofano *et al.*, 2021). In line with this, our findings suggest that mubritinib may preferentially target the mitochondrial subtype, although further studies with a larger panel of samples are required to conclusively establish this relationship.”

2/ Manuscript text:

- Please note that emails bounced for Samuel Weiss (Sam.Weiss@ucalgary.ca), Johanna Galvis (joahanna.galvis@u-bordeaux.fr), Artee Luchman (Artee.Luchman@ucalgary.ca), and Joris Guyon (Joris.guyon@u_bordeaux.fr).

The co-authors' emails have been corrected.

- Please correct the order of the manuscript sections as follows: Abstract, Keywords, Introduction, Results, Discussion, Methods, Acknowledgements, Disclosure and competing interests statement, References, Figure legends, Tables and their legends, Expanded View Figure legends.

The order of the manuscript sections has been corrected as requested.

- Please provide up to 5 keywords.

We provided 5 keywords: metabolic reliance, oxidative phosphorylation, radiotherapy, hypoxia, reactive oxygen species.

- Methods:

o All Materials and Methods need to be described in the main text using our 'Structured Methods' format. According to this format, the Methods section includes a Reagents and Tools Table (listing key reagents, experimental models, software and relevant equipment and including their sources and relevant identifiers) followed by a Methods and Protocols section describing the methods, ideally using a step-by-step protocol format. The aim is to facilitate

adoption of the methodologies across labs.

Please download and fill our Reagents and Tools Table template (.docx), which you can find in our author guidelines:

<https://www.embopress.org/doi/10.15252/msb.20178071>

A Reagents and Tools Table, listing key reagents, experimental models, software, relevant equipment, their sources, and identifiers, has been provided, as requested.

o Please indicate whether the cells were tested for mycoplasma contamination.

A sentence indicating that all cell lines were tested negative for mycoplasma is included in page 27.

o Please provide the strain and origin of the mice, as well as information on sex and age at time of experiments.

All these details are now provided on pages 25 and 39 of the manuscript.

o Patient material: please include a statement confirming that written informed consent was obtained from all subjects and that the experiments conformed to the principles set out in the WMA Declaration of Helsinki and the Department of Health and Human Services Belmont Report. Please also complete the related sections in the authors' checklist.

These details are now added in the 'Patient-derived and murine BTSC cultures' paragraph on page 25 and the related section is completed in the authors' checklist.

o Statistics: please add a statement on inclusion/exclusion criteria and complete the checklist accordingly.

This information is now added in the 'Statistical analysis' section on page 42 and completed in the check list.

- Data availability: thank you for depositing your sequencing data. Please provide the specific URL for GSE253362 dataset.

The URL is now added in the "Data and code availability" section.

- Please provide a 'Disclosure statement and competing interests' statement: We updated our journal's competing interests policy in January 2022 and request authors to consider both actual and perceived competing interests. Please review the policy

<https://www.embopress.org/competing-interests> and update your competing interests if necessary.

Disclosure and competing interests statement is now added page 43.

- Acknowledgements: The information provided in the manuscript and the submission system should match (currently, funding information is incomplete in the submission system).

All the funding are now added in the submission system.

3/ Figures and Appendix:

- EV Datasets: Please add the legends to the corresponding dataset files, in a separate tab/worksheet.

Legends are now added.

- Appendix: please include a table of content and page numbers and correct the nomenclature to "Appendix Table S1" and "Appendix Table S2".

A table of content is now added.

- Please make sure that all figures/figure panels are referenced in the text, and in chronological order. Currently, Fig. EV1 C-H are called out after Fig. EV2.

Corrected, all the figure panels are now referenced in the chronological order in the text.

- Please address the queries from our copy editors in the figure legends:

1. Please note that the exact p values are not provided in the legends of figures 1a-c, f, h-j, m; 2g, i; 3k, n; 4i; 5b, d, g, k; 6b, d, k, m; 7a-b, e, b, i, k; 8c-d; EV 2a-b, l-n, p-t, v, x; EV 4a-b; EV 5b-c, f, h-i.

All the exact p-values are now added as requested.

2. Please indicate the statistical test used for data analysis in the legends of figures EV 4h-i.

The statistical test is now added as requested.

3. Please note that the box plots need to be defined in terms of minima, maxima, centre, bounds of box and whiskers, and percentile in the legends of figures 4e, h; 5g; 7e, i; 8q; EV 4e.

The box plots are now defined as requested.

4/ Thank you for providing source data for your figures. Please also provide the completed SD checklist and upload the source data as one (zipped) file per figure.

SD checklist is now added.

5/ Please provide 'The paper explained': EMBO Molecular Medicine articles are accompanied by a summary of the articles to emphasize the major findings in the paper and their medical implications for the non-specialist reader. Please provide a draft summary of your article highlighting

- the medical issue you are addressing,

- the results obtained and

- their clinical impact.

Here is a draft of the 'The Paper Explained' section, which highlights the medical issue, the results obtained, and the clinical impact of this study:

The paper explained

Problem:

Glioblastoma is a highly aggressive brain tumour with limited treatment options and poor survival rates, typically not exceeding 18 months post-diagnosis. Current standard therapies, including surgical excision, radiotherapy, and chemotherapy with temozolomide (TMZ), have shown limited efficacy, largely due to the presence of brain tumour stem cells (BTSCs). These cells possess self-renewal properties and contribute to therapeutic resistance and tumour recurrence. A significant challenge in the field is to develop safe, brain-penetrant treatments that can suppress BTSCs and/or sensitize their response to current standard-of-care therapies.

Results:

Using multiple patient-derived BTSC models, we report that a drug called mubritinib effectively impairs BTSC stemness, self-renewal capacity, and growth. Mechanistic studies revealed that mubritinib targets complex I of the electron transport chain, disrupting mitochondrial respiration. Importantly, restoring mitochondrial respiration via rescue experiments reversed the effects of mubritinib, confirming its mechanism of action. Additionally, we found that mubritinib interferes with the AMPK/P27^{Kip1} pathway, resulting in cell-cycle arrest in BTSCs. *In vivo* pharmacokinetic studies demonstrated that mubritinib efficiently crosses the blood-brain barrier and accumulates in the brain. Using patient-derived xenograft and syngeneic *in vivo* preclinical mouse models, we showed that mubritinib effectively suppresses glioblastoma tumours and extends the lifespan of treated animals. Moreover, we found that mubritinib improves oxygen diffusion in tumours by inhibiting mitochondrial oxygen consumption. This enhanced oxygenation increases glioblastoma tumours' sensitivity to radiotherapy by promoting the generation of reactive oxygen species and enhancing radiotherapy-induced DNA damage. We also demonstrated that mubritinib enhances the DNA-damaging effects of TMZ, making glioblastoma tumours more responsive to this treatment. Finally, *in vitro* and *in vivo* toxicological and behavioral studies revealed that mubritinib is well-tolerated and spares normal cells from damage.

Impact:

This work highlights mitochondrial respiration as a critical metabolic dependency and an exploitable vulnerability in BTSCs. Our findings show that mubritinib efficiently crosses the blood-brain barrier, impairs glioblastoma stem cells and tumours with minimal side effects, positioning it as a promising candidate for glioblastoma treatment. Furthermore, the combinatorial benefits of mubritinib with standard care therapies further underscore its potential clinical impact, encouraging further investigation of this drug in clinical trials for glioblastoma patients.

6/ I slightly shortened your synopsis text, please let me know if you agree or amend as you see fit:

Using patient-derived and animal models, mubritinib effectively suppressed glioblastoma stem cells and tumours by targeting mitochondrial respiration. Combination with the current standard of care provided further therapeutic advantage, highlighting mubritinib's potential for glioblastoma treatment.

- Mubritinib effectively suppressed glioblastoma stem cell proliferation and self-renewal through the inhibition of mitochondrial respiration.
- A significant benefit in delaying tumour progression and extending survival was observed when mubritinib was combined with the current standard of care.
- Mubritinib alleviated tumour hypoxia, resulting in elevated ROS levels and increased DNA damage, thereby sensitizing tumours to ionizing radiation.
- Mubritinib has a well-tolerated profile and selectively targets glioblastoma stem cells while sparing normal cells.
- These findings highlight mubritinib's promising therapeutic potential as a novel treatment for glioblastoma and warrants its further clinical exploration.

Thank you for shortening the text. We agree with the revised synopsis.

Thank you for providing a nice synopsis picture. I have cropped a small portion (attached) to serve as thumbnail for the table of content on our webpage. Let me know if you agree, or provide a new eTOC (115 pixels x 70 pixels), as changes during proofing are usually not

allowed.

Thank you for cropping a portion of the synopsis picture to serve as a thumbnail. However, I could not find the attachment to review it. Any crop from the initial synopsis picture will be fine for us.

7/ As part of the EMBO Publications transparent editorial process initiative (see our Editorial at <http://embomolmed.embopress.org/content/2/9/329>), EMBO Molecular Medicine will publish online a Review Process File (RPF) to accompany accepted manuscripts.

This file will be published in conjunction with your paper and will include the anonymous referee reports, your point-by-point response and all pertinent correspondence relating to the manuscript. Let us know whether you agree with the publication of the RPF and as here, if you want to remove or not any figures from it prior to publication.

Thank you. We agree with the publication of the Review Process File.

12th Jan 2025

Dear Dr. Sharanek,

Thank you for submitting your revised files and please accept my apologies for the delay in getting back to you during this busy time of the year.

I have looked at everything, and there are a few last editorial issues that need to be addressed before I can accept your manuscript:

- Patient material: I understand all patient-derived material has been generated prior to this study and in other laboratories. If this is not the case, please include a statement confirming that the experiments conformed to the principles set out in the WMA Declaration of Helsinki and the Department of Health and Human Services Belmont Report.

- Data availability: thank you for depositing your sequencing data. Please note that datasets must be public before acceptance of the manuscript.

- I added minor modifications to your Paper Explained to shorten it, please let me know if you agree or amend as you see fit:

Problem:

Current standard therapies for glioblastoma, a highly aggressive brain tumor, include surgical excision, radiotherapy, and chemotherapy with temozolomide (TMZ). These treatments have however shown limited efficacy, largely due to the presence of brain tumour stem cells (BTSCs), which possess self-renewal properties. A significant challenge is to develop safe, brain-penetrant treatments that can suppress BTSCs and/or sensitize their response to current standard-of-care therapies.

Results:

Using multiple patient-derived BTSC models, we report that a drug called mubritinib effectively impairs BTSC stemness, self-renewal capacity, and growth. Mechanistic studies revealed that mubritinib targets complex I of the electron transport chain, disrupting mitochondrial respiration. Additionally, we found that mubritinib interferes with the AMPK/P27Kip1 pathway, resulting in cell-cycle arrest in BTSCs. In vivo, mubritinib efficiently crosses the blood-brain barrier and accumulates in the brain, leading to tumour growth suppression and increased lifespan in treated patient-derived xenografted mice and syngeneic preclinical mouse models. Moreover, mubritinib improves oxygen diffusion in tumours by inhibiting mitochondrial oxygen consumption, which ultimately increases glioblastoma tumours' sensitivity to radiotherapy. We also demonstrated that mubritinib enhances the DNA-damaging effects of TMZ, making glioblastoma tumours more responsive to this treatment. Finally, in vitro and in vivo toxicological and behavioral studies revealed that mubritinib is well-tolerated and spares normal cells from damage.

Impact:

This work highlights mitochondrial respiration as a critical metabolic dependency and an exploitable vulnerability in BTSCs. Our findings show that mubritinib efficiently crosses the blood-brain barrier, impairs glioblastoma stem cells and tumours with minimal side effects, positioning it as a promising candidate for glioblastoma treatment. The combinatorial benefits of mubritinib with standard of care therapies further underscore its potential clinical impact.

- I further modified your synopsis, please let me know if you agree or amend as you see fit:

"In patient-derived and animal tumor models, mubritinib suppressed glioblastoma stem cells and tumour growth by targeting mitochondrial respiration. Combination with the current standard of care provided further therapeutic advantage, highlighting mubritinib's potential for glioblastoma treatment.

- Mubritinib effectively suppressed glioblastoma stem cell proliferation and self-renewal through the inhibition of mitochondrial respiration.
- A significant benefit in delaying tumour progression and extending survival was observed when mubritinib was combined with the current standard of care.
- Mubritinib alleviated tumour hypoxia, resulting in elevated ROS levels and increased DNA damage, thereby sensitizing tumours to ionizing radiation.
- Mubritinib has a well-tolerated profile and selectively targets glioblastoma stem cells while sparing normal cells."

Please find attached the cropped portion of your synopsis to serve as thumbnail, let me know if you disagree.

Thank you for bearing with these last requests. I look forward to receiving your revised manuscript.

Yours sincerely,

Lise Roth

Lise Roth, PhD

Senior Editor
EMBO Molecular Medicine

To submit your manuscript, please follow this link:
<https://embomolmed.msubmit.net/cgi-bin/main.plex>

- Patient material: I understand all patient-derived material has been generated prior to this study and in other laboratories. If this is not the case, please include a statement confirming that the experiments conformed to the principles set out in the WMA Declaration of Helsinki and the Department of Health and Human Services Belmont Report.

Yes, all patient-derived materials have been generated prior to this study in other laboratories. A sentence has been added in the beginning of the methods section “Patient-derived and murine BTSC cultures”.

- Data availability: thank you for depositing your sequencing data. Please note that datasets must be public before acceptance of the manuscript.

All the sequencing data are now public.

- I added minor modifications to your Paper Explained to shorten it, please let me know if you agree or amend as you see fit:

Problem:

Current standard therapies for glioblastoma, a highly aggressive brain tumour, include surgical excision, radiotherapy, and chemotherapy with temozolomide (TMZ). These treatments have however shown limited efficacy, largely due to the presence of brain tumour stem cells (BTSCs), which possess self-renewal properties. A significant challenge is to develop safe, brain-penetrant treatments that can suppress BTSCs and/or sensitize their response to current standard-of-care therapies.

Results:

Using multiple patient-derived BTSC models, we report that a drug called mubritinib effectively impairs BTSC stemness, self-renewal capacity, and growth. Mechanistic studies revealed that mubritinib targets complex I of the electron transport chain, disrupting mitochondrial respiration. Additionally, we found that mubritinib interferes with the AMPK/P27Kip1 pathway, resulting in cell-cycle arrest in BTSCs. In vivo, mubritinib efficiently crosses the blood-brain barrier and accumulates in the brain, leading to tumour growth suppression and increased lifespan in treated patient-derived xenografted mice and syngeneic preclinical mouse models. Moreover, mubritinib improves oxygen diffusion in tumours by inhibiting mitochondrial oxygen consumption, which ultimately increases glioblastoma tumours' sensitivity to radiotherapy. We also demonstrated that mubritinib enhances the DNA-damaging effects of TMZ, making glioblastoma tumours more responsive to this treatment. Finally, in vitro and in vivo toxicological and behavioral studies revealed that mubritinib is well-tolerated and spares normal cells from damage.

Impact:

This work highlights mitochondrial respiration as a critical metabolic dependency and an exploitable vulnerability in BTSCs. Our findings show that mubritinib efficiently crosses the blood-brain barrier, impairs glioblastoma stem cells and tumours with minimal side effects, positioning it as a promising candidate for glioblastoma treatment. The combinatorial benefits of mubritinib with standard of care therapies further underscore its potential clinical impact.

For consistency, we corrected “tumor” to “tumour”.

- I further modified your synopsis, please let me know if you agree or amend as you see fit:
"In patient-derived and animal tumour models, mubritinib suppressed glioblastoma stem cells and tumour growth by targeting mitochondrial respiration. Combination with the current standard of care provided further therapeutic advantage, highlighting mubritinib's potential for glioblastoma treatment.

- Mubritinib effectively suppressed glioblastoma stem cell proliferation and self-renewal through the inhibition of mitochondrial respiration.
- A significant benefit in delaying tumour progression and extending survival was observed when mubritinib was combined with the current standard of care.
- Mubritinib alleviated tumour hypoxia, resulting in elevated ROS levels and increased DNA damage, thereby sensitizing tumours to ionizing radiation.
- Mubritinib has a well-tolerated profile and selectively targets glioblastoma stem cells while sparing normal cells."

For consistency, we corrected "tumor" to "tumour".

Please find attached the cropped portion of your synopsis to serve as thumbnail, let me know if you disagree.

It looks good for us.

14th Jan 2025

Dear Dr. Sharanek,

Thank for addressing the last editorial concerns. I am pleased to inform you that your manuscript is accepted for publication and is now being sent to our publisher to be included in the next available issue of EMBO Molecular Medicine!

With kind regards,

Lise Roth

Rev_Com_number: RC-2024-02361

New_manu_number: EMM-2024-19652-V4

Corr_author: Sharanek

Title: Exploiting metabolic vulnerability in glioblastoma using a brain-penetrant drug with a safe profile